# To Each Metric Its Decoding: Post-Hoc Optimal Decision Rules of Probabilistic Hierarchical Classifiers

Roman Plaud [1 2]  Alexandre Perez-Lebel [3 4]  Matthieu Labeau [1]  Antoine Saillenfest [2]  Thomas Bonald [1]

## Abstract

Hierarchical classification offers an approach to incorporate the concept of mistake severity by leveraging a structured, labeled hierarchy. However, decoding in such settings frequently relies on heuristic decision rules, which may not align with task-specific evaluation metrics. In this work, we propose a framework for the optimal decoding of an output probability distribution with respect to a target metric. We derive optimal decision rules for increasingly complex prediction settings, providing universal algorithms when candidates are limited to the set of nodes. In the most general case of predicting a *subset of nodes*, we focus on rules dedicated to the hierarchical $hF_\beta$ scores, tailored to hierarchical settings. To demonstrate the practical utility of our approach, we conduct extensive empirical evaluations, showcasing the superiority of our proposed optimal strategies, particularly in underdetermined scenarios. These results highlight the potential of our methods to enhance the performance and reliability of hierarchical classifiers in real-world applications. The code is available at https://github.com/RomanPlaud/hierarchical_decision_rules

## 1. Introduction

In many real-world classification tasks, the costs associated with misclassification are not uniform (Domingos, 1999; Elkan, 2001). In autonomous driving for instance, mistaking a lamppost for a tree is less critical than mistaking a pedestrian for a tree. One way to model the severity of misclassifications is by organizing classes into a hierarchical tree structure, where classes are grouped into progressively

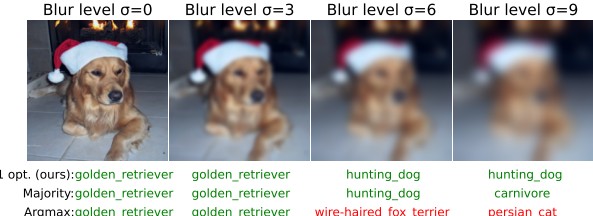

Blur level σ=0    Blur level σ=3    Blur level σ=6    Blur level σ=9

hF1 opt. (ours): golden_retriever    golden_retriever    hunting_dog    hunting_dog
Majority: golden_retriever    golden_retriever    hunting_dog    carnivore
Argmax: golden_retriever    golden_retriever    wire-haired_fox_terrier    persian_cat

Figure 1: **Underdetermination amplifies decision-making disparities between decoding strategies.** Predictions of three decoding strategies based on the probability estimates of a pretrained VGG11 model, for images with blur levels $\sigma \in \{0, 3, 6, 9\}$. Correct predictions are highlighted in green, incorrect in red.

broader superclasses (Deng et al., 2009; Chang et al., 2021). The severity of a misclassification error is then related to the distance between the predicted class and the true class in this hierarchy: the larger the distance, the more severe the error. For any given hierarchy, one can define misclassification costs not only between the leaf nodes (corresponding to the initial classes), but across all nodes of the tree, including internal nodes. It may then be interesting to allow the prediction of internal nodes of the hierarchy. For example, it can be difficult to distinguish between two specific types of cancer that require different treatments. In such a case, the best decision would probably be to predict a more general category – such as simply indicating that the patient has cancer – allowing for a general treatment, effective for different types of cancer.

Given a conditional probability distribution over the hierarchy, the optimal prediction can be theoretically defined as the one minimizing the expected misclassification cost. This principle has long been established: the optimal decision corresponds to minimizing the conditional risk (Duda & Hart, 1973; Berger, 1985; Smith, 2005). In the context of hierarchical classification, misclassification costs are often embedded in evaluation metrics that inherently reflect the hierarchical structure and therefore encode the severity of errors. Numerous works in the literature propose such hierarchical evaluation measures (Kosmopoulos et al., 2014; Amigo & Delgado, 2022). However, the task of optimally decoding from a posterior probability distribution, given a target hierarchical metric, has received comparatively little attention. Some optimal strategies exist for specific metrics (Bi & Kwok, 2012; Ramaswamy et al., 2015) but, in

[1]Institut Polytechnique de Paris [2]Onepoint, 29 rue des Sablons, 75016, Paris, France [3]Soda, Inria Saclay, France [4]Fundamental Technologies, USA. Correspondence to: Roman Plaud <plaud.roman@gmail.com>.

*Proceedings of the 42nd International Conference on Machine Learning*, Vancouver, Canada. PMLR 267, 2025. Copyright 2025 by the author(s).

practice, the decoding strategy often relies on straightforward heuristics, such as selecting the leaf class with the highest posterior probability (by applying argmax over the leaf nodes) or applying a threshold on the probability distribution (e.g., by summing probabilities bottom-up until a threshold is met).

As pre-trained models are ubiquitous in classification tasks, and misclassification costs are not necessarily known during training, our work solely focuses on *post-hoc* decoding, independently of the task of estimating a predictive model.

**Contributions.** We propose a framework for deriving optimal decoding strategies in hierarchical classification by adapting the cost-sensitive decoding framework, originally designed for flat classification (Section 3). Building on this characterization, and assuming access to the true *oracle* probability distribution, we develop tractable algorithms that yield the optimal prediction for a given hierarchical metric, including metrics such as the $hF_\beta$ scores, which generalize the well-known $F_\beta$ scores to hierarchical classification (Section 4). Finally, we validate our theoretical findings in practical situations (Section 5) by demonstrating the superiority of our decoding strategies on probability distributions *estimated* by a diverse set of models, across a wide range of metrics. We further demonstrate that the more underdetermined a classification task is—i.e., the more stochastic the relationship between features and labels—the more crucial it becomes to use our newly introduced decoding algorithms; principle which we illustrate in Figure 1.

## 2. Related work

Determining the action that minimizes the expected cost given a probability distribution over classes has been studied extensively since its introduction in early works (Duda & Hart, 1973; Berger, 1985; Smith, 2005). In practice, and even when misclassification costs are not uniform, this problem is often straightforward to address. For example, in the binary classification case, the optimal decision rule reduces to a simple threshold (Elkan, 2001). In the multiclass setting, a straightforward brute-force approach that computes the expected cost for each possible action remains practical as long as the number of classes is moderate. As a result, cost-sensitive classification (Petrides & Verbeke, 2022) has primarily focused on designing learning algorithms that account for cost non-uniformity during training, either through resampling techniques (Elkan, 2001; Chawla et al., 2002) or developing reweighted losses (Viola & Jones, 2001; Zadrozny & Elkan, 2001; Lin et al., 2017). This is because the challenge often lies in learning a model that produces accurate probability estimates, which can then be used for optimal decision-making. However, there are classification problems where identifying the optimal action is computationally prohibitive, even with access to the or-

acle probability distribution. For example, in multi-label classification, the exponential number of possible predictions makes exhaustive search impractical. Consequently, a significant body of work has emerged that aims to design optimal decoding strategies tailored to specific metrics—i.e., specific definitions of misclassification costs (Lewis, 1995; Dembczynski et al., 2010; Quevedo et al., 2011) and in particular for F-scores (Jansche, 2007; del Coz et al., 2009; Waegeman et al., 2014).

This topic has attracted reasonable interest in the context of hierarchical classification. As the candidate prediction set grows in cardinality, optimal decision-making becomes gradually more difficult. In practice, the literature often considers a specific hierarchical metric and proposes optimal strategies to decode probabilities: for example, Bi & Kwok (2012) develop decoding strategies for family of evaluation measures called HMC-loss, while Ramaswamy et al. (2015) and Cao et al. (2024) address the tree distance loss and its generalized version. Similarly, Karthik et al. (2021) introduce an optimal-decoding framework, restricting to leaf decoding only. However, most of the time, no clear connection is made between the decision-making process from trained classifiers and the actual evaluation measure we aim to optimize. Instead, heuristics strategies are widely used. Selecting the maximum probability over leaf nodes is a common practice but various rules exist, listed in Valmadre (2022). Also, Wang et al. (2017) propose a top-down stopping algorithm, Deng et al. (2012) find a trade-off between specificity and correctness while Jain et al. (2023) propose a strategy that reweigh each leaf probability based on its parent's. While some works advocate for a more systematic evaluation methodology, through bypasssing the decoding step (Plaud et al., 2024), efforts have largely focused on optimizing common metrics (Wu & Palmer, 1994; Zhao et al., 2017) coupled with a generic heuristic strategy. Therefore, our work bridges the gap between decoding and evaluation, two interdependent concepts, by proposing strategies to identify optimal predictions for a given hierarchical metric.

## 3. Problem formulation

Consider a standard single-label classification problem, with $\mathbf{x} \in \mathcal{X}$ the input and $\mathbf{y} \in \{l_1, \ldots, l_K\}$ the label, out of $K$ distinct classes. We denote $\mathbb{P}$ the probability measure governing the joint distribution of the input-label pairs $(\mathbf{x}, \mathbf{y})$.
**Hierarchy.** Additionally, we assume that there exists a directed tree $\mathcal{T}$, referred to as the hierarchy, defined as a set of nodes $\mathcal{N}$ and edges $\mathcal{E}$, such that the leaf nodes of $\mathcal{T}$ correspond to the set of classes $\mathcal{L} = \{l_1, \ldots, l_K\}$ in our single-label classification problem. An edge $e \in \mathcal{E}$ defines a parent-child relation. The unique root node of the tree is denoted by $\mathbf{r}$ and $\mathcal{P}(\mathcal{N})$ is the power set of $\mathcal{N}$.
**Remark.** This design implicitly assumes an exhaustive hierarchy *i.e.* every instance's most specific class is a leaf in

$\mathcal{T}$. However, by adding beforehand a "stopping" children node at any internal position, we could also accommodate datasets whose instance's most specific label lies at an internal node, thereby preserving the full generality of our framework.

**Model.** When designing a hierarchical classifier, one typically uses a probabilistic model $f : \mathcal{X} \to \Delta(\mathcal{L})$ where $\Delta(\mathcal{L})$ is the set of probability distributions over $\mathcal{L}$ and for each $x \in \mathcal{X}$, $f(x) = \hat{p}_x$ is an estimate of the posterior distribution $\mathbb{P}(\mathbf{y}|\mathbf{x} = x)$ over the leaf nodes $\mathcal{L}$.

**Remark.** This model formulation may seem restrictive, but it includes the more general *hierarchy-aware* models capable of predicting probabilities over internal nodes of the hierarchy—not just the leaves—since the leaf distribution can be derived from them.

In this work, we assume that $f$ is given (e.g., a trained neural network), and we have access to the predicted distributions $\hat{p}_{x_1}, \dots, \hat{p}_{x_N}$ over some test samples $x_1, \dots, x_N \in \mathcal{X}$. From these estimations, our objective is to make predictions $h_1, \dots, h_N$ that are as close as possible to the ground-truth labels $y_1, \dots, y_N$, in the sense that it minimizes the misclassification cost for each sample. This cost is defined through an evaluation metric (Definition 3.1). Before introducing it formally, we clarify the nature of the prediction set, denoted $\mathcal{H}$, which play a central role in our framework:

- **Leaf node prediction**: The metric is simply able to compare a leaf $h \in \mathcal{L}$ to the ground-truth leaf label.
- **Internal node prediction** : The metric can compare any node of the hierarchy $h \in \mathcal{N}$ to the ground-truth leaf label.
- **Set of nodes prediction**: The metric can compare a set of nodes $h \in \mathcal{P}(\mathcal{N})$ to the ground-truth leaf label.

**Definition 3.1. Evaluation metric.** An evaluation metric for a set $\mathcal{H}$ of candidate predictions is a function that quantifies the misclassification costs :

$$C : \mathcal{H} \times \mathcal{L} \to \mathbb{R}$$
$$(h, y) \mapsto C(h, y)$$

where $h$ is the prediction and $y$ the ground-truth leaf label.

We list in Appendix C.1 examples of metrics with different candidate sets. To perform the evaluation, a decision rule $\xi$ (Definition 3.2) is required. A decision rule maps a probability distribution to a prediction $h$, as defined below.

**Definition 3.2. Decision rule.** A decision rule for a set $\mathcal{H}$ of candidate predictions is a function $\xi : \Delta(\mathcal{L}) \to \mathcal{H}$.

Importantly, we highlight that even if probabilities are estimated for leaf nodes only, one may want to make a prediction that is not necessarily a leaf. Equipped with the decision rule, the cost of a prediction for the input-label pair $(x, y)$ can be computed as $C(\xi(\hat{p}_x), y)$. The decoding cost

of $\xi$ for a given metric $C$ and model $f$ can then be defined as follows:

**Definition 3.3. Decoding cost.** Given a model $f$, the cost of a decision rule $\xi$ for the metric $C$ is:

$$\mathbb{E}[C(\xi(f(\mathbf{x})), \mathbf{y})]$$

where the expectation is taken with respect to the joint distribution of $(\mathbf{x}, \mathbf{y})$ with probability $\mathbb{P}$.

In practice, the decoding cost is estimated by computing the empirical mean $\frac{1}{N} \sum_{i=1}^{N} C(\xi(\hat{p}_{x_i}), y_i)$ on a test set of $N$ samples. Our focus in this work is on the performance analysis of decision rules with respect to specific evaluation measures. In particular, we examine Bayes-optimal decodings that minimize the decoding cost by design.

### 3.1. Bayes-optimal decoding

Unless stated otherwise, we now assume that we know the true posterior probability distribution: $f^*(x) = \mathbb{P}(\mathbf{y} \mid \mathbf{x} = x) := (p_x(l))_{l \in \mathcal{L}} \in \Delta(\mathcal{L}), \quad \forall x \in \mathcal{X}$.

We formalize the task of deriving optimal predictions from an oracle posterior probability distribution for a given evaluation metric $c$. The goal is to identify a decoding function $\xi_C^* : \Delta(\mathcal{L}) \to \mathcal{H}$ that minimizes the decoding cost:

$$\xi_C^* \in \underset{\xi:\Delta(\mathcal{L})\to\mathcal{H}}{\operatorname{argmin}} \mathbb{E}[C(\xi(f^*(\mathbf{x})), \mathbf{y})].$$

For any $x \in \mathcal{X}$, this objective can be reformulated pointwise as:

$$\xi_C^*(p_x) \in \underset{h \in \mathcal{H}}{\operatorname{argmin}} \mathbb{E}[C(h, \mathbf{y}) \mid \mathbf{x} = x]$$

Then, using the definition of expectation, we can define the optimal decision rule as follows:

**Definition 3.4. Optimal decision rule.** An optimal decision rule for metric $C : \mathcal{H} \times \mathcal{L} \to \mathbb{R}$ is given by $\xi_C^* : \Delta(\mathcal{L}) \to \mathcal{H}$ where

$$\xi_C^*(p) = \underset{h \in \mathcal{H}}{\operatorname{argmin}} \sum_{l \in \mathcal{L}} p(l) C(h, l) \qquad (1)$$

We assume that the optimization problem has a unique solution; further details on this assumption are provided in Appendix C.2. This decision rule minimizes the expected evaluation metric $C$, ensuring that any alternative decoding is suboptimal and results in a higher expected cost. However, even if the misclassification cost matrix $(C(h, l))_{(h,l) \in \mathcal{H} \times \mathcal{L}}$ is precomputed and can be accessed in constant time, computing the risk $\sum_{l \in \mathcal{L}} p(l) C(h, l)$ for all $h \in \mathcal{H}$ via brute-force search has a time complexity of $\mathcal{O}(|\mathcal{H}| \cdot |\mathcal{L}|)$. As a result, when $|\mathcal{H}|$ grows exponentially (for instance, when $\mathcal{H} = \mathcal{P}(\mathcal{N})$), brute-force search quickly becomes infeasible. Even when restricting predictions to internal nodes (i.e.,

$\mathcal{H} = \mathcal{N}$), the computational cost can still be prohibitive for large hierarchies. Alternative strategies are hence necessary.

# 4. Optimal decoding strategies for hierarchical evaluation measures

For certain metrics $C$, it is possible to derive a closed-form solution for $\xi_C^*(p_x)$, which can be analytically defined using only $p_x$. In other cases, an algorithm is needed that computes the optimal prediction $\xi_C^*(p_x)$ with better time complexity than brute-force search. To derive closed-form solutions or algorithms, it is often necessary to compute $p_x(n)$, defined as the bottom-up sum of the probabilities of the leaf descendants of $n$.

**Definition 4.1.** Let $\mathcal{L}(n)$ be the leaf descendants of node $n$ (if $n \in \mathcal{L}$, $\mathcal{L}(n) = \{n\}$). Then we define

$$p_x(n) := \mathbb{P}(\mathbf{y} \in \mathcal{L}(n) | \mathbf{x} = x) = \sum_{l \in \mathcal{L}(n)} p_x(l)$$

We refer to this quantity simply as the probability of $n$. For the sake of readability, we omit the $p_x$ notation and use $p$ instead.

We organize our approach by distinguishing between different types of candidate sets $\mathcal{H}$, proposing adapted strategies for each type of frameworks and metrics.

## 4.1. Leaf candidate set

When $\mathcal{H} = \mathcal{L}$, the evaluation metric compares only the predicted leaf $h$ with the ground-truth leaf $y$. This setup closely resembles a standard cost-sensitive classification problem, with the key difference being that hierarchical information is incorporated into the metric's definition. In this case, the optimal decision rule from Definition 1 is expressed as:

$$\xi_C^*(p) = \underset{h \in \mathcal{L}}{\operatorname{argmin}} \sum_{l \in \mathcal{L}} p(l) C(h, l).$$

Karthik et al. (2021) introduced this exact framework for the metric $C = \eta_{\text{LCA}}$ where $\eta_{\text{LCA}}(h, l)$ is the height in $\mathcal{T}$ of the lowest common ancestor (LCA) between $h$ and $l$. They proved that, if $\max_{l \in \mathcal{L}} p(l) > 0.5$, then the optimal decision rule is the argmax over leaf nodes. In other cases, they perform a brute-force algorithm which results in a $\mathcal{O}(|\mathcal{L}|^2)$ time complexity, assuming that the cost matrix $(\eta_{\text{LCA}}(h, l))_{(h, l) \in \mathcal{L} \times \mathcal{L}}$ is computed beforehand.

Additionally, for the Top1 error metric defined as $\text{Top1}(h, y) = \mathbf{1}(h \neq y)$, the optimal decision rule is the argmax over leaf nodes. We highlight this seemingly trivial result because this decoding strategy is frequently used to evaluate models with metrics beyond Top1, despite not being the optimal decision rule.

## 4.2. Node candidate set

When $\mathcal{H} = \mathcal{N}$, a brute-force algorithm remains feasible: assuming that the cost matrix $(C(h, l))_{(h, l) \in \mathcal{N} \times \mathcal{L}}$ is pre-computed, the overall complexity of the brute-force approach is $\mathcal{O}(|\mathcal{N}| \cdot |\mathcal{L}|)$. However, hierarchies can often contain more than $10,000$ nodes (Ashburner et al., 2000; He & McAuley, 2016), and this algorithm must be applied for each individual data point, significantly impacting time efficiency.

To reduce the time complexity of a standard brute-force search, we focus on a category of metrics that are *hierarchically reasonable*[1] (Definition 4.2). For $n \neq \mathbf{r}$, let $\pi(n)$ denote the unique parent of $n$ in the hierarchy tree $\mathcal{T}$, and recall that $\mathcal{L}(n)$ represents the set of leaf-node descendants of $n$. We then aim to derive constraints on $p(n)$ to determine when a node $n$ or its parent $\pi(n)$ can be ruled out as sub-optimal. Specifically, the intuition is that if $n$ is *sufficiently unlikely*, it cannot be the optimal prediction. Conversely, if $n$ is *too likely*, its parent $\pi(n)$ cannot be the optimal prediction either. Using these constraints, we efficiently filter out a large portion of nodes and apply a simple brute-force algorithm to the remaining candidate set, which overall, drastically reduces time complexity.

**Definition 4.2.** Let $n \in \mathcal{N} \backslash \{\mathbf{r}\}$ and $l \in \mathcal{L}$. We call a metric $C : \mathcal{N} \times \mathcal{L} \to \mathbb{R}$ **hierarchically reasonable** a metric satisfying the following properties:

$$C(n, l) > C(\pi(n), l) \text{ if } l \in \mathcal{L} \backslash \mathcal{L}(n) \quad (2)$$

$$C(n, l) < C(\pi(n), l) \text{ if } l \in \mathcal{L}(n) \quad (3)$$

Equation (2) encodes the intuition that when the ground truth is *Guitar*, predicting *Musical Instrument* is better than predicting *Piano*. Similarly, Equation (3) implies that when the ground truth is *Guitar*, predicting *Musical Instrument* is better than predicting *Object*. Given $p \in \Delta(\mathcal{L})$ and a metric $L$ satisfying Equations (2) and (3), our objective is to find $\xi_C^*(p) \in \mathcal{N}$, the optimal decoding strategy for $p$. As explained, for $n \in \mathcal{N}$, we derive conditions on $p(n)$ for $n$ or $\pi(n)$ to be sub-optimal:

**Lemma 4.3.** *Let $C : \mathcal{N} \times \mathcal{L} \to \mathbb{R}$ be hierarchically reasonable. For $n \in \mathcal{N} \setminus \{\mathbf{r}\}$, define $\delta_{nl}^C = C(n, l) - C(\pi(n), l)$ the node-parent loss difference for label $l$ and*
- $\underline{M}_n = \max_{l \in \mathcal{L} \backslash \mathcal{L}(n)} \delta_{nl} > 0 \quad m_n = -\max_{l \in \mathcal{L}(n)} \delta_{nl} > 0$
- $M_n = -\min_{l \in \mathcal{L}(n)} \delta_{nl} > 0 \quad \underline{m}_n = \min_{l \in \mathcal{L} \backslash \mathcal{L}(n)} \delta_{nl} > 0$

*Then, for $p \in \Delta(\mathcal{L})$, we have:*
- $p(n) > \frac{\underline{M}_n}{\underline{M}_n + m_n} := q_{\max}^C(n) \implies \xi_C^*(p) \neq \pi(n)$
- $p(n) < \frac{m_n}{\underline{m}_n + M_n} := q_{\min}^C(n) \implies \xi_C^*(p) \neq n$

---

[1] We name these conditions after the concept of reasonableness introduced by Elkan (2001)

Proofs are provided in Appendix E.1.1. It naturally follows that $\forall n \in \mathcal{N}$, $q_{\min}^C(n) \leq q_{\max}^C(n)$. Here, $q_{\min}^C(n)$ and $q_{\max}^C(n)$ act as thresholds for $p(n)$, allowing to remove portions of the hierarchy from the search space. Hence, Lemma 4.3 will be at the core of our general algorithm. However, it also allows us to derive closed-form solutions for specific metrics: we recover two known results, which we detail below.

### 4.2.1. APPLICATIONS TO SPECIFIC METRICS

**Tree distance loss**. This metric denoted $\mathrm{DL}(h, y)$ is defined as the length of the shortest path between $h$ and $y$ in $\mathcal{T}$. Here, Lemma 4.3 gives that the optimal decoding corresponds to the deepest node with $p(n) > 0.5$ (Ramaswamy et al., 2015), referred to as *Majority* decoding (Valmadre, 2022).
**Generalized tree distance loss**. This metric is defined as $\mathrm{DL}_c(h, y) = \mathrm{DL}(h, y) + c \cdot d(h)$, where $d(h)$ denotes the depth of node $h$ and $c \geq 0$. Here, Lemma 4.3 shows that the optimal decoding corresponds to the deepest node with $p(n) > \frac{1+c}{2}$ (Cao et al., 2024).
Proofs are detailed in Appendix E.1.2.

### 4.2.2. GENERAL ALGORITHM

In the general case, where $C$ is defined via a cost matrix $(C(h, l))_{(h,l) \in \mathcal{N} \times \mathcal{L}}$ that is *hierarchically reasonable*, we propose an algorithm that improves upon the $\mathcal{O}(|\mathcal{N}| \cdot |\mathcal{L}|)$ brute-force search. The algorithm filters nodes using Lemma 4.3, followed by a brute-force search on the remaining candidates. The size of the candidate set is $\mathcal{O}(d_{\max})$ ($\star$), where $d_{\max}$ is the depth of $\mathcal{T}$, and in practice, $d_{\max} \approx \log(|\mathcal{N}|)$.[2] This leads to the following theorem:

**Theorem 4.4.** *Let $C$ be hierarchically reasonable and $p \in \Delta(\mathcal{L})$, then the optimal decision rule $\xi_C^*(p)$ can be computed with an algorithm of $\mathcal{O}(d_{\max} \cdot |\mathcal{L}| + |\mathcal{N}|)$ time complexity.*

*Sketch of the proof.*
Key elements of the decoding strategy are displayed in Algorithm 1. We give here some general insights on how the algorithm is derived.

1. We begin with computing the probability distribution over the whole hierarchy, by bottom-up summation of leaf probabilities as in Definition 4.1. This can be performed by a single tree traversal whose complexity is $\mathcal{O}(|\mathcal{N}|)$.
2. We compute a candidate set by pruning nodes that do not fulfill conditions enunciated in Lemma 4.3. This results in a candidate set whose cardinality is $\mathcal{O}(d_{\max})$.
3. We perform a brute-force search on the remaining candidate set, whose complexity $\mathcal{O}(d_{\max} \cdot \mathcal{O}(|\mathcal{L}|))$.

---
[2]For a complete binary tree, $d_{\max} = \log_2(|\mathcal{N}| + 1) - 1$.

---

**Algorithm 1** Theorem 4.4

1: **function** FINDOPTIMAL($p_{\text{leaves}}, q_{\max}, q_{\min}$)
2:     $p \leftarrow$ PROBANODES($p_{\text{leaves}}$)     ▷ Definition 4.1
3:     $S \leftarrow$ FINDCANDSET($p, q_{\max}, q_{\min}$) ▷ Lemma 4.3
4:     $n_{\text{opt}} \leftarrow$ BRUTEFORCE($S, p$)        ▷ ($\star$)
5:     **return** $n_{\text{opt}}$
6: **end function**

---

This leads to an overall complexity of $\mathcal{O}(d_{\max} \cdot |\mathcal{L}| + |\mathcal{N}|)$. The detailed proof is provided in Appendix E.1.1.

There exist certain metrics that do not fully satisfy Equation (2): it in fact implies that when the ground truth label is *Piano*, predicting a *Cat* is less severe than predicting a *Persian Cat*; however, some metrics treat these two incorrect predictions as equally severe. As a result, the condition (2) can be transformed to:

$$C(n, l) > C(\pi(n), l) \quad \text{if } l \in \mathcal{L} \backslash \mathcal{L}(n) \text{ and } \mathrm{LCA}(l, n) \neq \mathbf{r},$$
$$C(n, l) = C(\pi(n), l) \quad \text{if } l \in \mathcal{L} \backslash \mathcal{L}(n) \text{ and } \mathrm{LCA}(l, n) = \mathbf{r}.$$
$$(4)$$

We recall that $\mathrm{LCA}(n, l)$ denotes the lowest common ancestor of nodes $n$ and $l$. Under constraints (3) and (4), Theorem 4.4 is still valid. The proof is detailed in Appendix E.1.3. This transformation accommodates metrics commonly used in practice, such as the Wu-Palmer metric (Wu & Palmer, 1994) and its information-theoretic extension (Zhao et al., 2017).

### 4.3. Subset of nodes decoding

The most general candidate prediction set, $\mathcal{P}(\mathcal{N})$, includes all subsets of nodes in $\mathcal{N}$, with cardinality $2^{|\mathcal{N}|}$, representing a flexible framework, where one is allowed to decode one or more nodes. This is particularly useful in scenarios involving hesitation between two or more nodes. However, this generality comes at a significant computational cost: the exponential growth of the space makes a brute-force approach infeasible, with a time complexity of $\mathcal{O}(2^{|\mathcal{N}|} \cdot |\mathcal{L}|)$. From a decoding strategy perspective, one might argue that $\mathcal{P}(\mathcal{N})$ contains redundancy. For example, both {*Piano*} and {*Piano, Musical Instrument*} belong to $\mathcal{P}(\mathcal{N})$, yet they represent inherently the same prediction because *Piano* is included in *Musical Instrument* and therefore more specific. To address this, predictions can be restricted to those that contain **mutually exclusive** nodes. That is, predictions $h \subseteq \mathcal{N}$ such that:

$$\forall (n_1, n_2) \in h, \ n_1 \neq n_2 \Rightarrow n_1 \notin \mathcal{A}(n_2) \text{ and } n_2 \notin \mathcal{A}(n_1)$$

Here, $\mathcal{A}(n)$ denotes the set of ancestors of $n$ in $\mathcal{T}$ (it is defined inclusively, with $n \in \mathcal{A}(n)$). We denote this restricted set as $\mathcal{M}_{\mathcal{T}}(\mathcal{N})$. However, despite this restriction, the cardinality of $\mathcal{M}_{\mathcal{T}}(\mathcal{N})$ remains exponential:

**Proposition 4.5.** *For $\mathcal{T} = (\mathcal{N}, \mathcal{E})$, where each non-leaf node has at least two children, the cardinality of $\mathcal{M}_\mathcal{T}(\mathcal{N})$ satisfies $|\mathcal{M}_\mathcal{T}(\mathcal{N})| \geq 2^{\frac{|\mathcal{N}|}{2}} - 1$.*

The proof is detailed in Appendix E.2.1.
**Remark.** As shown in Aho & Sloane (1973), for a complete binary tree of depth $d$, the cardinality of $\mathcal{M}_\mathcal{T}(\mathcal{N})$ is given by $|\mathcal{M}_\mathcal{T}(\mathcal{N})| = \lfloor q \rfloor^{2^{d+1}} = \lfloor q \rfloor^{|\mathcal{N}|+1}$, where $q \simeq 1.502873$.

We hence focus on designing algorithms that address this issue without relying on brute-force approaches. In such scenarios, it becomes notably harder to design general algorithms that are agnostic to the choice of the metric. For example, Bi & Kwok (2012) derived optimal decoding algorithms for a family of losses called HMC-loss. Here, we focus on the hF$_\beta$-**score**, a family of metrics that balances precision and recall through the parameter $\beta$, which controls the level of emphasis on either metric. It is a natural extension of the $F_\beta$-score in the context of hierarchical classification (Kiritchenko et al., 2006; Kosmopoulos et al., 2014), and it has been shown to exhibit desirable properties (Amigo & Delgado, 2022). This set-based metric is computed by considering the cardinalities of overlap between the predicted set and the ground truth label. Specifically, the predictions and ground truth are augmented with their ancestors as follows:

$$h^{\text{aug}} = \underset{n \in h}{\cup} \mathcal{A}(n) \quad \text{and} \quad y^{\text{aug}} = \mathcal{A}(y)$$

Then, the hF$_\beta$-**score** is defined as follows:

$$\text{hPr}(h, y) = \frac{|h^{\text{aug}} \cap y^{\text{aug}}|}{|h^{\text{aug}}|} \quad \text{hRe}(h, y) = \frac{|h^{\text{aug}} \cap y^{\text{aug}}|}{|y^{\text{aug}}|}$$

$$\text{hF}_\beta(h, y) = \frac{1 + \beta^2}{\frac{\beta^2}{\text{hRe}(h,y)} + \frac{1}{\text{hPr}(h,y)}}$$

As hF$_\beta$ only uses $h^{\text{aug}}$, we get the intuition that various predictions may be redundant; in fact, the search space can be restricted to $\mathcal{M}_\mathcal{T}(\mathcal{N})$ (see Appendix E.2.3 for full explanations). The task, therefore, is to find the optimal decision rule for the metric hF$_\beta$, which is given by $\xi^*_{\text{hF}_\beta} : \Delta(\mathcal{L}) \to \mathcal{M}_\mathcal{T}(\mathcal{N})$, where:

$$\xi^*_{\text{hF}_\beta}(p) = \underset{h \in \mathcal{M}_\mathcal{T}(\mathcal{N})}{\text{argmax}} \sum_{l \in \mathcal{L}} p(l) \cdot \text{hF}_\beta(h, l) \qquad (5)$$

**Remark.** This time, our objective is to maximize expected utility, whereas we previously focused on minimizing expected cost.

Similarly to Lemma 4.3 we first derive a condition on node probability. The intuition is the same: if $p(n)$ is too small, $n$ cannot be an element of $\xi^*_{\text{hF}_\beta}(p)$.

**Lemma 4.6.** *Let $p \in \Delta(\mathcal{L})$ and $n \in \mathcal{N}\backslash\{\mathbf{r}\}$ and $d_{max}(n) = \max_{l \in \mathcal{L}(n)} d(l)$ the leaf nodes maximum depth among leaf descendants of $n$. Then,*
$$p(n) < \frac{1}{1 + \beta^2(d_{max}(n) + 1)} := q^{\text{hF}_\beta}_{min}(n) \implies n \notin \xi^*_{\text{hF}_\beta}(p)$$

We denote $\mathcal{Q}(p) = \{n \in \mathcal{N}, \, p(n) \geq \frac{1}{1 + \beta^2(d_{\max}(n) + 1)}\}$. This condition alone is not sufficient to exhibit a polynomial-time algorithm. Following the approach in Waegeman et al. (2014), we decompose the problem into an outer and inner maximization based on the cardinality of $h^{\text{aug}}$. Combined with Lemma 4.6, this decomposition enables the formulation of a tractable algorithm for computing $\xi^*_{\text{hF}_\beta}(p)$.

**Theorem 4.7.** *Let $d_{max} = \max_{l \in \mathcal{L}} d(l)$ be the leaf nodes maximum depth. Let $p \in \Delta(\mathcal{L})$, then the optimal decision rule $\xi^*_{\text{hF}_\beta}(p)$ can be computed with an algorithm of $\mathcal{O}(d^2_{max} \cdot |\mathcal{N}|)$ time complexity.*

*Sketch of the proof.*
The algorithm is based on the following result:

$$\xi^*_{\text{hF}_\beta}(p) = \underset{1 \leq k \leq |\mathcal{Q}(p)|}{\text{argmax}} \quad \underset{\substack{h \in \mathcal{M}_\mathcal{T}(\mathcal{N}) \\ |h^{\text{aug}}| = k, \, h \subset \mathcal{Q}(p)}}{\text{argmax}} \sum_{n \in \mathcal{N}} \mathbf{1}(n \in h) \Delta^\beta_k(n)$$

where:
$|\mathcal{Q}(p)| = \mathcal{O}(d^2_{\max}), \quad \Delta^\beta_k(n) = \sum_{l \in \mathcal{L}(n)} p(l) \frac{1 + \beta^2}{k + \beta^2(d(l) + 1)}$
The idea is straightforward: for each $k \in \{1, \dots, |\mathcal{Q}(p)|\}$, select the $k$ nodes belonging to $\mathcal{Q}(p)$ with the highest $\Delta^\beta_k(n)$. The algorithm is thus a for-loop with at most $\mathcal{O}(d^2_{\max})$ iterations. Each iteration involves:

- Computing $\left(\Delta^\beta_k(n)\right)_{n \in \mathcal{Q}(p)}$, which can be done via a single tree traversal, i.e., in $\mathcal{O}(|\mathcal{N}|)$.
- Selecting the top-$k$ nodes with highest $\Delta^\beta_k(n)$ which is also $\mathcal{O}(|\mathcal{N}|)$.

Hence, the total complexity is $\mathcal{O}(d^2_{\max} \cdot |\mathcal{N}|)$.
Detailed proofs are given in Appendix E.2.4.

**Summary of Contributions.** In the case where the candidate set is $\mathcal{H} = \mathcal{N}$, we derived an $\mathcal{O}(d_{\max} \cdot |\mathcal{L}| + |\mathcal{N}|)$ algorithm, significantly faster than the $\mathcal{O}(|\mathcal{N}| \cdot |\mathcal{L}|)$ brute-force search, applicable to any *hierarchically reasonable* metric (a condition generally met in practice). In the most general case, where $\mathcal{H} = \mathcal{P}(\mathcal{N})$, the brute-force search becomes exponential, and we derived an algorithm with time complexity $\mathcal{O}(d^2_{\max} \cdot |\mathcal{N}|)$, finding the optimal prediction with respect to the hF$_\beta$ scores. Equipped with these algorithms, we now evaluate their practical efficiency.

# 5. Experiments

## 5.1. Evaluation methodology

Although our proposed decoding strategies are proven to be optimal for the oracle probability distribution, trained models only approximate it. We therefore empirically assess the performance of our strategies against existing decodings. To evaluate the effectiveness of our approach, we first select hierarchical datasets, and define the evaluation metrics to optimize. Using trained models, we infer

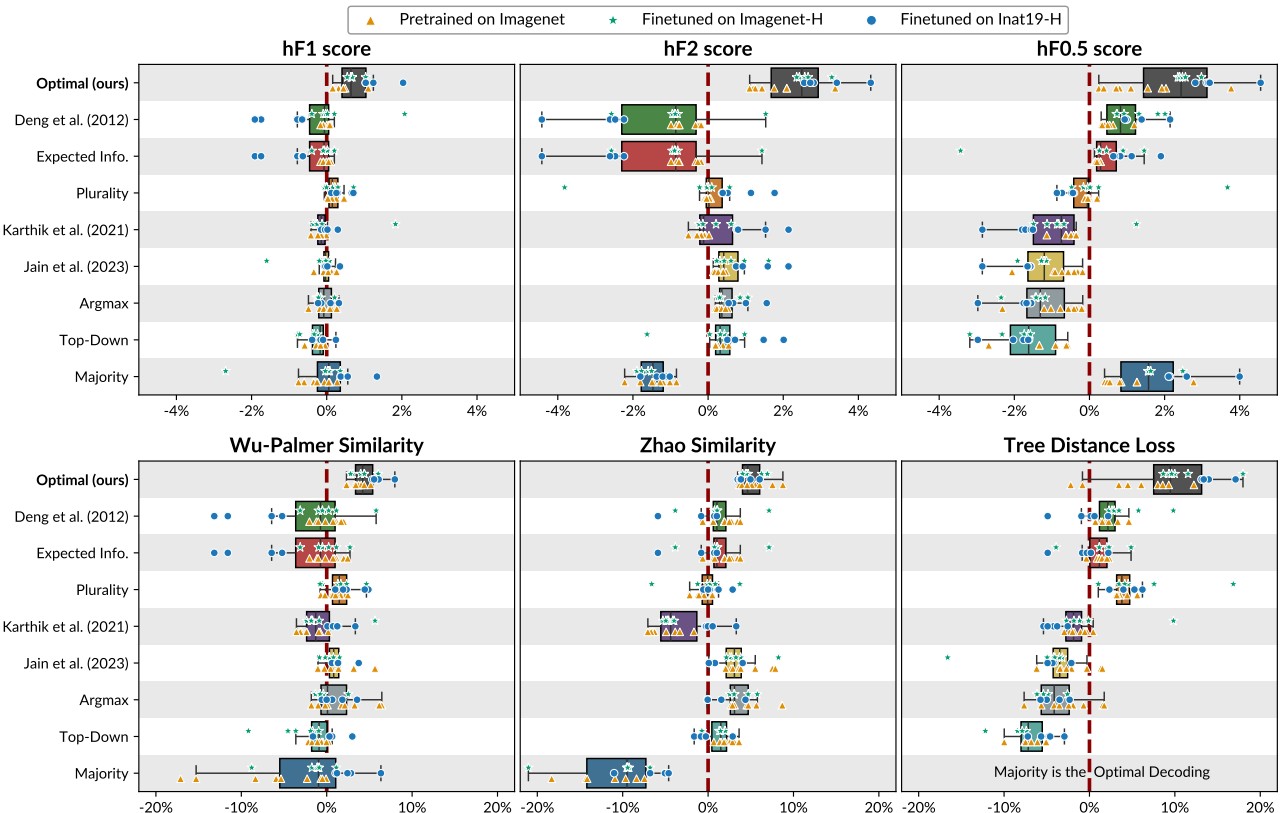

Figure 2: **Relative Gain (in %) of Decoding Strategies.** Each dot represents the relative gain of the optimal strategy vs. the average over all other strategies for a specific metric and model on the test set of a given dataset, with symbols indicating the dataset: ▲ for models pretrained on ImageNet, ★ for models fine-tuned on ImageNet-H, and ● for models fine-tuned on iNat19. Boxplots summarize the distribution of relative gains for each decoding strategy. The higher the gain, the better the decoding strategy. Results are reported for six metrics across various model architectures trained on three datasets.

probability distributions for each input instance in the test set. We then apply our optimal decoding strategy alongside commonly used heuristic strategies for comparison. Finally, we assess and compare the average performance of all decoding strategies across the test set. We list below the datasets, models, metrics and heuristics used. The detailed experimental setup can be found in Appendix D.1.

**Datasets**. While our approach can be applied to any kind of data, we solely rely in this work on computer vision datasets, due to the complexity of their label hierarchy and the availability of appropriate models. Specifically, we utilize TieredImageNet-H and iNat19. Introduced by Bertinetto et al. (2020) together with properly defined hierarchy trees, these datasets, summarized in Table 1, are respectively subsets of ImageNet-1k (Deng et al., 2009) and iNaturalist-19 (Van Horn et al., 2018).

**Models**. Given that TieredImageNet-H is a subset of ImageNet-1K, existing pre-trained models can be directly used. We select 10 models from the PyTorch library, which are listed in Appendix D.1.2. Following the recommenda-

Table 1: Key statistics of the selected datasets.

| Dataset | Nb. of leaves | Nb. of nodes | $d_{\max}$ | Test set size |
|---|---|---|---|---|
| TieredImageNet-H | 608 | 843 | 12 | 15,200 |
| Inat-19-H | 1010 | 1190 | 7 | 40,737 |

tions of Bertinetto et al. (2020), we also fine-tune a ResNet-18 architecture on both datasets using different loss functions. Specifically, we employ the hierarchical cross-entropy loss and soft label methods proposed by Bertinetto et al. (2020), a YOLO-v2 conditional softmax cross-entropy loss (Redmon & Farhadi, 2017), embedding methods from Barz & Denzler (2019), and a standard cross-entropy loss.

**Metrics**. We use several widely adopted metrics in hierarchical contexts. These include Tree Distance Loss, for which an optimal decoding is available in closed form (Ramaswamy et al., 2015), the Wu-Palmer metric (Wu & Palmer, 1994) and its information-theoretic extension (Zhao et al., 2017), for which we apply Theorem 4.4, and $hF_\beta$ for $\beta \in 0.5, 1, 2$, for which we leverage Theorem 4.7. All definitions of metrics are available in Appendix C.1.

**Decoding Heuristics**. A variety of decoding heuristics have been proposed, often using the node information

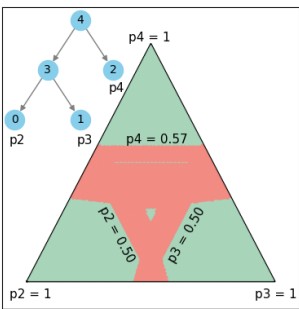

Figure 3: **Agreement map on the simplex of $\mathbb{R}^3$ for two decoding strategies**. For a hierarchy with three leaf nodes displayed in the top-left corner, this figure shows the agreement map between $\mathrm{hF}_1$ and Majority decoding : each point $p$ in the simplex is color-coded, a green dot indicates equality of the two predictions, while a red dot indicates a disagreement.

$I(n) = \log(\frac{|\mathcal{L}|}{|\mathcal{L}(n)|})$ to define these strategies. These can be categorized into leaf-node decoding and any-node decoding strategies. **Leaf-node decoding strategies** include:

1. `argmax` *i.e.* $\xi(p) = \mathrm{argmax}_{l \in \mathcal{L}} \, p(l)$, which selects the leaf node with the highest probability
2. Leaf optimal decoding for $\eta_{\mathrm{LCA}}$ (Karthik et al., 2021)
3. **HiE-self**, $\xi(p) = \mathrm{argmax}_{l \in \mathcal{L}} \, p(\pi(l)) \cdot p(l)$, combining parent probabilities $p(\pi(l))$ with leaf probabilities $p(l)$ (Jain et al., 2023)
4. `top-down`, which selects a leaf node by performing a top-down traversal of the hierarchy, choosing the most likely nodes from the root to the leaf level.

**Any-node decoding strategies** include:

5. **Majority Rule**: $\xi_\tau(p) = \mathrm{argmax}_{n \in \mathcal{N}} \, I(n)$ s.t $p(n) > 0.5$, selecting the most informative node with $p(n) > 0.5$ (Valmadre, 2022)
6. **Plurality Rule**, $\xi(p) = \mathrm{argmax}_{n \in \mathcal{N}} \, I(n)$ s.t. $\forall z \in \mathcal{N} \backslash \mathcal{A}(n), \, p(n) > p(z)$, selecting the most informative label more likely than any non-ancestor (Valmadre, 2022)
7. **Darts Algorithm**, $\xi_\lambda(p) = \mathrm{argmax}_{n \in \mathcal{N}} \, (I(n) + \lambda) p(n)$, balancing information $I(n)$ and confidence $p(n)$ with a parameter $\lambda$ (Deng et al., 2012)
8. **Expected Information**, maximizing expected information by setting $\lambda = 0$ in the Darts Algorithm

Heuristic (7.) requires an held-out set to tune its parameter. Currently, no heuristic explicitly addresses decoding sets of nodes, and our attempt based on Lemma 4.6 yielded poor results.

### 5.2. Analysis

Each plot in Figure 2 illustrates the performance of various decoding algorithms across different datasets and models. A clear takeaway is the consistent superiority of optimal decoding algorithms across all evaluation metrics. It means

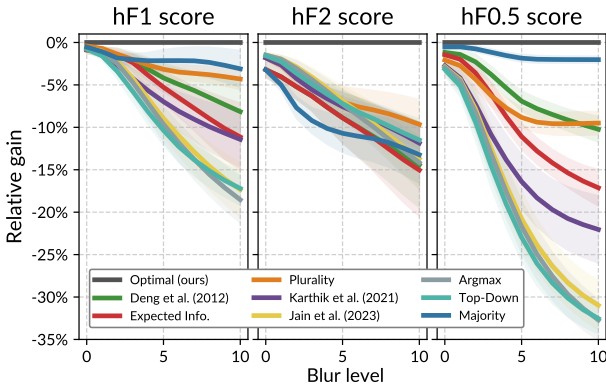

Figure 4: **Impact of blurring on the sub-optimality of heuristic decodings.** This figure shows the relative performance decrease (in %) of heuristics compared to the optimal decoding for the $\mathrm{hF}_\beta$-score, with $\beta \in \{1, 0.5, 2\}$, as a function of the blur level. Results are averaged across multiple models and datasets, with a 95% confidence interval displayed for each decoding strategy.

that, regardless of the dataset, model, or training algorithm used, applying the optimal decoding strategy is always advantageous. Appendix D.2, demonstrates the near-universal superiority of optimal decoding algorithms. Moreover, our decoding algorithms exhibit reasonable time complexity (a few $\mu s$ per sample). More details can be found on this topic in Appendix D.3. For the selected hierarchies, when $\mathcal{H} = \mathcal{N}$, our algorithms are, on average, $60\times$ faster than a brute-force approach. Furthermore, when $\mathcal{H} = \mathcal{P}(\mathcal{N})$, our algorithms efficiently solve problems that are intractable using brute-force methods.

Another noteworthy aspect is the behavior of the non-optimal heuristics. We differentiate between leaf-node heuristics, which always predict a leaf node, and node heuristics, which can predict both internal and leaf nodes. Leaf-node heuristics are more *recall-oriented*, i.e., they predict leaves which increases their ability to capture the correct leaf label but also leads to more errors. In contrast, node heuristics are more conservative and *precision-oriented*, capturing the most specific class less often but with fewer errors. This trade-off is quantified by the $\beta$-score, where $\beta = 2$ gives twice as much importance to recall as to precision. As a result, leaf-node heuristics, such as `argmax`, `top-down`, Karthik et al. (2021), and Jain et al. (2023), are more competitive among non-optimal decoding strategies under this setting. However, when precision becomes more important ($\beta = 0.5$), node heuristics, such as Deng et al. (2012), Expected Information, and Majority, perform better.

Nonetheless, the relative gains of the optimal strategy remain modest: from $1\%$ to $5\%$ for all metrics except Mistake Severity, which achieves a relative gain of $10\%$. In practice, we observe that, for a given data point, optimal predictions often align with heuristic predictions. This is expected: when the model is nearly $100\%$ confident that the label is *Piano*, any reasonably designed heuristic will predict *Piano*.

Disagreements arise when the entropy of the predicted probability distribution increases.

We provide an intuition for this phenomenon in Figure 3. For a simple hierarchy with five nodes, including three leaf nodes, the figure displays the agreement map between the optimal decoding strategy and the **Majority Rule** heuristic. We observe that when the probability distribution is skewed towards one of the leaf nodes, both strategies tend to agree. However, as the distribution approaches the center of the simplex, disagreements become more frequent. This suggests that the more underdetermined the problem becomes, the greater the overall relative performance gain from using optimal decoding strategies. While some domains exhibit greater intrinsic randomness (e.g., medicine), we propose an experiment in which we artificially introduce randomness into a computer vision task by progressively blurring the input images. As the images become increasingly blurred, the image features become less predictive, causing the posterior probability distribution to approach the center of the simplex. We expect to find more disagreements, thus increasing performance gap between optimal decoding and heuristics.

### 5.3. Blurring the images

We propose an experimental setting in which we keep the exact same models introduced in Section 5.1, trained on the same datasets. However, these models are evaluated on modified test sets in which inputs images are progressively blurred. In Figure 1, we illustrate the effect of gradually blurring an input image on the decision-making process. The model used is VGG11 (Simonyan & Zisserman, 2014), and the input image is labeled as a *golden retriever*. Under the $hF_1$-score optimal decision rule, the prediction transitions from *golden retriever* to *hunting dog*, which remains correct but less specific. A similar pattern is observed for majority decoding; however, at the highest level of blurring, the prediction becomes *carnivore* reflecting an even coarser classification. In contrast, `argmax` decoding consistently produces highly specific predictions but fails on the last two levels of blurring, yielding incorrect results. As decodings tend to disagree more, we expect that relative performance of heuristic decodings will gradually decrease with the level of blur, relatively to optimal one. Figure 4 illustrates the relative drop in performance compared to the optimal decoding strategies for $hF_\beta$-scores, averaged across datasets and models. As expected, the blurrier the image, the greater the relative decrease in performance. This confirms our intuition: for an underdetermined problem, it is important to use appropriate decoding strategies when aiming at optimizing a target hierarchical metric.

## 6. Conclusion

In this work, we addressed the challenge of optimal decoding in hierarchical classification tasks. Given a predefined hierarchical evaluation metric and a posterior probability distribution over leaf nodes, our goal was to identify the best possible prediction. To this end, we developed universal algorithms for hierarchically reasonable metrics when the prediction set is restricted to the nodes of the hierarchy. Additionally, we introduced a decoding algorithm specifically tailored to $hF_\beta$-scores . Our empirical results demonstrated the effectiveness of these optimal decoding strategies, specifically in underdetermined classification tasks.

Future work could explore extending our framework to non-tree hierarchies. Additionally, this study does not tackle the challenge of accurately predicting the posterior probability distribution. A promising direction would be to incorporate cost-sensitive learning during training, as suggested in prior work (Ramaswamy et al., 2015; Cao et al., 2024), or to develop post-hoc recalibration methods that provide guarantees on the estimated probability estimation.

## Impact Statement

This paper presents work whose goal is to advance the field of Machine Learning. There are many potential societal consequences of our work, none which we feel must be specifically highlighted here.

## Acknowledgments

The authors would like to thank Clémence Grislain and Nathanaël Beau for their thorough review of an earlier version of this paper and their valuable feedback. They also thank Jean Vassoyan and Pirmin Lemberger for insightful discussions during the development of this work.

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

# Appendices

## A. Notations

We display in Table A.1, the list of notations we use throughout the manuscript.

| Math Symbol | Domain | Description |
|:---:|:---:|:---:|
| $\mathcal{N}$ | | Set of nodes in the hierarchy |
| $\mathcal{L}$ | | Set of leaf nodes in the hierarchy |
| $\mathcal{E}$ | $\mathcal{N} \times \mathcal{N}$ | Set of edges |
| $\mathcal{T} = (\mathcal{N}, \mathcal{E})$ | | Hierarchy Tree |
| $\mathcal{P}(\mathcal{N})$ | | Power set of $\mathcal{N}$ |
| $\mathcal{X}$ | | Input domain |
| $\mathbf{x}$ | $\mathcal{X}$ | Random variable representing the input features |
| $\mathbf{y}$ | $\mathcal{L}$ | Random variable representing the label |
| $\mathbb{P}$ | $\mathcal{X} \times \mathcal{L} \to [0,1]$ | Joint probability distribution. |
| $n$ | $\mathcal{N}$ | A node |
| $l$ | $\mathcal{L}$ | A leaf node |
| $\mathbf{r}$ | $\mathcal{N}$ | The root node |
| $\pi(n)$ | $\mathcal{N}$ | unique parent of $n$ in $\mathcal{T}$ |
| $\mathcal{C}(n)$ | $\mathcal{P}(\mathcal{N})$ | Set of children of $n$ in $\mathcal{T}$ |
| $\mathcal{A}(n)$ | $\mathcal{P}(\mathcal{N})$ | Set of ancestors of $n$ in $\mathcal{T}$ (defined inclusively) |
| $\mathcal{L}(n)$ | $\mathcal{P}(\mathcal{N})$ | Set of leaf descendants of $n$ |
| $I(n)$ | $\mathbb{R}$ | $=\log(\frac{|\mathcal{L}|}{|\mathcal{L}(n)|})$ Information of node $n$ |
| $d(n)$ | $\mathbb{N}$ | Depth of node $n$ in $\mathcal{T}$ 
 Defined as the length of the shortest path between $n$ and $\mathbf{r}$ in $\mathcal{T}$ |
| $d_{\max}(n)$ | $\mathbb{N}$ | $= \max_{l \in \mathcal{L}(n)} d(l)$. Max depth of leaf descendants of $n$ |
| $d_{\max}$ | $\mathbb{N}$ | $=\max_{l \in \mathcal{L}} d(l)$. Max depth of leaf nodes |
| $d_{\min}$ | $\mathbb{N}$ | $=\min_{l \in \mathcal{L}} d(l)$. Min depth of leaf nodes |
| LCA | $\mathcal{N}^2 \to \mathcal{N}$ | Lowest common ancestor of two nodes. |
| $\Delta(\mathcal{L})$ | | Simplex over $\mathcal{L}$ |
| $\mathcal{H}$ | $\{\mathcal{L}, \mathcal{N}, \mathcal{P}(\mathcal{N})\}$ | Set of candidate prediction |
| $h$ | $\mathcal{H}$ | A prediction. |
| $C$ | $\mathcal{H} \times \mathcal{L} \to \mathbb{R}$ | An evaluation metric that compare a prediction $h$ to a ground truth $l$. |
| $\xi$ | $\Delta(\mathcal{L}) \to \mathcal{H}$ | A decision rule |
| $\xi_C^*$ | $\Delta(\mathcal{L}) \to \mathcal{H}$ | $=\operatorname*{argmin}_{h \in \mathcal{H}} \sum_{l \in \mathcal{L}} p(l) C(h, l)$ 
 Optimal decision rule for metric $C$ and proba $p$ |
| $p_x(l)$ | $[0,1]$ | $= \mathbb{P}(l \mid \mathbf{x} = x)$ 
 Posterior probability of class $l$ for input $x$ |
| $p_x(n)$ | $[0,1]$ | $= \sum_{l \in \mathcal{L}(n)} \mathbb{P}(l \mid \mathbf{x} = x)$ 
 Posterior probability of class $n$ for input $x$ |
| $h^{\text{aug}}$ | $\mathcal{P}(\mathcal{N})$ | $= \bigcup_{n \in h} \mathcal{A}(n)$ |
| $\mathcal{M}_{\mathcal{T}}(\mathcal{N})$ | $\mathcal{P}(\mathcal{P}(\mathcal{N}))$ | $= \{h \in \mathcal{P}(\mathcal{N}), n_1, n_2 \in h \implies \text{LCA}(n_1, n_2) = \mathbf{r}\}$ |
| $\mathcal{U}_{\mathcal{T}}(\mathcal{N})$ | $\mathcal{P}(\mathcal{P}(\mathcal{N}))$ | $= \{h^{\text{aug}}, h \in \mathcal{M}_{\mathcal{T}}(\mathcal{N})\}$ |

Table A.1: Notations used in the manuscript

# B. Limitations

While our algorithms demonstrate significant improvements over straightforward heuristics in specific contexts, it is natural to ask whether these empirical findings extend to other types of data. In this work, we focus on image datasets due to the abundance of pretrained models and well-established benchmarks in the vision domain. Exploring other modalities -such as text data- remains an important direction for future work, but falls outside the scope of this work.

Another important aspect we did not address is the accurate estimation of the posterior probability distribution. As discussed in the conclusion, a promising direction for future work is to incorporate hierarchical cost-sensitive learning during training or to develop post-hoc hierarchical recalibration methods that could offer theoretical guarantees on the quality of the estimated probabilities.

# C. Additional Content

## C.1. Metric Defintion

We recall that an evaluation measure for a set $\mathcal{H}$ of candidate predictions is a function that quantifies the misclassification costs as follows.

**Definition C.1. Evaluation Measure.** An evaluation metric for a set $\mathcal{H}$ of candidate predictions is a function that quantifies the misclassification costs :

$$C : \mathcal{H} \times \mathcal{L} \to \mathbb{R}$$
$$(h, y) \mapsto C(h, y)$$

where $h$ is the prediction and $y$ the ground-truth leaf label.

### C.1.1. $\mathcal{H} = \mathcal{L}$

Here, we provide examples of metrics for the case when $\mathcal{H}_e = \mathcal{L}$:

- **Lowest Common Ancestor (LCA) Height:**

$$\eta_{\mathrm{LCA}}(h, y) = \text{Height of the lowest common ancestor between } h \text{ and } y \text{ in } \mathcal{T}.$$

- **Top-1 Error:**

$$\mathrm{Top1}(h, y) = \mathbf{1}(h \neq y),$$

where $\mathbf{1}(\cdot)$ is the indicator function.

### C.1.2. $\mathcal{H} = \mathcal{N}$

For $\mathcal{H} = \mathcal{N}$, we list example of used metrics:

- **Tree Distance Loss** (Ramaswamy et al., 2015):

$$\mathrm{DL}(h, y) = \text{Length of the shortest path between } h \text{ and } y \text{ in } \mathcal{T}.$$

- **Generalized Tree Distance Loss** (Cao et al., 2024):

$$\mathrm{DL}_c(h, y) = \mathrm{DL}(h, y) + c \cdot d(h)$$

where $d(h)$ is the depth of node $h$ in $\mathcal{T}$.
- **Wu-Palmer Similarity** (Wu & Palmer, 1994):

$$\mathrm{WP}(h, y) = \frac{2 \cdot d(\mathrm{LCA}(h, y))}{d(h) + d(y)}$$

- **Zhao Similarity** (Zhao et al., 2017):

$$\mathrm{ZS}(h, y) = \frac{2 \cdot I(\mathrm{LCA}(h, y))}{I(h) + I(y)}$$

where $I(\cdot)$ represents the information content of a node.

### C.1.3. $\mathcal{H} = \mathcal{P}(\mathcal{N})$

For $\mathcal{H}_e = \mathcal{P}(\mathcal{N})$, we list below example of such metrics:

- **Hierarchical F-score ($\mathrm{hF}_\beta$):**

$$\mathrm{hF}_\beta(h, y) = \frac{1 + \beta^2}{\frac{\beta^2}{\mathrm{hRe}(h,y)} + \frac{1}{\mathrm{hPr}(h,y)}},$$

  where

$$\mathrm{hPr}(h, y) = \frac{|h^{\mathrm{aug}} \cap y^{\mathrm{aug}}|}{|h^{\mathrm{aug}}|}, \quad \mathrm{hRe}(h, y) = \frac{|h^{\mathrm{aug}} \cap y^{\mathrm{aug}}|}{|y^{\mathrm{aug}}|}.$$

- **Hamming Loss:**

$$\mathrm{HL}(h, y) = \frac{|(h^{\mathrm{aug}} \cup y^{\mathrm{aug}}) \setminus (h^{\mathrm{aug}} \cap y^{\mathrm{aug}})|}{|y^{\mathrm{aug}}|}.$$

- **Jaccard Similarity:**

$$\mathrm{Jacc}(h, y) = \frac{|h^{\mathrm{aug}} \cap y^{\mathrm{aug}}|}{|h^{\mathrm{aug}} \cup y^{\mathrm{aug}}|}.$$

### C.2. On the Uniqueness of $\xi_C^*(p)$

In the main manuscript, we defined the optimal decision rule for $p \in \Delta(\mathcal{L})$ and an evaluation measure $C$ as follows:

**Definition C.2. Optimal Decision Rule.** The optimal decision rule for the metric $C : \mathcal{H} \times \mathcal{L} \to \mathbb{R}$ is given by $\xi_C^* : \Delta(\mathcal{L}) \to \mathcal{H}$, where

$$\xi_C^*(p) = \underset{h \in \mathcal{H}}{\mathrm{argmin}} \sum_{l \in \mathcal{L}} p(l) C(h, l). \tag{6}$$

First, note that the $\mathrm{argmin}$ is well-defined, as $\mathcal{H}$ is non-empty and finite, ensuring the existence of at least one minimizer. However, we have no theoretical guarantee that $\xi_C^*(p)$ is uniquely defined, since multiple elements $h \in \mathcal{H}$ may attain the minimum. In practice, this ambiguity is not problematic. If multiple optimal predictions exist, one can simply select one at random, as they all yield the same expected risk. When dealing with expected probabilities this random can influence the estimation of the performance, however, such cases are rare in real-world scenarios. When working with estimated probabilities, ties between optimal predictions never occur.

# D. Detailed Experiments

## D.1. Experiment setup

### D.1.1. DATASETS

We give here additional details on how datasets were constructed.

**tieredImageNet-H**. Original tieredImageNet, was introduced by Ren et al. (2018) for few-shot classification. Original version features disjoint class splits based on the WordNet hierarchy and was designed to assess few-shot classifiers. Then Bertinetto et al. (2020) adapted it for standard image classification, resampling it to include all classes across train, validation, and test splits. They chose this dataset because it covered a large part of the $1,000$ classes of ImageNet. Additionally, they slightly modified the WordNet graph to turn it into a tree. This resulted in a tree of 843 nodes, including 608 leaf nodes and a depth of 12. This version is referred to as tieredImageNet-H.

**iNat19**. iNaturalist-19 (Van Horn et al., 2018) is a dataset of organism images primarily used for evaluating fine-grained visual categorization methods. It was introduced with hierarchical species relationships, providing an 8-level complete tree spanning 1010 leaf node classes which was directly usable. Since test set labels are not public, Bertinetto et al. (2020) re-sampled it into three splits (70% training, 15% validation, 15% test) from the original train and validation sets.

### D.1.2. MODELS

**Pretrained Models.** We leverage 10 pretrained models from the PyTorch library, including Swin Transformer V2 (Liu et al., 2022a), VGG-11 (Simonyan & Zisserman, 2014), AlexNet (Krizhevsky et al., 2012), EfficientNet V2-S (Tan & Le, 2021), ConvNeXt-Tiny (Liu et al., 2022b), ResNet-18 (He et al., 2016), DenseNet-121 (Huang et al., 2017), Vision Transformer (ViT-B/16) (Dosovitskiy et al., 2021), and Inception V3 (Szegedy et al., 2016). While these models could be directly evaluated on *ImageNet-1k* using the original *WordNet* hierarchy, we opted to evaluate them on *tieredImageNet-H*, the aformentionned subset of *ImageNet-1k*. Unlike *ImageNet-1k*, *tieredImageNet-H* spans 608 classes instead of the original $1,000$. To generate probability distributions over the 608 leaf nodes of the *tieredImageNet-H* hierarchy, we adopted a straightforward approach: for each model and input, we extracted the logits corresponding to the 608 leaf nodes and applied a softmax function. This process yields a valid probability distribution over the leaf nodes.

**Finetuned Models.** As explained in the core of the manuscript we follow the recommendations of Bertinetto et al. (2020) and follow-up works (Karthik et al., 2021; Jain et al., 2023; Garg et al., 2022; Valmadre, 2022) by finetuning a Resnet-18 architecture with various learning strategies. We detail them here. Bertinetto et al. (2020) propose two hierarchy-aware loss modifications: *Hierarchical Cross-Entropy* weights misclassifications based on their distance in the class hierarchy (it has an hyperparameter $\alpha > 0$), while *Hierarchical Smoothing* assigns non-zero probabilities to related classes to smooth the target distribution. We also use the YOLO-v2 conditional softmax cross-entropy loss (Redmon & Farhadi, 2017), which computes localized softmax. Additionally, we utilize embedding methods from Barz & Denzler (2019), which map class labels into a embedding space to capture semantic relationships between classes. Finally, we use also a standard cross-entropy loss.

### D.1.3. METRICS

As briefly explained in the core of the text. We use 6 different widely used metrics. We list them below together with their optimal decoding strategy.

- **Tree Distance Loss:**
$$\mathrm{DL}(h,y) = \text{Length of the shortest path between } h \text{ and } y \text{ in } \mathcal{T}.$$

 *Optimal Decoding Strategy: Majority Decoding :*
$$\xi_{\mathrm{DL}}(p) = \underset{n \in \mathcal{N}}{\mathrm{argmax}} \; I(n) \;\; s.t \;\; p(n) \geq 0.5 \;\; \text{(Ramaswamy et al., 2015)}$$

- **Wu-Palmer Similarity (WP):**
$$\mathrm{WP}(h,y) = \frac{2 \cdot d(\mathrm{LCA}(h,y))}{d(h) + d(y)},$$

 *Optimal Decoding Strategy:* Use Theorem 4.4.
- **Zhao Similarity (ZS):**
$$\mathrm{ZS}(h,y) = \frac{2 \cdot I(\mathrm{LCA}(h,y))}{I(h) + I(y)}.$$

*Optimal Decoding Strategy:* Use Theorem 4.4.

- **Hierarchical F-Score for** $\beta \in \{0.5, 1, 2\}$

$$\mathrm{hF}_\beta(h, y) = \frac{1 + \beta^2}{\frac{\beta^2}{\mathrm{hRe}(h,y)} + \frac{1}{\mathrm{hPr}(h,y)}},$$

*Optimal Decoding Strategy:* Use Theorem 4.7.

### D.1.4. DECODING HEURISTICS

A variety of decoding heuristics have been proposed to leverage class hierarchy information during decoding. These heuristics can be categorized into strategies for decoding **leaf nodes** and those for decoding **any node**.

**Leaf-Node Decoding Strategies:** These heuristics focus on selecting a leaf node from the hierarchy:

- Leaf Argmax:
$$\xi(p) = \operatorname*{argmax}_{l \in \mathcal{L}} p(l),$$

- Karthik et al. (2021). This method optimally decode for the $\eta_{\mathrm{LCA}}$ metric.
- HiE-self (Jain et al., 2023):
$$\xi(p) = \operatorname*{argmax}_{l \in \mathcal{L}} p(\pi(l)) \cdot p(l),$$

**Any-Node Decoding Strategies:** These heuristics allow for selecting nodes at any level of the hierarchy:

- Confidence Threshold (Valmadre, 2022):

$$\xi_\tau(p) = \operatorname*{argmax}_{n \in \mathcal{N}} I(n) \quad \text{s.t. } p(n) > \tau$$

- Majority Rule:
$$\text{Confidence Threshold with } \tau = 0.5$$

- Plurality Rule (Valmadre, 2022):

$$\xi(p) = \operatorname*{argmax}_{n \in \mathcal{N}} I(n) \quad \text{s.t. } \forall z \in \mathcal{N} \setminus \mathcal{A}(n), \ p(n) > p(z),$$

- Darts Algorithm (Deng et al., 2012):

$$\xi_\lambda(p) = \operatorname*{argmax}_{n \in \mathcal{N}} (I(n) + \lambda) \cdot p(n),$$

- Expected Information:
$$\text{Darts Algorithm with } \lambda = 0$$

### D.2. Detailed results of decoding performance

In this section we provide detailed results about the performance of all decoding strategies for all models, datasets and metrics. We display it Table D.1 the performance of each decoding and for each model and dataset for the **Mistake Severity** metric, in Table D.2 for the **Wu-Palmer similarity** metric, in Table D.3 for the **Zhao similarity** metric and in Tables D.4, D.5 and D.6 the results for hF$_\beta$-score for $\beta \in \{0.5, 1, 2\}$

### D.3. Time Computation Analysis

In Table D.7, we provide insights into the time taken by different decoding strategies, with a particular focus on the time required for optimal decoding strategies introduced by our newly proposed algorithms. As mentioned in the main text, all optimal decoding strategies exhibit reasonable inference times. The worst-case scenario is observed for the optimal decoding of the Zhao similarity metric on the iNat19 dataset, with an average decoding time of approximately 13 milliseconds per sample. Beyond this, we note that for all hF$_\beta$ scores, the inference time is around 1 millisecond per input sample.

| | | Optimal (ours) | Deng et al. | Expected Info Valmadre | Plurality Valmadre | Karthik et al. | Jain et al. | Argmax | Top-Down |
|---|---|---|---|---|---|---|---|---|---|
| tieredImageNet-H | Barz & Denzler | **4.18** | 4.54 | 5.13 | 4.23 | 4.54 | 5.65 | 5.66 | 6.47 |
| | Cross-entropy | **1.61** | 1.71 | 1.74 | 1.69 | 1.77 | 1.82 | 1.83 | 1.88 |
| | Hxe ($\alpha = 0.1$) | **1.62** | 1.71 | 1.73 | 1.71 | 1.78 | 1.82 | 1.85 | 1.89 |
| | Hxe ($\alpha = 0.6$) | **1.96** | 2.14 | 2.14 | 2.04 | 2.22 | 2.26 | 2.3 | 2.42 |
| | Soft-labels | **1.59** | 1.71 | 1.73 | 1.68 | 1.75 | 1.82 | 1.84 | 1.86 |
| | Yolo-V2 | **1.79** | 1.84 | 1.86 | 1.92 | 1.99 | 1.99 | 1.98 | 2.14 |
| iNat19 | Cross-entropy | **1.63** | 1.84 | 1.85 | 1.79 | 1.91 | 1.92 | 1.95 | 1.93 |
| | Hxe ($\alpha = 0.1$) | **1.58** | 1.80 | 1.81 | 1.74 | 1.87 | 1.87 | 1.88 | 1.88 |
| | Hxe ($\alpha = 0.6$) | **1.96** | 2.33 | 2.33 | 2.2 | 2.41 | 2.41 | 2.42 | 2.42 |
| | Soft-labels | **1.68** | 1.99 | 1.99 | 1.8 | 1.95 | 1.94 | 1.94 | 1.95 |
| | Yolo-V2 | **1.68** | 1.86 | 1.86 | 1.86 | 1.99 | 1.97 | 1.96 | 2.02 |
| tieredImageNet-H | Alexnet | **2.12** | 2.29 | 2.34 | 2.26 | 2.4 | 2.51 | 2.54 | 2.59 |
| | Convnext Tiny | 0.83 | 0.80 | 0.80 | **0.79** | 0.83 | 0.81 | 0.81 | 0.86 |
| | Densenet121 | **1.19** | 1.27 | 1.28 | 1.24 | 1.27 | 1.32 | 1.32 | 1.34 |
| | Efficientnet_v2_s | 0.74 | **0.72** | **0.72** | **0.72** | 0.76 | 0.73 | 0.73 | 0.78 |
| | Inception_v3 | **1.05** | 1.07 | 1.09 | 1.07 | 1.12 | 1.13 | 1.13 | 1.18 |
| | Resnet18 | **1.41** | 1.52 | 1.53 | 1.49 | 1.54 | 1.58 | 1.59 | 1.63 |
| | Swin_v2_t | **0.81** | 0.83 | 0.83 | 0.82 | 0.85 | 0.84 | 0.84 | 0.88 |
| | Vgg11 | **1.41** | 1.50 | 1.51 | 1.47 | 1.53 | 1.57 | 1.60 | 1.62 |
| | Vit_b_16 | **0.87** | 0.88 | 0.89 | 0.88 | 0.92 | 0.92 | 0.92 | 0.94 |

Table D.1: Performance on **Mistake Severity** metric of different decoding strategy for various models trained on various datasets

| | | Optimal (ours) | Deng et al. | Expected Info Valmadre | Plurality Valmadre | Karthik et al. | Jain et al. | Argmax | Top-Down | Majority |
|---|---|---|---|---|---|---|---|---|---|---|
| tieredImageNet-H | Barz & Denzler | 33.71 | **33.15** | 34.09 | 35.17 | 33.19 | 34.85 | 34.83 | 37.7 | 37.59 |
| | Cross-entropy | **11.77** | 12.30 | 12.30 | 12.07 | 12.45 | 12.27 | 12.29 | 12.43 | 12.36 |
| | Hxe ($\alpha = 0.1$) | **11.96** | 12.27 | 12.27 | 12.15 | 12.49 | 12.21 | 12.39 | 12.5 | 12.42 |
| | Hxe ($\alpha = 0.6$) | **13.88** | 15.07 | 15.07 | 14.06 | 14.89 | 14.62 | 14.77 | 15.14 | 14.52 |
| | Soft-labels | **11.70** | 12.28 | 12.28 | 11.92 | 12.27 | 12.27 | 12.32 | 12.27 | 12.27 |
| | Yolo-V2 | **13.27** | 13.48 | 13.48 | 13.67 | 13.90 | 13.43 | 13.31 | 14.16 | 13.82 |
| iNat19 | Cross-entropy | **13.14** | 14.47 | 14.47 | 13.59 | 13.68 | 13.72 | 13.9 | 13.79 | 13.70 |
| | Hxe ($\alpha = 0.1$) | **12.74** | 14.21 | 14.21 | 13.17 | 13.36 | 13.37 | 13.45 | 13.40 | 13.11 |
| | Hxe ($\alpha = 0.6$) | **16.17** | 19.18 | 19.18 | 16.72 | 17.21 | 17.21 | 17.31 | 17.31 | 16.42 |
| | Soft-labels | **13.57** | 16.0 | 16.0 | 13.72 | 13.92 | 13.87 | 13.89 | 13.96 | 14.19 |
| | Yolo-V2 | **13.54** | 14.87 | 14.87 | 14.09 | 14.2 | 14.06 | 13.98 | 14.42 | 13.91 |
| tieredImageNet-H | Alexnet | **15.81** | 16.59 | 16.59 | 16.2 | 16.91 | 16.72 | 16.84 | 16.89 | 16.58 |
| | Convnext Tiny | 5.57 | 5.67 | 5.67 | 5.79 | 5.93 | 5.48 | **5.46** | 5.78 | 6.64 |
| | Densenet121 | **8.59** | 9.04 | 9.04 | 8.76 | 8.87 | 8.91 | 8.9 | 8.93 | 8.93 |
| | Efficientnet_v2_s | 5.10 | 5.18 | 5.15 | 5.29 | 5.43 | 4.99 | **4.96** | 5.28 | 5.96 |
| | Inception_v3 | **7.56** | 7.74 | 7.74 | 7.71 | 8.0 | 7.69 | 7.64 | 7.87 | 8.16 |
| | Resnet18 | **10.35** | 10.79 | 10.79 | 10.6 | 10.79 | 10.65 | 10.69 | 10.86 | 10.77 |
| | Swin_v2_t | 5.77 | 5.83 | 5.83 | 5.86 | 6.04 | **5.72** | **5.72** | 5.91 | 6.32 |
| | Vgg11 | **10.24** | 10.77 | 10.77 | 10.45 | 10.73 | 10.59 | 10.75 | 10.74 | 10.87 |
| | Vit_b_16 | **6.17** | 6.25 | 6.25 | 6.26 | 6.44 | 6.23 | 6.21 | 6.29 | 6.63 |

Table D.2: Performance on **Wu-Palmer Similarity** metric of different decoding strategy for various models trained on various datasets

| | | Optimal (ours) | Deng et al. | Expected Info Valmadre | Plurality Valmadre | Karthik et al. | Jain et al. | Argmax | Top-Down | Majority |
|---|---|---|---|---|---|---|---|---|---|---|
| tieredImageNet-H | Barz & Denzler | 48.32 | 47.92 | 47.92 | 54.08 | 52.92 | 47.42 | **47.40** | 50.27 | 60.32 |
| | Cross-entropy | **18.94** | 19.55 | 19.55 | 19.69 | 20.57 | 19.29 | 19.19 | 19.49 | 21.34 |
| | Hxe ($\alpha = 0.1$) | **19.04** | 19.68 | 19.68 | 19.89 | 20.74 | 19.29 | 19.33 | 19.64 | 21.54 |
| | Hxe ($\alpha = 0.6$) | **24.27** | 26.76 | 26.76 | 25.02 | 27.09 | 25.25 | 24.80 | 25.54 | 27.43 |
| | Soft-labels | **18.81** | 19.45 | 19.45 | 19.43 | 20.29 | 19.2 | 19.13 | 19.23 | 21.20 |
| | Yolo-V2 | 21.14 | 21.72 | 21.72 | 22.17 | 22.83 | 21.18 | **20.8** | 22.07 | 23.76 |
| iNat19 | Cross-entropy | **22.71** | 23.3 | 23.3 | 23.52 | 23.4 | 23.39 | 23.53 | 23.57 | 24.92 |
| | Hxe ($\alpha = 0.1$) | **22.46** | 23.03 | 23.03 | 23.14 | 23.18 | 23.13 | 23.18 | 23.26 | 24.17 |
| | Hxe ($\alpha = 0.6$) | **29.67** | 31.59 | 31.59 | 31.02 | 31.43 | 31.34 | 31.4 | 31.6 | 32.65 |
| | Soft-labels | **23.58** | 25.96 | 25.96 | 24.05 | 23.95 | 23.79 | 23.71 | 24.05 | 27.06 |
| | Yolo-V2 | **23.51** | 24.17 | 24.17 | 24.44 | 24.39 | 24.15 | 23.99 | 24.67 | 25.41 |
| tieredImageNet-H | Alexnet | **25.5** | 26.28 | 26.28 | 26.58 | 28.38 | 26.08 | 26.04 | 26.36 | 28.99 |
| | Convnext Tiny | **8.60** | 9.04 | 9.04 | 9.42 | 9.86 | 8.70 | 8.60 | 9.03 | 11.67 |
| | Densenet121 | **13.69** | 14.23 | 14.23 | 14.09 | 14.35 | 13.89 | 13.81 | 13.97 | 15.08 |
| | Efficientnet_v2_s | 7.89 | 8.18 | 8.18 | 8.63 | 8.99 | 7.87 | **7.81** | 8.23 | 10.4 |
| | Inception_v3 | **11.88** | 12.22 | 12.22 | 12.54 | 13.23 | 12.07 | 11.97 | 12.29 | 14.04 |
| | Resnet18 | **16.61** | 17.12 | 17.10 | 17.29 | 17.72 | 16.77 | 16.74 | 17.11 | 18.49 |
| | Swin_v2_t | **9.02** | 9.30 | 9.30 | 9.64 | 10.06 | 9.06 | 9.05 | 9.32 | 11.05 |
| | Vgg11 | **16.46** | 17.07 | 17.07 | 17.11 | 17.75 | 16.63 | 16.74 | 16.86 | 18.81 |
| | Vit_b_16 | **9.69** | 9.92 | 9.92 | 10.15 | 10.54 | 9.78 | **9.69** | 9.84 | 11.36 |

Table D.3: Performance on **Zhao Similarity** metric of different decoding strategy for various models trained on various datasets

| | | Optimal (ours) | Deng et al. | Expected Info Valmadre | Plurality Valmadre | Karthik et al. | Jain et al. | Argmax | Top-Down | Majority |
|---|---|---|---|---|---|---|---|---|---|---|
| tieredImageNet-H | Barz & Denzler | **79.0** | 74.75 | 71.25 | 75.81 | 74.26 | 68.92 | 68.92 | 65.49 | 78.21 |
| | Cross-entropy | **92.32** | 90.93 | 90.54 | 90.3 | 89.68 | 89.37 | 89.29 | 89.03 | 91.6 |
| | Hxe ($\alpha = 0.1$) | **92.27** | 91.08 | 90.71 | 90.22 | 89.65 | 89.42 | 89.2 | 88.95 | 91.6 |
| | Hxe ($\alpha = 0.6$) | **91.29** | 90.38 | 90.09 | 89.13 | 87.76 | 87.43 | 87.08 | 86.42 | 90.90 |
| | Soft-labels | **92.30** | 90.99 | 90.63 | 90.43 | 89.87 | 89.39 | 89.25 | 89.15 | 91.72 |
| | Yolo-V2 | **91.35** | 90.37 | 90.04 | 88.95 | 88.43 | 88.38 | 88.39 | 87.48 | 90.56 |
| iNat19 | Cross-entropy | **91.76** | 89.97 | 89.73 | 88.88 | 88.03 | 87.99 | 87.84 | 87.94 | 90.93 |
| | Hxe ($\alpha = 0.1$) | **92.03** | 90.34 | 90.09 | 89.17 | 88.31 | 88.30 | 88.23 | 88.27 | 91.24 |
| | Hxe ($\alpha = 0.6$) | **90.67** | 88.82 | 88.63 | 86.49 | 84.94 | 84.94 | 84.85 | 84.85 | 90.24 |
| | Soft-labels | **91.40** | 89.96 | 89.82 | 88.85 | 87.82 | 87.87 | 87.84 | 87.78 | 91.22 |
| | Yolo-V2 | **91.48** | 90.09 | 89.88 | 88.42 | 87.57 | 87.70 | 87.77 | 87.38 | 90.66 |
| tieredImageNet-H | Alexnet | **89.98** | 88.00 | 87.3 | 87.23 | 86.19 | 85.48 | 85.28 | 84.99 | 89.21 |
| | Convnext Tiny | 95.63 | 95.76 | 95.67 | 95.38 | 95.11 | 95.26 | 95.24 | 94.90 | **95.82** |
| | Densenet121 | **94.14** | 93.2 | 92.97 | 92.79 | 92.51 | 92.26 | 92.23 | 92.12 | 93.56 |
| | Efficientnet_v2_s | **96.16** | 96.13 | 96.04 | 95.8 | 95.54 | 95.68 | 95.69 | 95.33 | **96.16** |
| | Inception_v3 | **94.71** | 94.19 | 93.92 | 93.77 | 93.38 | 93.32 | 93.32 | 93.04 | 94.46 |
| | Resnet18 | **93.16** | 91.94 | 91.67 | 91.4 | 90.99 | 90.76 | 90.67 | 90.42 | 92.53 |
| | Swin_v2_t | **95.92** | 95.58 | 95.45 | 95.26 | 95.00 | 95.05 | 95.03 | 94.79 | 95.77 |
| | Vgg11 | **93.15** | 92.10 | 91.83 | 91.59 | 91.09 | 90.81 | 90.60 | 90.5 | 92.61 |
| | Vit_b_16 | **95.63** | 95.21 | 95.07 | 94.92 | 94.64 | 94.60 | 94.60 | 94.46 | 95.40 |

Table D.4: Performance on $hF_{0.5}$ metric of different decoding strategy for various models trained on various datasets

| | | Optimal (ours) | Deng et al. | Expected Info Valmadre | Plurality Valmadre | Karthik et al. | Jain et al. | Argmax | Top-Down | Majority |
|---|---|---|---|---|---|---|---|---|---|---|
| tieredImageNet-H | Barz & Denzler | **73.98** | 71.28 | 70.03 | 70.01 | 71.13 | 69.04 | 69.04 | 66.29 | 68.36 |
| | Cross-entropy | **89.88** | 89.32 | 89.3 | 89.51 | 89.15 | 89.24 | 89.21 | 89.08 | 89.39 |
| | Hxe ($\alpha = 0.1$) | **89.82** | 89.32 | 89.32 | 89.43 | 89.1 | 89.29 | 89.13 | 89.01 | 89.33 |
| | Hxe ($\alpha = 0.6$) | **87.97** | 86.86 | 86.86 | 87.72 | 86.95 | 87.11 | 86.98 | 86.61 | 87.45 |
| | Soft-labels | **89.91** | 89.33 | 89.33 | 89.64 | 89.3 | 89.24 | 89.19 | 89.21 | 89.47 |
| | Yolo-V2 | **88.59** | 88.32 | 88.32 | 88.11 | 87.88 | 88.22 | 88.32 | 87.55 | 88.13 |
| iNat19 | Cross-entropy | **88.83** | 87.51 | 87.51 | 88.22 | 88.03 | 87.99 | 87.84 | 87.94 | 88.3 |
| | Hxe ($\alpha = 0.1$) | **89.18** | 87.73 | 87.73 | 88.58 | 88.31 | 88.3 | 88.23 | 88.27 | 88.79 |
| | Hxe ($\alpha = 0.6$) | **86.48** | 83.55 | 83.55 | 85.5 | 84.94 | 84.94 | 84.85 | 84.85 | 85.96 |
| | Soft-labels | **88.56** | 86.23 | 86.23 | 88.11 | 87.82 | 87.87 | 87.84 | 87.78 | 87.88 |
| | Yolo-V2 | **88.49** | 87.19 | 87.19 | 87.78 | 87.57 | 87.7 | 87.77 | 87.38 | 88.1 |
| tieredImageNet-H | Alexnet | **86.41** | 85.57 | 85.57 | 85.92 | 85.25 | 85.31 | 85.2 | 85.13 | 85.78 |
| | Convnext Tiny | 95.13 | 95.07 | 95.07 | 95.01 | 94.84 | 95.2 | **95.22** | 94.93 | 94.34 |
| | Densenet121 | **92.62** | 92.13 | 92.13 | 92.38 | 92.26 | 92.19 | 92.2 | 92.16 | 92.31 |
| | Efficientnet_v2_s | 95.57 | 95.51 | 95.53 | 95.44 | 95.29 | 95.63 | **95.66** | 95.37 | 94.93 |
| | Inception_v3 | **93.49** | 93.27 | 93.27 | 93.32 | 93.03 | 93.26 | 93.3 | 93.09 | 93.01 |
| | Resnet18 | **91.08** | 90.61 | 90.61 | 90.78 | 90.59 | 90.66 | 90.62 | 90.46 | 90.74 |
| | Swin_v2_t | **95.03** | 94.93 | 94.93 | 94.92 | 94.74 | 94.99 | 94.99 | 94.81 | 94.59 |
| | Vgg11 | **91.07** | 90.64 | 90.64 | 90.92 | 90.64 | 90.71 | 90.57 | 90.56 | 90.66 |
| | Vit_b_16 | **94.64** | 94.55 | 94.55 | 94.56 | 94.39 | 94.54 | 94.56 | 94.48 | 94.31 |

Table D.5: Performance on $hF_1$ metric of different decoding strategy for various models trained on various datasets

| | | Optimal (ours) | Deng et al. | Expected Info Valmadre | Plurality Valmadre | Karthik et al. | Jain et al. | Argmax | Top-Down | Majority |
|---|---|---|---|---|---|---|---|---|---|---|
| tieredImageNet-H | Barz & Denzler | **74.72** | 69.26 | 69.2 | 66.05 | 68.71 | 69.31 | 69.32 | 67.37 | 62.14 |
| | Cross-entropy | **90.81** | 88.31 | 88.31 | 88.96 | 88.76 | 89.2 | 89.22 | 89.2 | 87.76 |
| | Hxe ($\alpha = 0.1$) | **90.93** | 88.21 | 88.21 | 88.88 | 88.71 | 89.24 | 89.14 | 89.15 | 87.64 |
| | Hxe ($\alpha = 0.6$) | **88.67** | 84.18 | 84.18 | 86.6 | 86.32 | 86.9 | 86.97 | 86.9 | 84.71 |
| | Soft-labels | **90.92** | 88.3 | 88.3 | 89.08 | 88.9 | 89.19 | 89.2 | 89.35 | 87.81 |
| | Yolo-V2 | **89.75** | 86.94 | 86.94 | 87.5 | 87.49 | 88.15 | 88.34 | 87.71 | 86.3 |
| iNat19 | Cross-entropy | **89.52** | 85.67 | 85.67 | 87.72 | 88.03 | 87.99 | 87.84 | 87.94 | 86.35 |
| | Hxe ($\alpha = 0.1$) | **89.7** | 85.78 | 85.78 | 88.12 | 88.31 | 88.3 | 88.23 | 88.27 | 86.91 |
| | Hxe ($\alpha = 0.6$) | **86.53** | 79.32 | 79.32 | 84.67 | 84.94 | 84.94 | 84.85 | 84.85 | 82.5 |
| | Soft-labels | **89.27** | 83.22 | 83.22 | 87.52 | 87.82 | 87.87 | 87.84 | 87.78 | 85.25 |
| | Yolo-V2 | **89.15** | 84.96 | 84.96 | 87.27 | 87.57 | 87.7 | 87.77 | 87.38 | 86.12 |
| tieredImageNet-H | Alexnet | **87.5** | 84.21 | 84.21 | 84.98 | 84.56 | 85.26 | 85.24 | 85.37 | 83.27 |
| | Convnext Tiny | **95.79** | 94.58 | 94.58 | 94.75 | 94.65 | 95.18 | 95.23 | 95.01 | 93.26 |
| | Densenet121 | **93.51** | 91.45 | 91.45 | 92.1 | 92.09 | 92.19 | 92.23 | 92.25 | 91.38 |
| | Efficientnet_v2_s | **96.21** | 95.12 | 95.12 | 95.19 | 95.1 | 95.62 | 95.67 | 95.44 | 94.01 |
| | Inception_v3 | **94.22** | 92.76 | 92.76 | 93.01 | 92.79 | 93.26 | 93.33 | 93.2 | 91.98 |
| | Resnet18 | **92.08** | 89.77 | 89.77 | 90.35 | 90.31 | 90.63 | 90.64 | 90.57 | 89.43 |
| | Swin_v2_t | **95.79** | 94.5 | 94.5 | 94.69 | 94.55 | 94.97 | 95.0 | 94.88 | 93.73 |
| | Vgg11 | **92.06** | 89.7 | 89.7 | 90.44 | 90.33 | 90.68 | 90.6 | 90.7 | 89.22 |
| | Vit_b_16 | **95.37** | 94.14 | 94.14 | 94.33 | 94.22 | 94.52 | 94.57 | 94.54 | 93.52 |

Table D.6: Performance on $hF_2$ metric of different decoding strategy for various models trained on various datasets

| Metric | Group | Optimal (Brute-force) | Optimal (ours) | Deng et al. (2012) | Expected Info. | Plurality | Karthik et al. (2021) | Jain et al. (2023) | Argmax | Top-Down | Majority |
|---|---|---|---|---|---|---|---|---|---|---|---|
| Wu-Palmer | iNat19 (finetuned models) | 289.8 (× 127) | 2.286 | 0.212 | 0.211 | 0.033 | 0.033 | 0.036 | 0.018 | 0.021 | 0.004 |
| | tieredImageNet-H (finetuned models) | 125.7 (× 77) | 1.641 | 0.157 | 0.166 | 0.043 | 0.031 | 0.024 | 0.010 | 0.021 | 0.003 |
| | tieredImageNet-H (pretrained models) | 125.7 (× 84) | 1.499 | 0.159 | 0.168 | 0.046 | 0.031 | 0.024 | 0.010 | 0.022 | 0.003 |
| Zhao | iNat19 (finetuned models) | 289.8 (× 22) | 13.319 | 0.212 | 0.210 | 0.033 | 0.033 | 0.036 | 0.018 | 0.020 | 0.004 |
| | tieredImageNet-H (finetuned models) | 125.7 (× 32) | 3.908 | 0.156 | 0.165 | 0.046 | 0.032 | 0.024 | 0.010 | 0.022 | 0.003 |
| | tieredImageNet-H (pretrained models) | 125.7 (× 36) | 3.496 | 0.158 | 0.168 | 0.047 | 0.031 | 0.024 | 0.010 | 0.022 | 0.003 |
| Mistake Severity | iNat19 (finetuned models) | – | 0.011 | 0.211 | 0.214 | 0.034 | 0.034 | 0.036 | 0.018 | 0.021 | 0.004 |
| | tieredImageNet-H (finetuned models) | – | 0.016 | 0.156 | 0.158 | 0.046 | 0.032 | 0.024 | 0.010 | 0.022 | 0.003 |
| | tieredImageNet-H (pretrained models) | – | 0.016 | 0.158 | 0.160 | 0.045 | 0.032 | 0.024 | 0.010 | 0.022 | 0.003 |
| $hF_{0.5}$ | iNat19 (finetuned models) | Intractable | 0.802 | 0.210 | 0.220 | 0.035 | 0.034 | 0.036 | 0.018 | 0.021 | 0.004 |
| | tieredImageNet-H (finetuned models) | Intractable | 1.012 | 0.157 | 0.164 | 0.046 | 0.030 | 0.022 | 0.010 | 0.021 | 0.001 |
| | tieredImageNet-H (pretrained models) | Intractable | 0.963 | 0.157 | 0.164 | 0.043 | 0.031 | 0.022 | 0.009 | 0.020 | 0.001 |
| $hF_1$ | iNat19 (finetuned models) | Intractable | 1.053 | 0.213 | 0.220 | 0.032 | 0.034 | 0.036 | 0.018 | 0.020 | 0.004 |
| | tieredImageNet-H (finetuned models) | Intractable | 1.491 | 0.154 | 0.163 | 0.047 | 0.030 | 0.022 | 0.011 | 0.022 | 0.001 |
| | tieredImageNet-H (pretrained models) | Intractable | 1.387 | 0.158 | 0.163 | 0.046 | 0.031 | 0.023 | 0.008 | 0.021 | 0.002 |
| $hF_2$ | iNat19 (finetuned models) | Intractable | 1.944 | 0.213 | 0.233 | 0.032 | 0.034 | 0.036 | 0.018 | 0.020 | 0.004 |
| | tieredImageNet-H (finetuned models) | Intractable | 2.921 | 0.156 | 0.168 | 0.044 | 0.031 | 0.024 | 0.010 | 0.021 | 0.003 |
| | tieredImageNet-H (pretrained models) | Intractable | 2.881 | 0.159 | 0.170 | 0.050 | 0.032 | 0.024 | 0.010 | 0.025 | 0.003 |

Table D.7: Decoding time for each strategy, reported in milliseconds per sample and averaged across all models within each group.

### D.4. Additional Information for the Blurring Motivation

In Section 5.1, we provide the motivation for introducing image blurring. We illustrate this motivation with a single example based on Figure 3, which visualizes the agreement map for $hF_1$-score and Majority decoding strategies. An agreement map is constructed as follows:

- A simplex mesh in $\mathbb{R}^3$ is created, where each point corresponds to a single probability distribution over leaf nodes.
- For each point in the mesh, the probability is decoded using both the optimal decoding strategy and the heuristic decoding strategy.
- The agreement map is constructed by coloring areas in green where the two decoding strategies agree, and in red where they disagree.

In the following, we display agreement maps for all heuristic strategies selected in the main text versus the optimal strategy, evaluated across all six metrics.

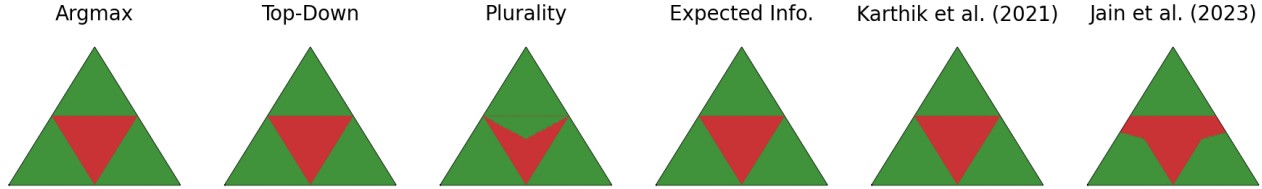

Figure D.1: Agreements maps of heuristic decoding strategies vs. **Mistake Severity** optimal decoding strategy.

Overall, it is evident that the more central the probability is within the simplex, the greater the disagreement between the optimal decoding strategy and the heuristic strategy. This observation motivates the introduction of the blurring effect. Specifically, blurring an image increases label uncertainty and randomness, causing the oracle probability distribution to shift towards the center of the simplex. Similarly, the estimated oracle probability distribution follows this shift.

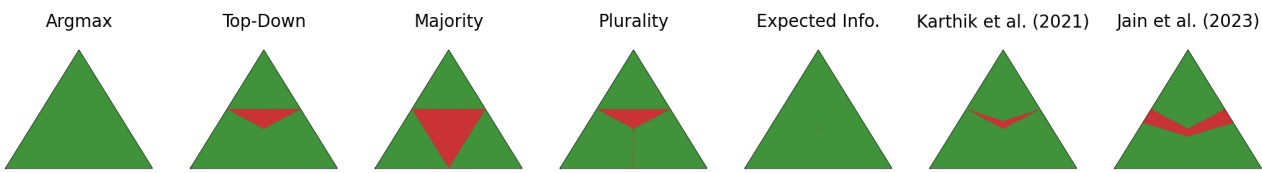

Figure D.2: Agreements maps of heuristic decoding strategies vs. **Wu-Palmer similarity** optimal decoding strategy.

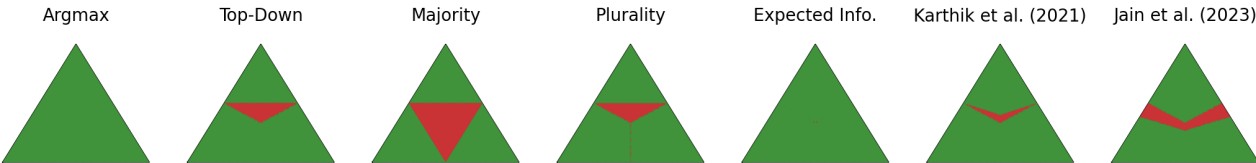

Figure D.3: Agreements maps of heuristic decoding strategies vs. **Zhao similarity** optimal decoding strategy.

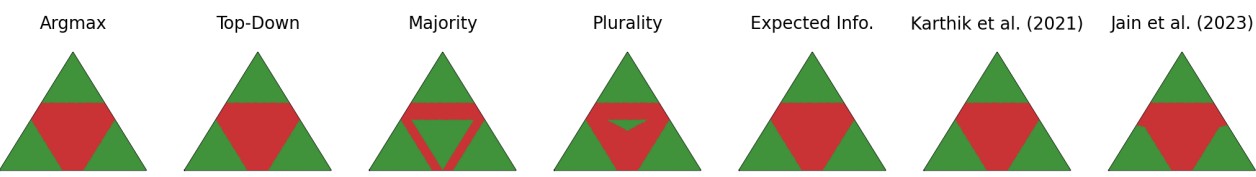

Figure D.4: Agreements maps of heuristic decoding strategies vs. $\mathrm{hF}_{0.5}$ optimal decoding strategy.

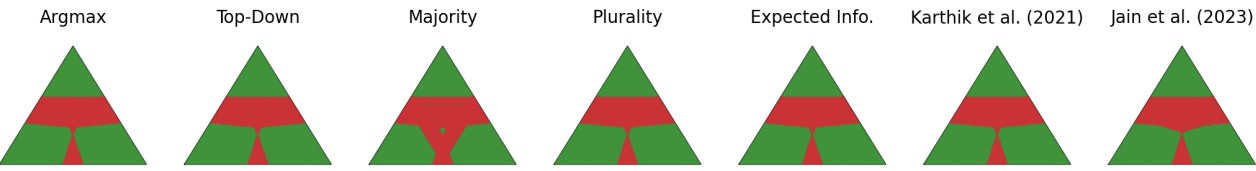

Figure D.5: Agreements maps of heuristic decoding strategies vs. $\mathrm{hF}_1$ optimal decoding strategy.

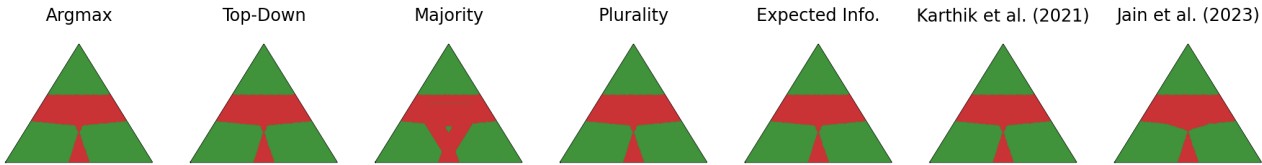

Figure D.6: Agreements maps of heuristic decoding strategies vs. $\mathrm{hF}_2$ optimal decoding strategy.

## D.5. Additional Experiments for the blurring experiment

We begin by describing the experimental setup used for these evaluations. The models from previous experiments of Appendix D.2, trained on various datasets, remain unchanged. However, we modify the test set images by applying increasing levels of blur, parameterized by the standard deviation $\sigma$. This blur is applied to resized images, typically of size $224 \times 224$, using a Gaussian Blur transformation with a kernel size of $61$ and a standard deviation of $\sigma$.

We recall also the intuition of this blurring experiment. As the images become progressively more blurred, their features become less informative, causing the posterior probability distribution to converge toward the center of the simplex. As illustrated in Figure 1, even a human expert would find it challenging to confidently predict the true class label for the image on the far right. This experimental setup is designed to amplify the frequency of disagreements between heuristic strategies and optimal decoding.

### D.5.1. MORE VISUAL EXAMPLES

| | Blur level σ=0 | Blur level σ=3 | Blur level σ=6 | Blur level σ=9 |
|---|---|---|---|---|

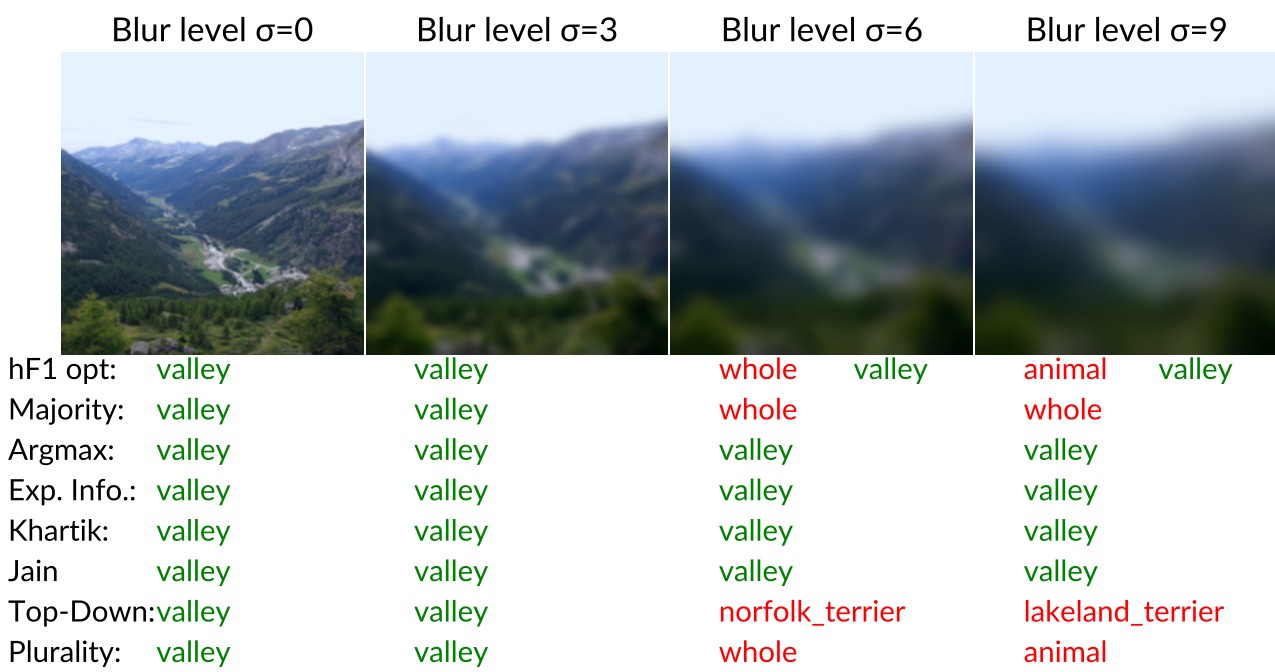

| | | | | |
|---|---|---|---|---|
| hF1 opt: | valley | valley | whole valley | animal valley |
| Majority: | valley | valley | whole | whole |
| Argmax: | valley | valley | valley | valley |
| Exp. Info.: | valley | valley | valley | valley |
| Khartik: | valley | valley | valley | valley |
| Jain | valley | valley | valley | valley |
| Top-Down: | valley | valley | norfolk_terrier | lakeland_terrier |
| Plurality: | valley | valley | whole | animal |

Figure D.7: Influence of blurring to the decision-making of various decoding strategies. Image displayed is labeled *valley*

| | Blur level σ=0 | Blur level σ=2 | Blur level σ=4 | Blur level σ=7 |
|---|---|---|---|---|

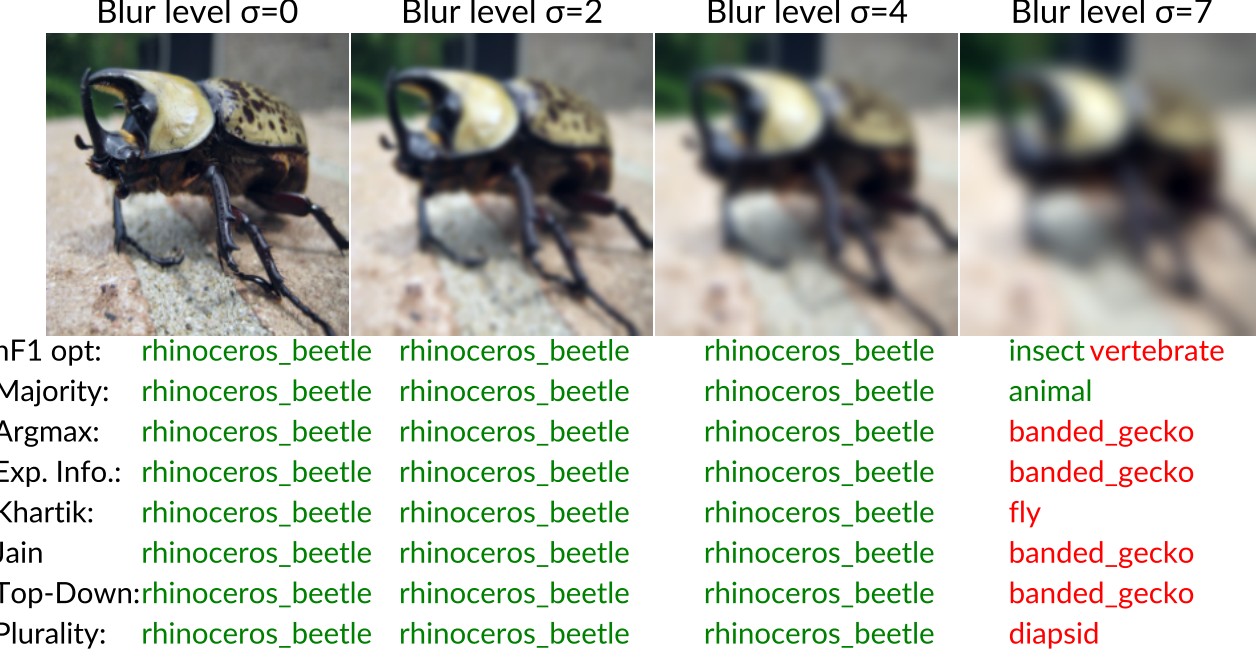

| | | | | |
|---|---|---|---|---|
| hF1 opt: | rhinoceros_beetle | rhinoceros_beetle | rhinoceros_beetle | insect vertebrate |
| Majority: | rhinoceros_beetle | rhinoceros_beetle | rhinoceros_beetle | animal |
| Argmax: | rhinoceros_beetle | rhinoceros_beetle | rhinoceros_beetle | banded_gecko |
| Exp. Info.: | rhinoceros_beetle | rhinoceros_beetle | rhinoceros_beetle | banded_gecko |
| Khartik: | rhinoceros_beetle | rhinoceros_beetle | rhinoceros_beetle | fly |
| Jain | rhinoceros_beetle | rhinoceros_beetle | rhinoceros_beetle | banded_gecko |
| Top-Down: | rhinoceros_beetle | rhinoceros_beetle | rhinoceros_beetle | banded_gecko |
| Plurality: | rhinoceros_beetle | rhinoceros_beetle | rhinoceros_beetle | diapsid |

Figure D.8: Influence of blurring to the decision-making of various decoding strategies. Image displayed is labeled *rhinoceros beetle*

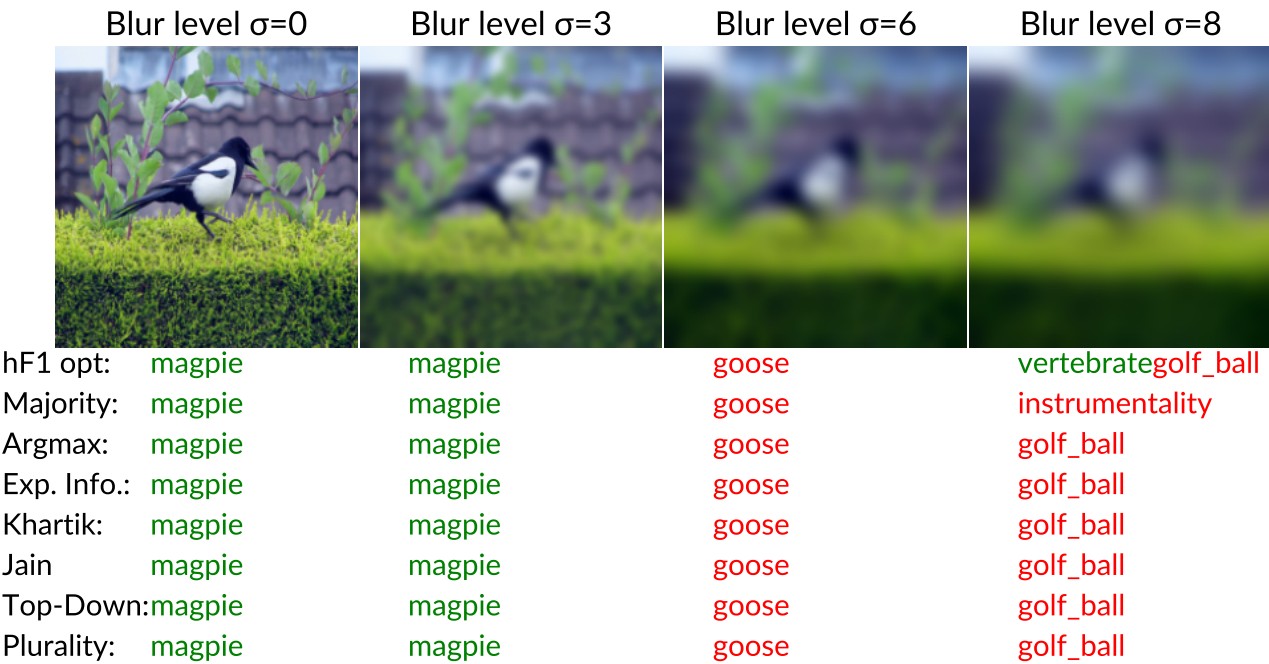

| | Blur level σ=0 | Blur level σ=3 | Blur level σ=6 | Blur level σ=8 |
|---|---|---|---|---|
| hF1 opt: | magpie | magpie | goose | vertebrate golf_ball |
| Majority: | magpie | magpie | goose | instrumentality |
| Argmax: | magpie | magpie | goose | golf_ball |
| Exp. Info.: | magpie | magpie | goose | golf_ball |
| Khartik: | magpie | magpie | goose | golf_ball |
| Jain | magpie | magpie | goose | golf_ball |
| Top-Down: | magpie | magpie | goose | golf_ball |
| Plurality: | magpie | magpie | goose | golf_ball |

Figure D.9: Influence of blurring to the decision-making of various decoding strategies. Image displayed is labeled *magpie*

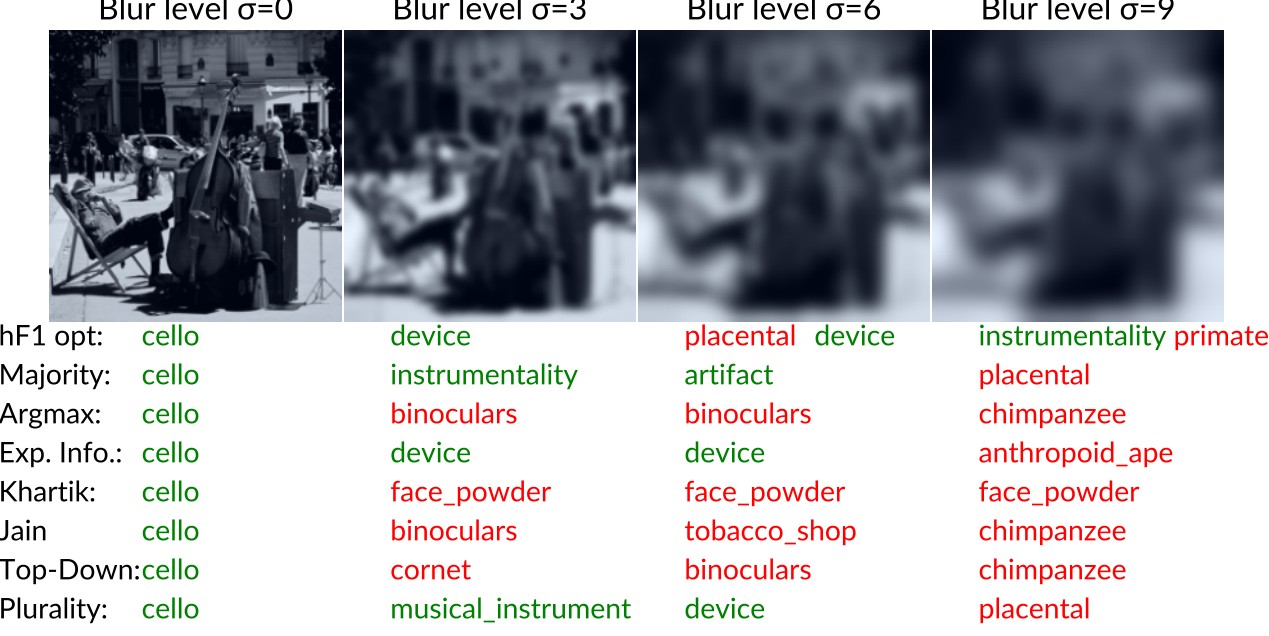

| | Blur level σ=0 | Blur level σ=3 | Blur level σ=6 | Blur level σ=9 |
|---|---|---|---|---|
| hF1 opt: | cello | device | placental device | instrumentality primate |
| Majority: | cello | instrumentality | artifact | placental |
| Argmax: | cello | binoculars | binoculars | chimpanzee |
| Exp. Info.: | cello | device | device | anthropoid_ape |
| Khartik: | cello | face_powder | face_powder | face_powder |
| Jain | cello | binoculars | tobacco_shop | chimpanzee |
| Top-Down: | cello | cornet | binoculars | chimpanzee |
| Plurality: | cello | musical_instrument | device | placental |

Figure D.10: Influence of blurring to the decision-making of various decoding strategies. Image displayed is labeled *cello*

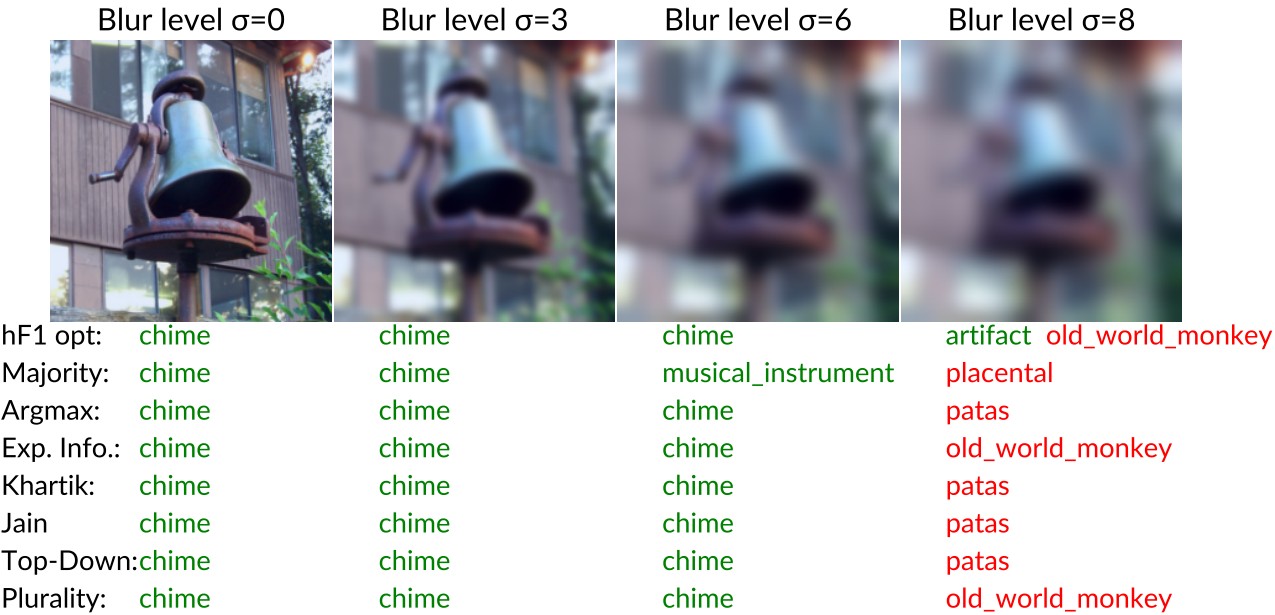

Figure D.11: Influence of blurring to the decision-making of various decoding strategies. Image displayed is labeled *chime*

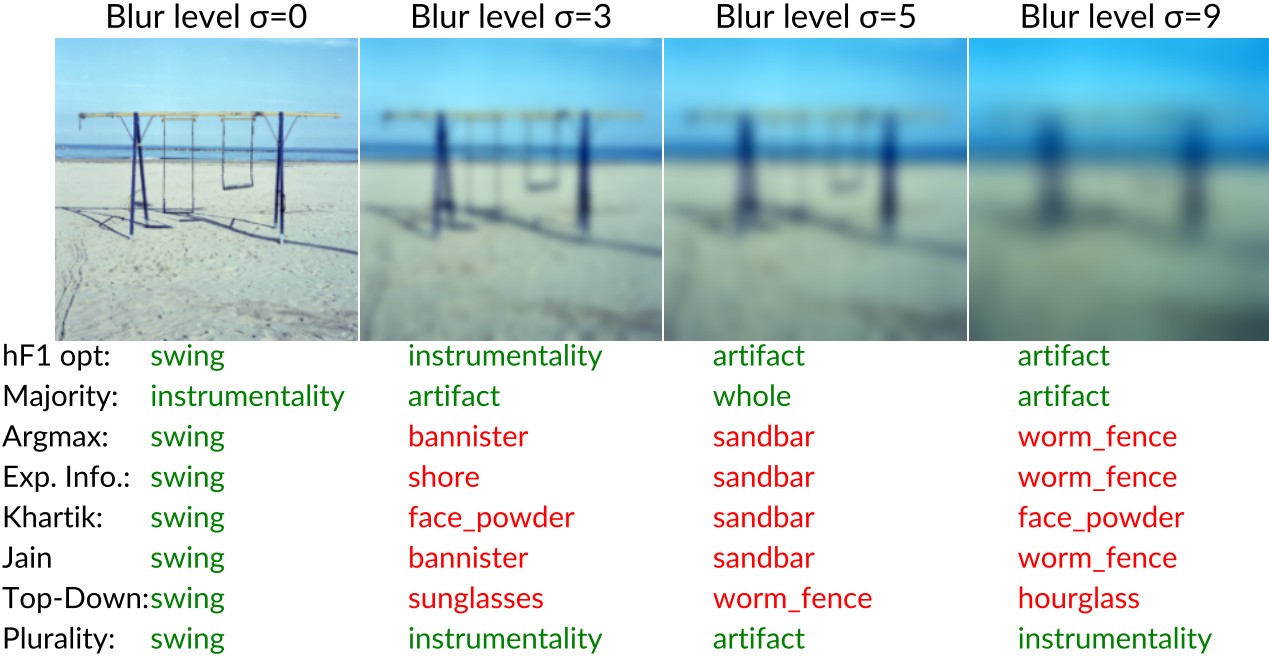

Figure D.12: Influence of blurring to the decision-making of various decoding strategies. Image displayed is labeled *swing*

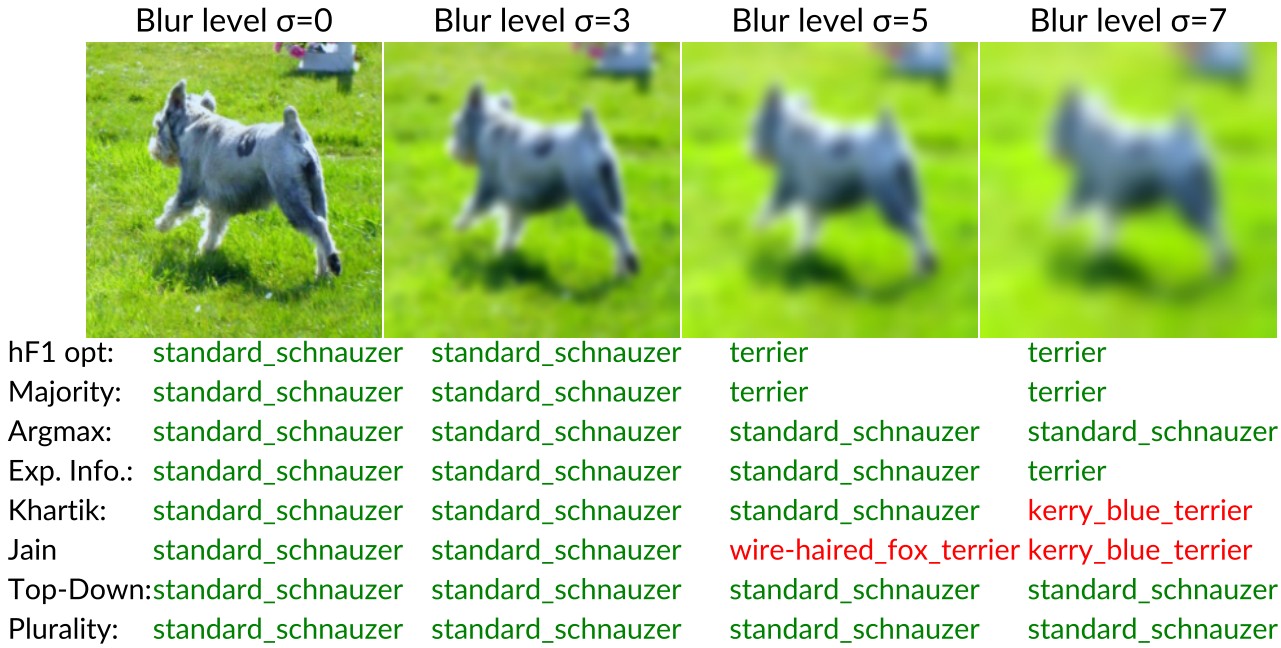

| | Blur level σ=0 | Blur level σ=3 | Blur level σ=5 | Blur level σ=7 |
|---|---|---|---|---|
| hF1 opt: | standard_schnauzer | standard_schnauzer | terrier | terrier |
| Majority: | standard_schnauzer | standard_schnauzer | terrier | terrier |
| Argmax: | standard_schnauzer | standard_schnauzer | standard_schnauzer | standard_schnauzer |
| Exp. Info.: | standard_schnauzer | standard_schnauzer | standard_schnauzer | terrier |
| Khartik: | standard_schnauzer | standard_schnauzer | standard_schnauzer | kerry_blue_terrier |
| Jain | standard_schnauzer | standard_schnauzer | wire-haired_fox_terrier | kerry_blue_terrier |
| Top-Down: | standard_schnauzer | standard_schnauzer | standard_schnauzer | standard_schnauzer |
| Plurality: | standard_schnauzer | standard_schnauzer | standard_schnauzer | standard_schnauzer |

Figure D.13: Influence of blurring to the decision-making of various decoding strategies. Image displayed is labeled *standard schnauzer*

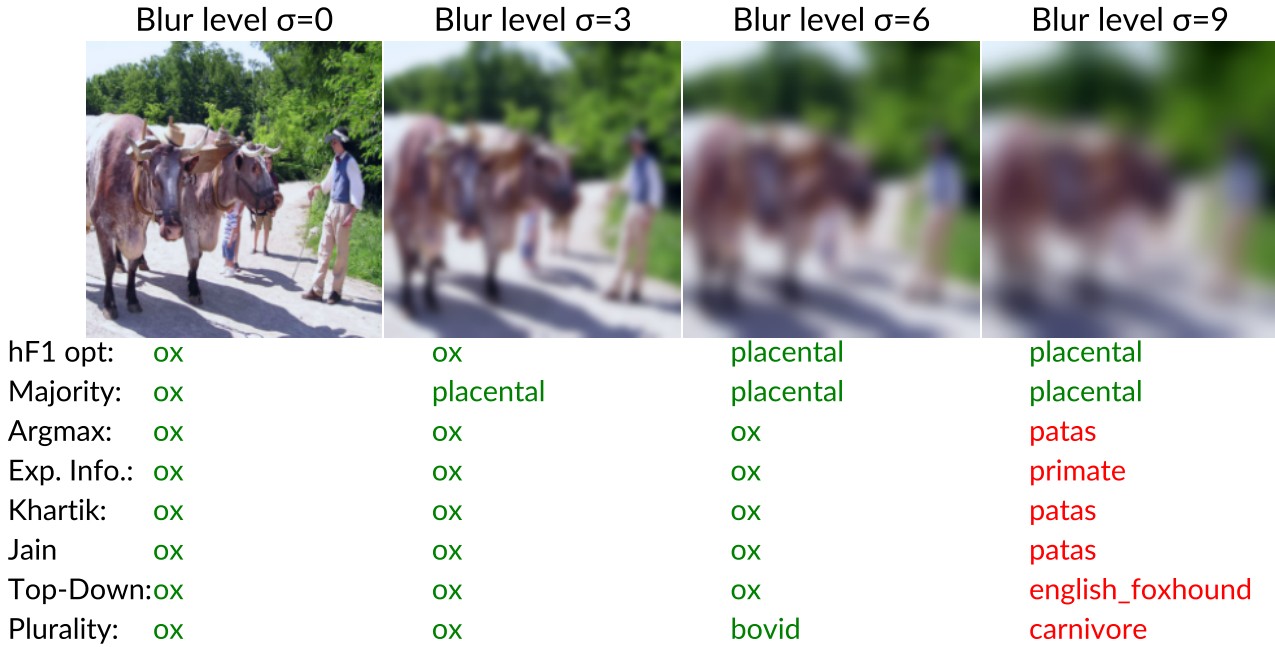

| | Blur level σ=0 | Blur level σ=3 | Blur level σ=6 | Blur level σ=9 |
|---|---|---|---|---|
| hF1 opt: | ox | ox | placental | placental |
| Majority: | ox | placental | placental | placental |
| Argmax: | ox | ox | ox | patas |
| Exp. Info.: | ox | ox | ox | primate |
| Khartik: | ox | ox | ox | patas |
| Jain | ox | ox | ox | patas |
| Top-Down: | ox | ox | ox | english_foxhound |
| Plurality: | ox | ox | bovid | carnivore |

Figure D.14: Influence of blurring to the decision-making of various decoding strategies. Image displayed is labeled *ox*

### D.5.2. FULL QUANTITATIVE RESULTS

Figure D.15 illustrates the relative gain in performance compared to optimal decoding strategies for the six selected metrics, averaged across datasets and models. For the $hF_\beta$ scores and the **Mistake Severity** metric, we observe the expected trend: as image blurriness increases, the relative loss in performance of heuristic strategies becomes more pronounced. This aligns with our intuition that as the problem becomes more uncertain, the choice of an appropriate decoding strategy becomes increasingly important for optimizing a hierarchical target metric.

Interestingly, the results differ for the **Zhao Similarity** and **Wu-Palmer** metrics. While the optimal decoding strategy consistently remains the best, the relative performance of heuristics exhibits an unexpected trend. Specifically, the relative loss decreases significantly as the blur level increases up to $\sigma = 3$, after which it begins to increase again. Despite this variability, the optimal decoding strategy maintains a statistically significant advantage over the heuristics in all cases.

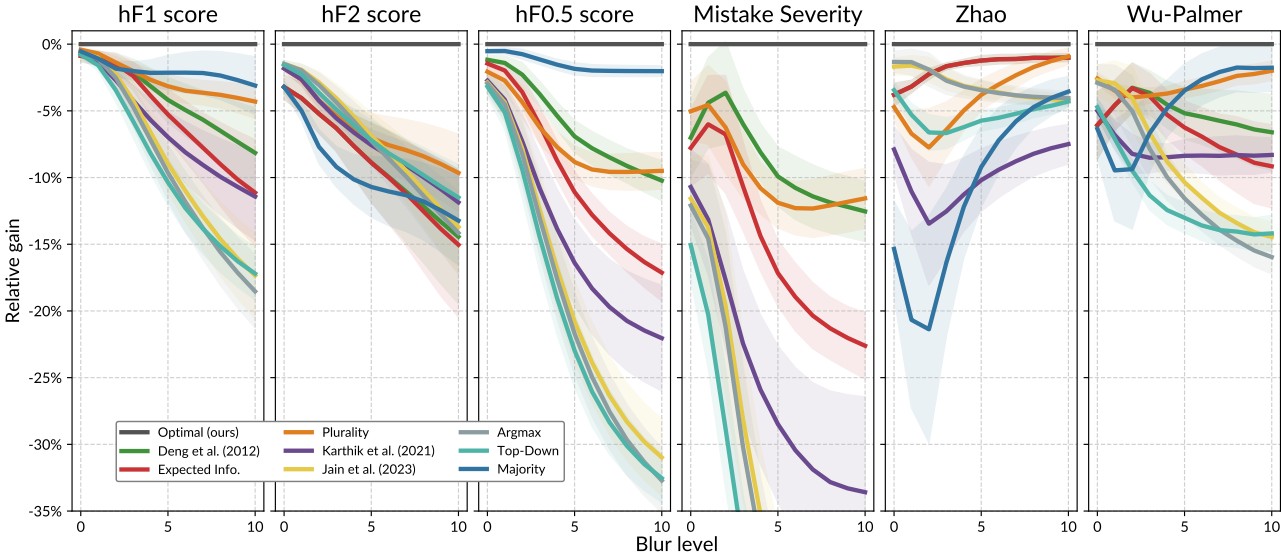

Figure D.15: **Impact of blurring on the sub-optimality of heuristic decodings.** This figure shows the relative performance gain (in %) of heuristic decodings compared to the optimal decoding for the 6 selected metrics, as a function of the blur level. Results are averaged across multiple models and datasets, with a 95% confidence interval displayed for each decoding strategy.

# E. Proofs.

This section provides the detailed proofs of the theoretical results.

## E.1. Nodes decoding ($\mathcal{H} = \mathcal{N}$)

In this section we will rely on the conditional risk defined as follows:

**Definition E.1.** Let $p \in \Delta(\mathcal{L})$ and $h \in \mathcal{N}$ then we define the conditional risk of $h$ for metric $C$ as follows:

$$\mathcal{R}_C(h|p) = \sum_{l \in \mathcal{L}} p(l) \cdot C(h, l)$$

E.1.1. HIERARCHICALLY REASONABLE METRICS : CASE IN WHICH EQUATIONS (2) AND (3) ARE VERIFIED.

**Lemma E.2.** Let $C : \mathcal{N} \times \mathcal{L} \to \mathbb{R}$ be hierarchically reasonable. For $n \in \mathcal{N} \setminus \{\mathbf{r}\}$, define $\delta_{nl}^C = C(n, l) - C(\pi(n), l)$ the node-parent loss difference for label $l$ and:

$$\underline{M}_n = \max_{l \in \mathcal{L} \setminus \mathcal{L}(n)} \delta_{nl} > 0 \quad m_n = -\max_{l \in \mathcal{L}(n)} \delta_{nl} > 0 \quad M_n = -\min_{l \in \mathcal{L}(n)} \delta_{nl} > 0 \quad \underline{m}_n = \min_{l \in \mathcal{L} \setminus \mathcal{L}(n)} \delta_{nl} > 0$$

*The, for $p \in \Delta(\mathcal{L})$, we have:*
1. $p(n) > \frac{\underline{M}_n}{\underline{M}_n + m_n} := q_{\max}^C(n) \implies \xi_C^*(p) \neq \pi(n)$
2. $p(n) < \frac{\underline{m}_n}{\underline{m}_n + M_n} := q_{\min}^C(n) \implies \xi_C^*(p) \neq n$

*Proof.*
Let $L : \mathcal{N} \times \mathcal{L} \to \mathbb{R}$ satisfy Equations (2) and (3). We now proceed to prove statement 1.

Assume $p(n) > \frac{\underline{M}_n}{\underline{M}_n + m_n}$. Then, we have:

$$p(n) > \frac{\underline{M}_n}{\underline{M}_n + m_n}$$

$$\iff m_n \underbrace{p(n)}_{= \sum_{l \in \mathcal{L}(n)} p(l)} > \underline{M}_n \underbrace{(1 - p(n))}_{= \sum_{l \in \mathcal{L} \setminus \mathcal{L}(n)} p(l)}$$

This implies

$$\sum_{l \in \mathcal{L}(n)} p(l) \cdot \underbrace{m_n}_{\leq C(\pi(n),l) - C(n,l)} > \sum_{l \in \mathcal{L} \setminus \mathcal{L}(n)} p(l) \cdot \underbrace{\underline{M}_n}_{\geq C(n,l) - C(\pi(n),l)}$$

Rewriting:

$$\sum_{l \in \mathcal{L}(n)} p(l) C(\pi(n), l) - \sum_{l \in \mathcal{L}(n)} p(l) C(n, l) > \sum_{l \in \mathcal{L} \setminus \mathcal{L}(n)} p(l) C(n, l) - \sum_{l \in \mathcal{L} \setminus \mathcal{L}(n)} p(l) C(\pi(n), l)$$

Thus:

$$\sum_{l \in \mathcal{L}} p(l) C(\pi(n), l) > \sum_{l \in \mathcal{L}} p(l) C(n, l)$$

And finally:

$$\mathcal{R}_C(\pi(n)|p) > \mathcal{R}_C(n|p)$$

Therefore, $n$ has a strictly lower conditional risk than $\pi(n)$, which shows that $\pi(n)$ is not the optimal decoding.

Statement 2 follows this exact same proof.

Let us also prove that

$$\frac{m_n}{\underline{m}_n + M_n} := q^C_{\min}(n) \leq q^C_{\max}(n) = \frac{M_n}{\underline{M}_n + m_n}$$

The function $x \mapsto \frac{x}{x+C}$ where $C$ is a constant is an increasing function on $\mathbb{R}^+$ and as $\underline{m}_n \leq \underline{M}_n$ we have,

$$q^C_{\min}(n) \leq \frac{M_n}{\underline{M}_n + \underbrace{\underline{M}_n}_{\geq m_n}} \leq \frac{M_n}{\underline{M}_n + m_n} = q^C_{\max}(n)$$

**Theorem E.3.** *Let $L$ be a hierarchically reasonable metric and $p \in \Delta(\mathcal{L})$, then the optimal decision rule $\xi^*_C(p)$ can be computed with an algorithm of $\mathcal{O}(d_{max} \cdot |\mathcal{L}| + |\mathcal{N}|)$ time complexity.*

---

**Algorithm 2** $q^C_{\min}(n)$ and $q^C_{\max}(n)$ computation.

[tb]

**function** GETCST($\mathcal{L}_n$, $\mathcal{L}$)
    $\underline{M}_n \leftarrow \max\limits_{l \in \mathcal{L} \setminus \mathcal{L}_n} C(n,l) - C(\pi(n),l) > 0$
    $m_n \leftarrow \min\limits_{l \in \mathcal{L}_n} C(\pi(n),l) - C(n,l) > 0$
    $M_n \leftarrow \max\limits_{l \in \mathcal{L}_n} C(\pi(n),l) - C(n,l) > 0$
    $\underline{m}_n \leftarrow \min\limits_{l \in \mathcal{L} \setminus \mathcal{L}_n} C(n,l) - C(\pi(n),l) > 0$
    **return** $\frac{m_n}{\underline{m}_n + M_n}$, $\frac{M_n}{\underline{M}_n + m_n}$
**end function**
**function** QCOMP($n$, $q^C_{\min}$, $q^C_{\max}$, $\mathcal{L}$)
    **if** $n \in \mathcal{L}$ **then**
        $q_1, q_2 \leftarrow$ GETCST($\{n\}$, $\mathcal{L}$)
        $q^C_{\min}(n) \leftarrow q_1$
        $q^C_{\max}(n) \leftarrow q_2$
        **return** $\{n\}$
    **else**
        $\mathcal{L}(n) \leftarrow \bigcup\limits_{c \in \mathcal{C}(n)}$ QCOMP($c$, $q^C_{\min}$, $q^C_{\max}$, $\mathcal{L}$)
        $q_1, q_2 \leftarrow$ GETCST($\mathcal{L}(n)$, $\mathcal{L}$)
        $q^C_{\min}(n) \leftarrow q_1$
        $q^C_{\max}(n) \leftarrow q_2$
        **return** $\mathcal{L}(n)$
    **end if**
**end function**
$q^C_{\min}, q^C_{\max} \leftarrow 0_{|\mathcal{N}|}, 0_{|\mathcal{N}|}$
PROBANODES($\mathbf{r}$, $q^C_{\min}$, $q^C_{\max}$, $\mathcal{L}$)

---

---

**Algorithm 3** Probability Computation

---

1: **function** PROBANODESREC($n, p, p_{\text{leaves}}$)
2:     **if** $n \in \mathcal{L}$ **then**
3:         $p(n) \leftarrow p_{\text{leaves}}(n)$
4:         **return** $p(n)$
5:     **else**
6:         $p(n) \leftarrow \sum\limits_{c \in \mathcal{C}(n)}$ PROBANODESREC($c, p, p_{\text{leaves}}$)
7:     **end if**
8: **end function**
9: **function** PROBANODES($p_{\text{leaves}}$)
10:     $p \leftarrow 0_{|\mathcal{N}|}$
11:     PROBANODESREC($\mathbf{r}, p, p_{\text{leaves}}$)
12:     **return** $p$
13: **end function**

---

**Algorithm 4** Candidate Selection

---

1: **function** FINDCANDSETREC($n, p, S, q_{\max}, q_{\min}$)
2:     **if** $p(n) > q_{\max}(n)$ **then**
3:         $S \leftarrow S \setminus \{\pi(n)\}$
4:     **else if** $p(n) < q_{\min}(n)$ **then**
5:         $S \leftarrow S \setminus \{n\}$
6:     **else if** $n \in \mathcal{L}$ or $p(n) < \min\limits_{n \in \mathcal{N}} q_{\min}(n)$ **then**
7:         **return**
8:     **end if**
9:     **for** $c \in \mathcal{C}(n)$ **do**
10:         FINDCANDSETREC($c, p, S, q_{\max}, q_{\min}$)
11:     **end for**
12: **end function**
13: **function** FINDCANDSET($p, q_{\max}, q_{\min}$)
14:     $S \leftarrow \mathcal{N}$
15:     FINDCANDSETREC($\mathbf{r}, p, S, q_{\max}, q_{\min}$)
16:     **return** $S$
17: **end function**

---

**Algorithm 5** Brute-force search

---

1: **function** BRUTEFORCE($S, p$)
2:     $n_{\text{opt}} \leftarrow \emptyset$
3:     $r_{\text{opt}} \leftarrow \infty$
4:     **for** $n \in S$ **do**
5:         $r \leftarrow \sum\limits_{l \in \mathcal{L}} p(l)C(n, l)$
6:         **if** $r < r_{\text{opt}}$ **then**
7:             $r_{\text{opt}} \leftarrow r$
8:             $n_{\text{opt}} \leftarrow n$
9:         **end if**
10:     **end for**
11:     **return** $n_{\text{opt}}$
12: **end function**

---

---

**Algorithm 6** General Algorithm

---

1: **function** FINDOPTIMAL($p_{\text{leaves}}, q_{\max}, q_{\min}$)
2:     $p \leftarrow$ PROBANODES($p_{\text{leaves}}$)
3:     $S \leftarrow$ FINDCANDSET($p, q_{\max}, q_{\min}$)
4:     $n_{\text{opt}} \leftarrow$ BRUTEFORCE($S, p$)
5:     **return** $n_{\text{opt}}$
6: **end function**

---

*Proof.* The general algorithm is summarized in Algorithm 6. It consists in

1. Computing probability distribution over nodes, given probability distribution over leaves. (Algorithm 3)
2. Filtering each node based on condition of Lemma E.2. It outputs a candidate set $S$ which can all be the optimal prediction (Algorithm 4)
3. A brute-force algorithm is performed on this remaining candidate set (Algorithm 5)

- Algorithm 2 correspond to the computation of $(q_{\min}^C(n))_{n \in \mathcal{N}}$ and $(q_{\max}^C(n))_{n \in \mathcal{N}}$. This is independant of the input probability distribution $p$ and can be performed beforehand. To this extent, the time complexity of Algorithm 1 is not added to the overall complexity of the optimal decoding. However this algorithm has a $\mathcal{O}(|\mathcal{N}| \cdot |\mathcal{L}|)$ complexity. In fact, function QCOMP consists in a tree traversal in which each node is visited exactly once. For each visit, constants $\underline{M}_n, m_n, M_n, \underline{m}_n$ are computed in a $\mathcal{O}(|\mathcal{L}|)$ time complexity. Therefore, the overall complexity of Algorithm 2 is $\mathcal{O}(|\mathcal{N}| \cdot |\mathcal{L}|)$.
- Algorithm 3 correspond to the computation the whole probability distribution over all nodes given probability distribution over leaf nodes. This seemingly trivial computation is a tree traversal which necessitates the visit of exactly each node. Therefore the overall complexity of Algorithm 3 is $\mathcal{O}(|\mathcal{N}|)$. This has to be done for each probability distribution, and therefore contributes to Theorem E.3 time complexity.
- Algorithm 7 correspond to the candidate selection step. It performs once again a tree traversal, which also necessitates to visit each node exactly once. Therefore the overall complexity of Algorithm 3 is $\mathcal{O}(|\mathcal{N}|)$. For convenience, we separate Algorithm 3 and Algorithm 4, but this can be done simultaneously.
- Algorithm 5 is a basic brute-force algorithm. For each node $n \in S$, the computation of the risk $r$ necessite $|\mathcal{L}|$ operation. Therefore the overall complexity of Algorithm 4 is $\mathcal{O}(|S| \cdot |\mathcal{L}|)$. Let us now prove that $|S| = \mathcal{O}(d_{\max})$.

*Proof of $|S| = \mathcal{O}(d_{max})$.*

We recall condition 2 of Theorem E.3:

$$p(n) < q_{\min}^C(n) \implies \xi_C^*(p) \neq n$$

Then, the contrapositive writes :

$$n \in S \implies p(n) \geq q_{\min}^C(n) \geq \min_{n \in \mathcal{N}} q_{\min}^C(n) := q_{\min}^C$$

Therefore,

$$\sum_{n \in S} \underbrace{p(n)}_{\geq q_{\min}^C} \geq |S| \cdot q_{\min}^C$$

Inequality holds also for $n = \mathbf{r}$ because $p(\mathbf{r}) = 1 \geq q_{\min}^C$. This can be rewritten:

$$|S| \leq \frac{1}{q_{\min}^C} \cdot \underbrace{\sum_{n \in S} p(n)}_{\leq \sum_{n \in \mathcal{N}} p(n)}$$

We recall. $p(n) = \sum_{n \in \mathcal{L}(n)} p(l)$. Then, on a given level of the hierarchy *i.e* on a set $\{n \in \mathcal{N}, d(n) = d\}$, there is no common leaf descendant between two elements of the set. We therefore have

$$\sum_{\substack{n \in \mathcal{N} \\ d(n) = d}} p(n) \leq p(l) = 1$$
$$\phantom{\sum_{\substack{n \in \mathcal{N} \\ d(n) = d}} p(n) \leq} \,{}_{l \in \mathcal{L}}$$

Eventually,

$$\sum_{n \in \mathcal{N}} p(n) = \sum_{d=0}^{d_{\max}} \underbrace{\sum_{\substack{n \in \mathcal{N} \\ d(n)=d}} p(n)}_{\leq 1} \leq d_{\max} + 1 \tag{7}$$

Which finally give,

$$|S| \leq \frac{1}{q_{\min}^C} \cdot (d_{\max} + 1) = \mathcal{O}(d_{\max})$$

Wrapping up everything we obtain a time Complexity for Algorithm 6 of $\mathcal{O}(d_{\max} \cdot |\mathcal{L}| + |\mathcal{N}|)$

### E.1.2. APPLICATION TO SPECIFIC METRICS

In this section, we detailed the proof of statements made in Section 4.2.1. We aim to retrieve the two following propositions.

**Proposition E.4.** *Tree distance Loss Optimal Decision Rule (Ramaswamy et al., 2015). Let* $\mathrm{DL}$ *be the **Tree distance Loss** whose definition is* $\mathrm{DL}(h, y) = $ *Length of the shortest path between $h$ and $y$ in $\mathcal{T}$. Let $p \in \Delta(\mathcal{L})$ then optimal decision rule $\xi_{\mathrm{DL}}(p)$ is given by :*

$$\xi_{\mathrm{DL}}(p) = \operatorname*{argmax}_{n \in \mathcal{N}} d(n) \ \ s.t \ \ p(n) \geq 0.5$$

**Proposition E.5.** *Generalized Tree distance Loss Optimal Decision Rule (Cao et al., 2024). Let* $\mathrm{DL}_c$ *be the **Generalized Tree distance Loss** whose definition is* $\mathrm{DL}_c(h, y) = \mathrm{DL}(h, y) + c \cdot d(h)$. *Where $d(h)$ is the depth of node $h$. Let $p \in \Delta(\mathcal{L})$ then optimal decision rule $\xi_{\mathrm{DL}}(p)$ is given by :*

$$\xi_{\mathrm{DL}_c}(p) = \operatorname*{argmax}_{n \in \mathcal{N}} d(n) \ \ s.t \ \ p(n) \geq \frac{1+c}{2}$$

*Proof.*

We prove the second Proposition, as the first one is included in the second. Let $n \in \mathcal{N} \backslash \{r\}$ and $l \in \mathcal{L} \backslash \mathcal{L}(n)$ then :

$$\begin{aligned}
\delta_{nl}^{\mathrm{DL}_c} &= \mathrm{DL}_c(n, l) - \mathrm{DL}_c(\pi(n), l) \\
&= \mathrm{DL}(n, l) + c \cdot d(n) - (\mathrm{DL}(\pi(n), l) + c \cdot (d(n) - 1)) \\
&= \underbrace{\mathrm{DL}(n, l) - \mathrm{DL}(\pi(n), l)}_{=1 \, (\text{because } l \in \mathcal{L} \backslash \mathcal{L}(n))} + c(d(n) - d(n) + 1) \\
&= 1 + c
\end{aligned}$$

Then,

$$\underline{M}_n = \max_{l \in \mathcal{L} \backslash \mathcal{L}(n)} \delta_{nl}^{\mathrm{DL}_c} = 1 + c$$

Similarly,

$$m_n = 1 - c \ \ M_n = 1 - c \ \ \underline{m}_n = 1 + c$$

Therefore, $q_{\max}^{\mathrm{DL}_c} = q_{\max}^{\mathrm{DL}_c} = \frac{1+c}{2}$. Lemma E.2 gives:

- if $p(n) > \frac{1+c}{2}$ then $\pi(n)$ is not the optimal prediction
- if $p(n) < \frac{1+c}{2}$ then $n$ is not the optimal prediction

Then, as there can be at most node per level whose probability is greater than $\frac{1+c}{2} \geq \frac{1}{2}$ we retrieve that $\xi_{\mathrm{DL}_c}(p)$ is the deepest node whose probability is greater than $\frac{1+c}{2}$. This concludes the proof.

E.1.3. HIERARCHICALLY REASONABLE METRICS : CASE IN WHICH EQUATIONS (3) AND (4) ARE VERIFIED.

In such case, we have the following result:

**Proposition E.6.** *Let $C : \mathcal{N} \times \mathcal{L} \to \mathbb{R}$ be a metric that satisfy Equations (3) and (4). For $n \in \mathcal{N} \setminus \{\mathbf{r}\}$, define $\delta_{nl}^C = C(n, l) - C(\pi(n), l)$ the node-parent loss difference for label $l$ and*

- $\underline{\tilde{M}}_n = \max\limits_{\substack{l \in \mathcal{L} \setminus \mathcal{L}(n) \\ \mathrm{LCA}(n,l) \neq \mathbf{r}}} C(n, l) - C(\pi(n), l) > 0 \qquad \tilde{m}_n = \min\limits_{l \in \mathcal{L}(n)} C(\pi(n), l) - C(n, l) > 0$

- $\tilde{M}_n = \max\limits_{l \in \mathcal{L}(n)} C(\pi(n), l) - C(n, l) > 0 \qquad \underline{\tilde{m}}_n = \min\limits_{\substack{l \in \mathcal{L} \setminus \mathcal{L}(n) \\ \mathrm{LCA}(n,l) \neq \mathbf{r}}} C(n, l) - C(\pi(n), l) > 0$

*Let $a_n$ be the shallowest non-root ancestor of $n$ ($a_n \in \mathcal{C}(\mathbf{r})$) the set of children of $\mathbf{r}$. For $p \in \Delta(\mathcal{L})$, we have:*

1. $p(n) > \frac{\tilde{M}_n}{\underline{\tilde{M}}_n + \tilde{m}_n} \cdot p(a_n) := q_{\max}^C(n) \cdot p(a_n) \implies \xi_C^*(p) \neq \pi(n)$
2. $p(n) < \frac{\tilde{m}_n}{\underline{\tilde{m}}_n + \tilde{M}_n} \cdot p(a_n) := q_{\min}^C(n) \cdot p(a_n) \implies \xi_C^*(p) \neq n$

*Proof.*

Let $L : \mathcal{N} \times \mathcal{L} \to \mathbb{R}$ be a hierarchical metric that satisfy Equations (3) and (4). We now proceed to prove statement 2.

Assume $p(n) < \frac{\tilde{m}_n}{\underline{\tilde{m}}_n + \tilde{M}_n} p(a_n)$. Then, we have:

$$p(n) < \frac{\tilde{m}_n}{\underline{\tilde{m}}_n + \tilde{M}_n} p(a_n)$$

$$\iff \quad \tilde{M}_n \underbrace{p(n)}_{= \sum_{l \in \mathcal{L}(n)} p(l)} < \tilde{m}_n \underbrace{(p(a_n) - p(n))}_{\substack{= \sum_{l \in \mathcal{L} \setminus \mathcal{L}(n)} p(l) \\ \mathrm{LCA}(n,l) \neq \mathbf{r}}}$$

This implies

$$= \sum_{\substack{l \in \mathcal{L} \setminus \mathcal{L}(n) \\ \mathrm{LCA}(n,l) = \mathbf{r}}} p(l) \overbrace{(C(n, l) - C(\pi(n), l))}^{=0}$$

$$\sum_{l \in \mathcal{L}(n)} p(l) \cdot \underbrace{\tilde{M}_n}_{\geq C(\pi(n),l) - C(n,l)} < \overbrace{0} + \sum_{\substack{l \in \mathcal{L} \setminus \mathcal{L}(n) \\ \mathrm{LCA}(n,l) \neq \mathbf{r}}} p(l) \cdot \underbrace{\tilde{m}_n}_{\leq C(n,l) - C(\pi(n),l)}$$

Rewriting:

$$\sum_{l \in \mathcal{L}(n)} p(l) C(\pi(n), l) - \sum_{l \in \mathcal{L}(n)} p(l) C(n, l) < \sum_{l \in \mathcal{L} \setminus \mathcal{L}(n)} p(l) C(n, l) - \sum_{l \in \mathcal{L} \setminus \mathcal{L}(n)} p(l) C(\pi(n), l)$$

Thus:

$$\sum_{l \in \mathcal{L}} p(l) C(\pi(n), l) < \sum_{l \in \mathcal{L}} p(l) C(n, l)$$

And finally:

$$\mathcal{R}_C(\pi(n)|p) < \mathcal{R}_C(n|p)$$

Therefore, $\pi(n)$ has a strictly lower risk than $n$, which shows that $n$ is not the optimal decoding.

Statement 1 follows this exact same proof.

**Proposition E.7.** *Let $L$ be an evaluation measure that verifies 3 and 4. Let $p \in \Delta(\mathcal{L})$, then the optimal decision rule $\xi_C^*(p)$ can be obtained through an algorithm of $\mathcal{O}(d_{max} \cdot |\mathcal{L}| + |\mathcal{N}|)$ time complexity.*

---

**Algorithm 7** Candidate Selection

---

1: **function** FINDCANDSETRECBIS($n$, $p$, $S$, $q_{max}$, $q_{min}$)
2:     **if** $p(n) > q_{max}(n) \cdot p(a_n)$ **then**
3:         $S \leftarrow S \setminus \{\pi(n)\}$
4:     **else if** $p(n) < q_{min}(n) \cdot p(a_n)$ **then**
5:         $S \leftarrow S \setminus \{n\}$
6:     **else if** $n \in \mathcal{L}$ or $p(n) < \min_{n \in \mathcal{N}} q_{min}(n) \cdot p(a_n)$ **then**
7:         **return**
8:     **end if**
9:     **for** $c \in \mathcal{C}(n)$ **do**
10:         FINDCANDSETRECBIS($c$, $p$, $S$, $q_{max}$, $q_{min}$)
11:     **end for**
12: **end function**
13: **function** FINDCANDSETBIS($p$, $q_{max}$, $q_{min}$)
14:     $S \leftarrow \mathcal{N}$
15:     **Call:** FINDCANDSETRECBIS($\mathbf{r}$, $p$, $S$, $q_{max}$, $q_{min}$)
16:     **return** $S$
17: **end function**

---

**Algorithm 8** General Algorithm Bis

---

1: **function** FINDOPTIMAL($p_{leaves}$, $q_{max}$, $q_{min}$)
2:     $p \leftarrow$ PROBANODES($p_{leaves}$)
3:     $S \leftarrow$ FINDCANDSETBIS($p$, $q_{max}$, $q_{min}$)
4:     $n_{opt} \leftarrow$ BRUTEFORCE($S$, $p$)
5:     **return** $n_{opt}$
6: **end function**

---

The general algorithm is summarized in Algorithm 8. It follows the exact same intuition of Algorithm 56. It consists in

1. Compute probability distribution over nodes, given probability distribution over leaves. (Algorithm 3)
2. Filtering each node based on condition of Theorem E.6. It outputs a candidate set $S$ which can all be the optimal prediction (Algorithm 7)
3. A brute-force algorithm is performed on this remaining candidate set (Algorithm 5)

As before, We obtain a $\mathcal{O}(|S| \cdot |\mathcal{L}| + |\mathcal{N}|)$ time complexity for Algorithm 8. Let us now prove that $|S| = \mathcal{O}(d_{max})$.

We recall second condition 2 of Proposition E.6:

$$p(n) < q_{min}^C(n) \cdot p(a_n) \implies \xi_C^*(p) \neq n$$

Then, the contrapositive writes :

$$n \in S \implies p(n) \geq q_{min}^C(n) \cdot p(a_n)$$
$$\geq p(a_n) \cdot \underbrace{\min_{n \in \mathcal{N}} q_{min}^C(n)}_{:=q_{min}^C}$$

We write $\mathcal{D}(n)$ the set of descendants of $n$ in $\mathcal{T}$. (defined inclusively : $n \in \mathcal{D}(n)$. Let $c \in \mathcal{C}(\mathbf{r})$ Then,

$$\sum_{\substack{n \in S \\ n \in \mathcal{D}(c)}} \underbrace{p(n)}_{\geq p(c) \cdot q_{min}^C} \geq |S \cap \mathcal{D}(c)| \cdot q_{min}^C$$

Inequality holds also for $n = c$ because $p(c) \geq q_{\min}^C \cdot p(c)$. This can be rewritten:

$$|S \cap \mathcal{D}(c)| \leq \frac{1}{q_{\min}^C} \cdot \underbrace{\sum_{\substack{n \in S \\ n \in \mathcal{D}(c)}} p(n)}_{\leq \sum_{n \in \mathcal{D}(c)} p(n)}$$

We recall. $p(n) = \sum_{n \in \mathcal{L}(n)} p(l)$. Then, on a given level of the hierarchy *i.e* on a set $\{n \in \mathcal{N}, d(n) = d\}$, there is no common leaf descendant between two elements of the set. We therefore have

$$\sum_{\substack{n \in \mathcal{D}(c) \\ d(n) = d}} p(n) \leq \sum_{l \in \mathcal{L}(c)} p(l) = p(c)$$

Then,

$$\sum_{n \in \mathcal{D}(c)} p(n) = \sum_{d=1}^{d_{\max}} \underbrace{\sum_{\substack{n \in \mathcal{N} \\ d(n) = d}} p(n)}_{\leq p(c)} \leq d_{\max} \cdot p(c)$$

Which gives,

$$|S \cap \mathcal{D}(c)| \leq \frac{1}{q_{\min}^C} \cdot d_{\max} \cdot p(c)$$

And eventually :

$$|S| \leq 1 + \sum_{c \in \mathcal{C}(r)} |S \cap \mathcal{D}(c)| \leq 1 + \frac{1}{q_{\min}^C} \cdot d_{\max} \cdot \underbrace{\sum_{c \in \mathcal{C}(r)} p(c)}_{=1}$$

Finally this yields :

$$|S| = \mathcal{O}(d_{\max})$$

which completes the proof.

## E.2. Subset of nodes decoding

### E.2.1. ON THE CARDINALITY OF $\mathcal{M}_\mathcal{T}(\mathcal{N})$

**Proposition E.8.** *For $\mathcal{T} = (\mathcal{N}, \mathcal{E})$, where each non-leaf node has at least two children, the cardinality of $\mathcal{M}_\mathcal{T}(\mathcal{N})$ satisfies $|\mathcal{M}_\mathcal{T}(\mathcal{N})| \geq 2^{\frac{|\mathcal{N}|}{2}} - 1$.*

*Proof.*

We define the following property:

$\mathcal{P}(n) = $ "For a tree $\mathcal{T} = (\mathcal{N}, \mathcal{E})$ with $|\mathcal{N}| = n$, where every non-leaf node has at least two children, it holds that $|\mathcal{L}| \geq \frac{n}{2}$."

Let $|\mathcal{N}| = n$. We will use strong induction on $n$.

**Base Case:** For $n = 1$, the tree $\mathcal{T}$ has only one node, which is a leaf. Thus, $|\mathcal{L}| = 1$, and the inequality holds since $1 \geq \frac{1}{2}$.

**Induction Hypothesis:** Assume that for all trees with $k$ nodes (where $k < n$), the number of leaves is at least $\frac{k}{2}$.

**Inductive Step:** Consider a tree $\mathcal{T}$ with $n$ nodes. We need to show that $|\mathcal{L}| \geq \frac{n}{2}$.

Let $v$ be the deepest node in $\mathcal{T}$. Let $u$ be the parent of $v$. Since all internal nodes have at least two children, $u$ has at least two children.

Let $k$ be the number of children of $u$. All these $k$ children are leaves because $v$ is the deepest node.

We then removing these $k$ leaves and their edges from $\mathcal{T}$. This results in a smaller tree $\mathcal{T}'$ with $n - k$ nodes.

By the induction hypothesis, the number of leaves in $\mathcal{T}'$ is at least $\frac{n-k}{2}$.

The total number of leaves in the original tree $\mathcal{H}$ is the sum of:

- The $k$ leaves that were removed.
- The leaves in the smaller tree $\mathcal{T}'$.
- We need to remove $u$ from the leaf count in $\mathcal{T}'$ as $t$ is a leaf in $\mathcal{T}'$ but not in $\mathcal{H}$.

Therefore, by induction hypothesis, the number of leaves in $\mathcal{T}$ is at least:

$$k + \frac{n - k}{2} - 1$$

By simplifying the expression we obtain:

$$k + \frac{n - k}{2} - 1 = \frac{2k + n - k - 2}{2} = \frac{n + k - 2}{2}$$

We need to show:

$$\frac{n + k - 2}{2} \geq \frac{n}{2}$$

Which is true if and only if:

$$n + k - 2 \geq n$$
$$k - 2 \geq 0$$

Since $k \geq 2$ (all internal nodes have at least 2 children), this is true.

Therefore, by induction, for any tree with $n$ nodes, where every internal node has at least two children, the number of leaves $|\mathcal{L}|$ is at least $\frac{n}{2}$.

Let $\mathcal{T} = (\mathcal{N}, \mathcal{E})$ be a hierarchy. By applying $\mathcal{P}(n)$ with $n = |\mathcal{N}|$ we have that $\mathcal{T}$ has at least $\frac{|\mathcal{N}|}{2}$ leaves: $|\mathcal{L}| \geq \frac{|\mathcal{N}|}{2}$. Each subset of these $|\mathcal{L}|$ leaves corresponds to a unique element of $\mathcal{M}_\mathcal{T}(\mathcal{N})$.

There are $2^{|\mathcal{L}|} - 1$ different subsets of the $|\mathcal{L}|$ leaves (excluding the empty subset). Each of these subsets, forms a unique element of in $\mathcal{M}_\mathcal{T}(\mathcal{N})$.

Therefore,

$$|\mathcal{M}_\mathcal{T}(\mathcal{N})| \geq 2^{\frac{|\mathcal{N}|}{2}} - 1$$

This completes the proof.

### E.2.2. THE CASE OF $hF_\beta$ SCORES

Throughout this section, we will use a concept analogous to risk, referred to as utility.

**Definition E.9. Utility**. Let $p \in \Delta(\mathcal{L})$ and $h \in \mathcal{P}(\mathcal{N})$ then we define the $hF_\beta$ conditional expected utility of $h$ as follows:

$$U_{hF_\beta}(h|p) = \sum_{l \in \mathcal{L}} p(l) \cdot hF_\beta(h, l)$$

### E.2.3. REFRAMING THE PROBLEM

We recall that

$$\mathcal{M}_\mathcal{T}(\mathcal{N}) = \{h \in \mathcal{P}(\mathcal{N}), \forall(n_1, n_2) \in h, n_1 \neq n_2 \implies n_1 \notin \mathcal{A}(n_2) \text{ and } n_2 \notin \mathcal{A}(n_1)\}$$

We also define,

$$\mathcal{U}_\mathcal{T}(\mathcal{N}) = \{h^{\text{aug}}, \ h \in \mathcal{P}(\mathcal{N})\}$$

**Proposition E.10.** *The function $\phi : \mathcal{M}_\mathcal{T}(\mathcal{N}) \to \mathcal{U}_\mathcal{T}(\mathcal{N})$, defined by $\phi(h) = h^{aug}$, is bijective.*

*Proof.* An element of $\mathcal{U}_\mathcal{T}(\mathcal{N})$ represent a subtree of $\mathcal{T}$ which contains the root node. This subtree is uniquely defined by the set of leaves in the subtree, which is an element of $\mathcal{M}_\mathcal{T}(\mathcal{N})$. This completes the proof.

We first prove that

**Proposition E.11.** *Let $p \in \Delta(\mathcal{L})$ then*

$$\xi^*_{hF_\beta}(p) \in \underset{h \in \mathcal{P}(\mathcal{N})}{\operatorname{argmin}} U_{hF_\beta}(h|p) = \underset{h \in \mathcal{U}_\mathcal{T}(\mathcal{N})}{\operatorname{argmin}} U_{hF_\beta}(h|p) = \underset{h \in \mathcal{M}_\mathcal{T}(\mathcal{N})}{\operatorname{argmin}} U_{hF_\beta}(h|p)$$

*Proof.* We prove first the first equality. By definition of $hF_\beta$, $hF_\beta(h, l) = hF_\beta(h^{\text{aug}}, l)$ for any $l$. Therefore, $U_{hF_\beta}(h|p) = U_{hF_\beta}(h^{\text{aug}}|p)$ and

$$\underset{h \in \mathcal{P}(\mathcal{N})}{\operatorname{argmin}} U_{hF_\beta}(h|p) = \underset{h \in \mathcal{P}(\mathcal{N})}{\operatorname{argmin}} U_{hF_\beta}(\underbrace{h^{\text{aug}}}_{\in \mathcal{U}_\mathcal{T}(\mathcal{N})}|p) = \underset{h \in \mathcal{U}_\mathcal{T}(\mathcal{N})}{\operatorname{argmin}} U_{hF_\beta}(h|p)$$

Then for the second equality :

$$\underset{h \in \mathcal{U}_\mathcal{T}(\mathcal{N})}{\operatorname{argmin}} U_{hF_\beta}(h|p) = \underset{h \in \mathcal{U}_\mathcal{T}(\mathcal{N})}{\operatorname{argmin}} U_{hF_\beta}(\phi^{-1}(\underbrace{\phi(h)}_{=h})|p) = \underset{h \in \mathcal{M}_\mathcal{T}(\mathcal{N})}{\operatorname{argmin}} U_{hF_\beta}(h|p)$$

Last equlity holds by setting $h' = \phi^{-1}(h)$ and using Proposition E.2.3.

From now on the problem at stake is therefore :

$$\xi^*_{hF_\beta}(p) \in \underset{h \in \mathcal{U}_\mathcal{T}(\mathcal{N})}{\operatorname{argmin}} U_{hF_\beta}(h|p)$$

### E.2.4. MAIN RESULTS ABOUT $\mathrm{hF}_\beta$

We begin by proving a result we are going to use a lot. Let $\beta \in \mathbb{R}^*$, let $h \in \mathcal{U}_\mathcal{T}(\mathcal{N})$ and for $n \in \mathcal{N}$ we denote $d(n)$ the depth of node $n$. We have :

$$\mathrm{hF}_\beta(h,l) = \frac{(1+\beta^2) \cdot |h \cap \mathcal{A}(l)|}{|h| + \beta^2 \cdot (d(l)+1)} \tag{8}$$

In fact,

$$\mathrm{hR}(h,l) = \frac{|h \cap \mathcal{A}(l)|}{\underbrace{|\mathcal{A}(l)|}_{=d(l)+1}} = \frac{|h \cap \mathcal{A}(l)|}{d(l)+1}$$

And,

$$\mathrm{hP}(h,l) = \frac{|h \cap \mathcal{A}(l)|}{|h|}$$

And therefore,

$$\mathrm{hF}_\beta(h,l) = \frac{(1+\beta^2)}{\frac{1}{\mathrm{hP}(h,l)} + \beta^2 \frac{1}{\mathrm{hR}(h,l)}} = \frac{(1+\beta^2)}{\frac{|h|}{|h \cap \mathcal{A}(l)|} + \beta^2 \frac{d(l)+1}{|h \cap \mathcal{A}(l)|}} = \frac{(1+\beta^2) \cdot |h \cap \mathcal{A}(l)|}{|h| + \beta^2 \cdot (d(l)+1)}$$

We continue by proving the following result:

**Lemma E.12.** *Let $p \in \Delta(\mathcal{L})$ and $d_{min} = \max_{l \in \mathcal{L}} d(l)$ the minimum depth of leaf nodes, then $\xi^*_{\mathrm{hF}_\beta}(p) \neq \emptyset$ and the optimal expected utility $U_{\mathrm{hF}_\beta}(\xi^*_{\mathrm{hF}_\beta}(p)|p)$ is lower-bounded as follows*

$$U_{\mathrm{hF}_\beta}(\xi^*_{\mathrm{hF}_\beta}(p)|p) \geq \frac{1+\beta^2}{1+\beta^2 \cdot (d_{min}+1)} \tag{9}$$

*Proof.*

$$U_{\mathrm{hF}_\beta}(\emptyset|p) = 0$$

And,

$$U_{\mathrm{hF}_\beta}(\mathbf{r}|p) = \frac{(1+\beta^2) \cdot \overbrace{|\mathbf{r} \cap \mathcal{A}(l)|}^{=1}}{\underbrace{|\mathbf{r}|}_{=1} + \beta^2 \cdot \underbrace{(d(l)+1)}_{\geq d_{min}}} \leq \frac{1+\beta^2}{1+\beta^2(d_{min}+1)}$$

Therefore, as $U_{\mathrm{hF}_\beta}(\mathbf{r}|p) > U_{\mathrm{hF}_\beta}(\emptyset|p)$, $\xi^*_{\mathrm{hF}_\beta}(p) \neq \emptyset$ and,

$$U_{\mathrm{hF}_\beta}(\xi^*_{\mathrm{hF}_\beta}(p)|p) \geq U_{\mathrm{hF}_\beta}(\mathbf{r}|p) \geq \frac{1+\beta^2}{1+\beta^2 \cdot (d_{min}+1)}$$

This completes the proof.

Let us now prove the following lemma.

**Lemma E.13.** *Let $p \in \Delta(\mathcal{L})$ and $n \in \mathcal{N} \backslash \{\mathbf{r}\}$ and $d_{max}(n) = \max_{l \in \mathcal{L}(n)} d(l)$ the leaf nodes maximum depth among leaf descendants of $n$. Then,*
$$p(n) < \frac{1}{1+\beta^2(d_{max}(n)+1)} := q_{min}^{\mathrm{hF}_\beta}(n) \implies n \notin \xi^*_{\mathrm{hF}_\beta}(p)$$

*Proof.* Let $h \in \mathcal{U}_\mathcal{T}(\mathcal{N})$ be non-empty. Let $n \in \mathcal{N} \backslash h$ then,

$$U_{\mathrm{hF}_\beta}(h \cup \{n\}) = \sum_{l \in \mathcal{L}} p(l)\, \mathrm{hF}_\beta(h \cup \{y\}, l) = \sum_{l \in \mathcal{L}} p(l)\, \frac{(1+\beta^2) \cdot |h \cup \{y\} \cap \mathcal{A}(l)|}{\underbrace{|h \cup \{y\}|}_{=|h|+1} + \beta^2(d(l)+1)}$$

$$= \sum_{l \in \mathcal{L}(n)} p(l)\, \frac{(1+\beta^2) \cdot \overbrace{|h \cup \{y\} \cap \mathcal{A}(l)|}^{=|h \cap \mathcal{A}(l)|+1}}{|h|+1+\beta^2(d(l)+1)} + \sum_{l \in \mathcal{L} \setminus \mathcal{L}(n)} p(l)\, \frac{(1+\beta^2) \cdot \overbrace{|h \cup \{y\} \cap \mathcal{A}(l)|}^{=|h \cap \mathcal{A}(l)|}}{|h|+1+\beta^2(d(l)+1)}$$

$$= \sum_{l \in \mathcal{L}(n)} p(l)\, \frac{1+\beta^2}{|h|+1+\beta^2(d(l)+1)} + \sum_{l \in \mathcal{L}} p(l)\, \frac{(1+\beta^2) \cdot |h \cap \mathcal{A}(l)|}{|h|+1+\beta^2(d(l)+1)}$$

Hence,

$$U_{\mathrm{hF}_\beta}(h \cup \{n\}) - U_{\mathrm{hF}_\beta}(h) =$$
$$\sum_{l \in \mathcal{L}(n)} p(l)\, \frac{1+\beta^2}{|h|+1+\beta^2(d(l)+1)} + \sum_{l \in \mathcal{L}} p(l)\, \frac{(1+\beta^2) \cdot |h \cap \mathcal{A}(l)|}{|h|+1+\beta^2(d(l)+1)} - \sum_{l \in \mathcal{L}} p(l)\, \frac{(1+\beta^2) \cdot |h \cap \mathcal{A}(l)|}{|h|+\beta^2(d(l)+1)}$$

Combining second and third term we obtain :

$$U_{\mathrm{hF}_\beta}(h \cup \{n\}|p) - U_{\mathrm{hF}_\beta}(h|p)$$
$$= \sum_{l \in \mathcal{L}(n)} p(l)\, \frac{1+\beta^2}{|h|+1+\beta^2(\underbrace{d(l)+1}_{\geq d_{\min}+1})} - \sum_{l \in \mathcal{L}} p(l)\, \frac{(1+\beta^2) \cdot |h \cap \mathcal{A}(l)|}{(|h|+1+\beta^2(\underbrace{d(l)+1}_{\leq d_{\max}(n)+1}))(|h|+\beta^2(d(l)+1))}$$
$$\leq \frac{1+\beta^2}{|h|+1+\beta^2(d_{\min}+1)} \underbrace{\sum_{l \in \mathcal{L}(n)} p(l)}_{=p(n)} - \frac{1}{|h|+1+\beta^2(d_{\max}(n)+1)} \underbrace{\sum_{l \in \mathcal{L}} p(l)\, \frac{(1+\beta^2) \cdot |h \cap \mathcal{A}(l)|}{|h|+\beta^2(d(l)+1)}}_{=U_{\mathrm{hF}_\beta}(h|p)}$$
$$\leq \frac{1+\beta^2}{|h|+1+\beta^2(d_{\min}+1)} p(n) - \frac{1}{|h|+1+\beta^2(d_{\max}(n)+1)} U_{\mathrm{hF}_\beta}(h|p)$$

Now, let us conclude the proof.

If $\xi^*_{\mathrm{hF}_\beta}(p) = \{\mathbf{r}\}$ then $\mathrm{hF}_\beta \subset \{n \in \mathcal{N}, p(n) \geq \frac{1}{1+\beta^2(d_{\max}(n)+1)}\}$ because $p(\mathbf{r}) = 1$. If $\xi^*_{\mathrm{hF}_\beta}(p) \neq \{\mathbf{r}\}$ then Lemma E.2.4 gives us $\xi^*_{\mathrm{hF}_\beta}(p) \neq \emptyset$.

Now suppose, $\xi^*_{\mathrm{hF}_\beta}(p) \not\subset \{n \in \mathcal{N}, p(n) \geq \frac{1}{1+\beta^2(d_{\max}(n)+1)}\}$: it exists $n \in \xi^*_{\mathrm{hF}_\beta}(p)$ such that $p(n) < \frac{1}{1+\beta^2(d_{\max}(n)+1)}$. Then consider $h = \xi^*_{\mathrm{hF}_\beta}(p) \setminus \{n\}$, then previous result give us :

$$U_{\mathrm{hF}_\beta}(\xi^*_{\mathrm{hF}_\beta}(p)|p) - U_{\mathrm{hF}_\beta}(h|p) \leq \frac{1+\beta^2}{|h|+1+\beta^2(d_{\min}+1)} \underbrace{p(n)}_{<\frac{1}{1+\beta^2(d_{\max}(n)+1)}} - \frac{1}{|h|+1+\beta^2(d_{\max}(n)+1)} \underbrace{U_{\mathrm{hF}_\beta}(h|p)}_{\geq \frac{1+\beta^2}{1+\beta^2 \cdot (d_{\min}+1)}}$$

By reducing to the same denominator we obtain:

$$U_{\mathrm{hF}_\beta}(\xi^*_{\mathrm{hF}_\beta}(p)|p) - U_{\mathrm{hF}_\beta}(h|p)$$
$$< \frac{|h|\beta^2}{(|h|+1+\beta^2(d_{\min}+1))(1+\beta^2(d_{\max}(n)+1))(|h|+1+\beta^2(d_{\max}(n)+1))(1+\beta^2(d_{\min}+1))} \underbrace{(d_{\min} - d_{\max}(n))}_{\leq 0} < 0$$

which is impossible because by definition of $\xi^*_{\mathrm{hF}_\beta}(p)$, we have $U_{\mathrm{hF}_\beta}(\xi^*_{\mathrm{hF}_\beta}(p)|p) > U_{\mathrm{hF}_\beta}(h|p)$. This concludes the proof :
$\xi^*_{\mathrm{hF}_\beta}(p) \subset \{n \in \mathcal{N}, p(n) \geq \frac{1}{1+\beta^2(d_{\max}(n)+1)}\}$

**Lemma E.14.** *Let $\mathcal{T}$ be a hierarchy, and $d_{max} = \max_{l \in \mathcal{L}} d(l)$ the maximum depth of its leaf nodes. Let $p \in \Delta(\mathcal{L})$ and $n \in \mathcal{N} \setminus \{\mathbf{r}\}$. Then, $\xi^*_{\mathrm{hF}_\beta}(p) \subset \{n \in \mathcal{N}, p(n) \geq \frac{1}{1+\beta^2(d_{max}+1)}\} := \mathcal{Q}(p)$. and therefore, $|\xi^*_{\mathrm{hF}_\beta}(p)| \leq |\mathcal{Q}(p)|$ where $|\mathcal{Q}(p)| \leq N_{max} = (1 + \beta^2(d_{max} + 1)) \cdot (d_{max} + 1) = \mathcal{O}(d^2_{max})$*

*Proof.* The first result is a direct implication of Lemma E.13. Let us now show that, $|\mathcal{Q}(p)| = \mathcal{O}(d^2_{\max})$.

Let $p \in \Delta(\mathcal{L})$, the first result implies that:

$$n \in \xi^*_{\mathrm{hF}_\beta}(p) \implies p(n) \geq \frac{1}{1 + \beta^2(d_{\max} + 1)}$$

Therefore,

$$\sum_{n \in \mathrm{hF}_\beta} \underbrace{p(n)}_{\geq \frac{1}{1+\beta^2(d_{\max}+1)}} \geq |\xi^*_{\mathrm{hF}_\beta}(p)| \cdot \frac{1}{1 + \beta^2(d_{\max} + 1)}$$

This can be rewritten:

$$|\xi^*_{\mathrm{hF}_\beta}(p)| \leq (1 + \beta^2(d_{\max} + 1)) \cdot \underbrace{\sum_{n \in \xi^*_{\mathrm{hF}_\beta}(p)} p(n)}_{\leq \sum_{n \in \mathcal{N}} p(n)}$$

We already proved in Equation 7 of the proof of Lemma E.3 that,

$$\sum_{n \in \mathcal{N}} p(n) \leq d_{\max} + 1$$

This finally gives,

$$|\xi^*_{\mathrm{hF}_\beta}(p)| \leq (1 + \beta^2(d_{\max} + 1)) \cdot (d_{\max} + 1)$$

Denoting $N_{\max} = (1 + \beta^2(d_{\max} + 1)) \cdot (d_{\max} + 1)$, we have

Therefore $|\xi^*_{\mathrm{hF}_\beta}(p)| \leq N_{\max}$ and $N_{\max} = \mathcal{O}(d^2_{\max})$.

This completes the proof.

Let us now prove the main theoretical result of the article.

**Theorem E.15.** *Let $d_{max} = \max_{l \in \mathcal{L}} d(l)$ be the leaf nodes maximum depth. Let $p \in \Delta(\mathcal{L})$, then the optimal decision rule $\xi^*_{\mathrm{hF}_\beta}(p)$ can be obtained through an algorithm of $\mathcal{O}(d^2_{max} \cdot |\mathcal{N}|)$ time complexity.*

Let us recall that the task, is to find the optimal decision rule for the metric $\mathrm{hF}_\beta$, which is given by $\xi^*_{\mathrm{hF}_\beta} : \Delta(\mathcal{L}) \to \mathcal{H}$, where:

$$\xi^*_{\mathrm{hF}_\beta}(p) = \underset{h \in \mathcal{U}_{\mathcal{T}}(\mathcal{N})}{\mathrm{argmax}} \; U_{\mathrm{hF}_\beta}(h|p)$$

We adopt here a similar approach to Waegeman et al. (2014) to deal with the optimization problem. Let $k \leq |\mathcal{N}|$, let us denote $\mathcal{U}^k_{\mathcal{T}}(\mathcal{N}) = \{h \in \mathcal{P}(\mathcal{N}), h = h^{\mathrm{aug}} \text{ and } |h| = k\}$ the set of parts of $\mathcal{N}$ of $k$ elements which belong to $\mathcal{U}_{\mathcal{T}}(\mathcal{N})$ then

$$\xi^*_{\mathrm{hF}_\beta}(p) = \underset{k \in \{1 \dots |\mathcal{N}|\}}{\mathrm{argmax}} \; \underset{h_k \in \mathcal{U}^k_{\mathcal{T}}(\mathcal{N})}{\mathrm{argmax}} \; U_{\mathrm{hF}_\beta}(h_k|p)$$

With Lemma E.14 we can restrict the search to

- $k \leq |\mathcal{Q}(p)|$
- $h_k \subset \mathcal{Q}(p)$

And therefore, we can rewrite the problem as follows:

$$\xi^*_{\mathrm{hF}_\beta}(p) = \underset{\substack{k \in \{1...|\mathcal{Q}(p)|\} \\ }}{\mathrm{argmax}} \; \underset{\substack{h_k \in \mathcal{U}^k_{\mathcal{T}}(\mathcal{N}) \\ h_k \subset \mathcal{Q}(p)}}{\mathrm{argmax}} \; U_{\mathrm{hF}_\beta}(h_k|p)$$

Next, we rewrite $U_{\mathrm{hF}_\beta}(h|p)$ as follows :

$$
\begin{aligned}
U_{\mathrm{hF}_\beta}(h|p) &= \sum_{l \in \mathcal{L}} p(l) \, \mathrm{hF}_\beta(h, l) \\
&= \sum_{l \in \mathcal{L}} p(l) \frac{(1 + \beta^2) \cdot \overbrace{|h \cap \mathcal{A}(l)|}^{= \sum_{n \in \mathcal{N}} \mathbf{1}(n \in h \cap \mathcal{A}(l))}}{|h| + \beta^2 \cdot (d(l) + 1)} \\
&= \sum_{l \in \mathcal{L}} p(l) \frac{(1 + \beta^2)}{|h| + \beta^2(d(l) + 1)} \sum_{n \in \mathcal{N}} \mathbf{1}(n \in \mathcal{A}(l)) \mathbf{1}(n \in h) \\
&= \sum_{n \in \mathcal{N}} \mathbf{1}(y \in h) \sum_{l \in \mathcal{L}} p(l) \mathbf{1}(y \in \mathcal{A}(l)) \frac{2}{|h| + d(l) + 1} \\
&= \sum_{n \in \mathcal{N}} \mathbf{1}(n \in h) \sum_{l \in \mathcal{L}(n)} p(l) \frac{1 + \beta^2}{|h| + \beta^2(d(l) + 1)}
\end{aligned}
$$

So that, when $|h| = k$ we have,

$$U_{\mathrm{hF}_\beta}(h_k|p) = \sum_{n \in \mathcal{N}} \mathbf{1}(n \in h) \underbrace{\sum_{l \in \mathcal{L}(n)} p(l) \frac{1 + \beta^2}{k + \beta^2(d(l) + 1)}}_{:= \Delta^\beta_k(n)}$$

Therefore the problem can be rewritten:

$$\xi^*_{\mathrm{hF}_\beta}(p) = \underset{\substack{k \in \{1...|\mathcal{Q}(p)|\} \\ }}{\mathrm{argmax}} \; \underset{\substack{h_k \in \mathcal{U}^k_{\mathcal{T}}(\mathcal{N}) \\ h_k \subset \mathcal{Q}(p)}}{\mathrm{argmax}} \sum_{n \in \mathcal{N}} \mathbf{1}(n \in h) \Delta^\beta_k(n) \qquad (10)$$

Based on Equation 10 we propose then the following strategy to find the optimal decision rule

1. We obtain $\mathcal{Q}(p)$ and compute $\Delta^\beta_k(n)$ for $k \leq |\mathcal{Q}(p)|$ and $n \in \mathcal{Q}(p)$
2. For each $k \leq |\mathcal{Q}(p)|$, we find $h^*_k$ which consists in the $k$ nodes that correspond to the top-$k$ $(\Delta^\beta_k(n))_{n \in \mathcal{Q}(p)}$ and compute $U_{\mathrm{hF}_\beta}(h^*_k|p) = \sum_{n \in h^*_k} \Delta^\beta_k(n)$
3. We find optimal rule with $k^{\mathrm{opt}} = \max_{k \leq |\mathcal{Q}(p)|} U_{\mathrm{hF}_\beta}(h^*_k|p)$ and $\xi^*_{\mathrm{hF}_\beta}(p) = h^*_{k^{\mathrm{opt}}}$

---

**Algorithm 9** Computation of depths

---

1: **function** DREC($n, \delta, d$)
2:      **if** $n \in \mathcal{L}$ **then**
3:          $d(n) \leftarrow \delta$
4:          **return**
5:      **else**
6:          **for** $c \in \mathcal{C}(r)$ **do**
7:              $d(n) \leftarrow \delta$
8:              DREC($c, \delta + 1, d$)
9:          **end for**
10:      **end if**
11: **end function**
12: **function** D()
13:      $d \leftarrow 0_{|\mathcal{N}|}$
14:      DREC($\mathbf{r}, 0, d$)
15:      **return** $d$
16: **end function**

---

**Algorithm 10** Computation of $\mathcal{Q}(p)$

---

1: **function** QREC($n, p, d, Q$)
2:      **if** $p(n) \geq \frac{1}{1+\beta^2(d_{\max(n)}+1)}$ **then**
3:          $Q \leftarrow Q \cup \{n\}$
4:          **if** $n \in \mathcal{L}$ **then**
5:              **return**
6:          **else**
7:              **for** $c \in \mathcal{C}(n)$ **do**
8:                  QREC($c, p, d, Q$)
9:              **end for**
10:          **end if**
11:      **else**
12:          **return**
13:      **end if**
14: **end function**
15: **function** Q($p, d$)
16:      $Q \leftarrow \emptyset$
17:      QREC($\mathbf{r}, p, d, Q$)
18:      **return** $Q$
19: **end function**

---

**Algorithm 11** Computation of $\Delta$

---

1: **function** DELTAREC($n, p, d, k, \Delta, \beta, d$)
2:      **if** $n \in \mathcal{L}$ **then**
3:          $\Delta(n) \leftarrow p(n) \cdot \frac{1+\beta^2}{k+\beta^2(d(n)+1)}$
4:          **return** $\Delta(n)$
5:      **else**
6:          $\Delta(n) \leftarrow \sum\limits_{c \in \mathcal{C}(n)}$ DELTAREC($c, p, k, \Delta, \beta, d$)
7:          **return** $\Delta(n)$
8:      **end if**
9: **end function**
10: **function** DELTA($p, d, \beta, q$)
11:      $\Delta \leftarrow 0_{q \times \mathcal{N}}$
12:      **for** $k \leftarrow 0$ to $q$ **do**
13:          DELTAREC($\mathbf{r}, p, d, k, \Delta(k), \beta$)
14:      **end for**
15:      **return** $\Delta$
16: **end function**

---

**Algorithm 12** Optimal decision rule for $\mathrm{hF}_\beta$

---

[H]
1: **function** OPTIMALK($k, \Delta, \mathcal{Q}$)
2:      $h^* \leftarrow \underset{n \in \mathcal{Q}}{\operatorname{argtop}k}\ \Delta(n)$
3:      $u \leftarrow \sum_{n \in h^*} \Delta(n)$
4:      **return** $h^*, u$
5: **end function**
6: **function** OPTIMAL($\Delta, \mathcal{Q}$)
7:      $h^* \leftarrow \emptyset$
8:      $u_{\max} \leftarrow -\infty$
9:      **for** $k \in \{1 \ldots |\mathcal{Q}|\}$ **do**
10:          $h, u \leftarrow$ OPTIMALK($k, \Delta(k), \mathcal{Q}$)
11:          **if** $u > u_{\max}$ **then**
12:              $u_{\max} \leftarrow u$
13:              $h^* \leftarrow h$
14:          **end if**
15:      **end for**
16:      **return** $h^*$
17: **end function**

---

**Algorithm 13** General Algorithm for $\mathrm{hF}_\beta$

---

1: **function** FINDOPTIMAL($p_{\text{leaves}}, \beta$)
2:      $p \leftarrow$ PROBANODES($p_{\text{leaves}}$)
3:      $d \leftarrow$ D()
4:      $Q \leftarrow$ Q($p, d$)
5:      $\Delta \leftarrow$ DELTA($p, d, \beta, |Q|$)
6:      $h^* \leftarrow$ OPTIMAL($\Delta, Q$)
7:      **return** $h^*$
8: **end function**

---

**Complexity**

General algorithm for finding $\xi^*_{\mathrm{hF}_\beta}(p)$ is summarized in Algorithm 12.

It consists in :

1. Node probability computation. This is performed through a tree traversal which visit exactly once each node. Therefore the time complexity is $\mathcal{O}(|\mathcal{N}|)$
2. Depths computation. Similarly, it is a tree traversal which has $\mathcal{O}(|\mathcal{N}|)$ time complexity
3. $\mathcal{Q}(p)$ computation. This is also performed through a tree traversal which visit at most each node one. Therefore the time complexity is $\mathcal{O}(|\mathcal{N}|)$. This three first steps can be performed simultaneously. For convenience, we separate to make it more understandable.
4. $\Delta$ computation. For each $k$ in $\{1 \dots |\mathcal{Q}(p)|\}$, $\Delta(k)$ is computed through a tree traversal in which each node is visited exactly once. Therefore this step has a $\mathcal{O}(|\mathcal{Q}(p)| \cdot |\mathcal{N}|) = \mathcal{O}(d^2_{\max} \cdot |\mathcal{N}|)$.
5. The last step consists in enumerating all $k \in \{1 \dots |\mathcal{Q}(p)|\}$. Each step requires first to find the top $k$ elements of among $|\mathcal{Q}(p)|$ which can be performed in $|\mathcal{Q}(p)|$ operation and then compute the sum this $k$ element. Overall each step has then a $\mathcal{O}(|\mathcal{Q}(p)| + k) = \mathcal{O}(|\mathcal{Q}(p)|) = \mathcal{O}(d^2_{\max})$ time complexity. Then, in total the total complexity is $\mathcal{O}(d^4_{\max})$

In total the algorithm has therefore an overall time complexity of $\mathcal{O}(d^2_{\max} \cdot |\mathcal{N}|)$. Ths completes the proof.

