# OpenReview forum: "To Each Metric Its Decoding: Post-Hoc Optimal Decision Rules of Probabilistic Hierarchical Classifiers"
_ICML.cc/2025/Conference — ICML 2025 poster_

### Official Review · Reviewer_vH3L · 2025-02-19

**Overall Recommendation:** 2

**Summary:**

This papers investigates the problem of Bayes-optimal prediction in hierarchical multi-class classification. A decision-theoretic framework is assumed, where probabilities are estimated during training, and the Bayes-optimal prediction is computed at test time.

The authors consider three types of settings, where the output of the Bayes-optimal classifier is a leaf, a tree node, or a set of tree nodes, respectively. For the three settings, they show how Bayes-optimal predictions can be obtained for various loss functions. In the experiments, Bayes-optimal and heuristic decodings are compared to each other, showing that Bayes-optimal decodings give better results. Additional experiments are conducted to show that the advantage of Bayes-optimal decondings becomes bigger for specific probability distributions.

**Claims And Evidence:**

I think that most claims are supported by convincing evidence.

However, the claim in Theorem 4.7 is hard to believe. I think that this claim can only hold if additional restrictions are made for the loss function. For the hierarchical F-measure, this claim is probably true, but I would be surprised that the claim holds for the hierarchical Jaccard-measure (as far as I know, this measure has not been used in hierarchical classification, but it only differs from the F-measure via an extra term in the denominator). Some authors, such as Chierchetti et al. have claimed that optimal decoding for the Jaccard is and NP-hard problem.

F. Chierichetti, R. Kumar, S. Pandey, and S. Vassilvitskii. Finding the Jaccard median. In
Proceedings of ACM-SIAM Symposium on Discrete Algorithms (SODA), pages 293–311,
2010.

So, I think that Theorem 4.7 has to be reviewed by the authors.

Theorem 4.4 also looks a bit weird to me. It says that the Bayes-optimal prediction can be computed in O(d_max x |L| + |N|) time, but the the prediction is a single node, so a brute-force algorithm that investigates all nodes is also linear in the number of nodes? What I am missing here?

Overall I find the paper not very easy to follow, because essential information is put in appendices. I would have liked to see the algorithms in the main paper. Section 3 could be written in a more compact manner by removing the definition environments for basic concepts. It is also not very easy to assess what the key novelty is. The paper considers a rather general setting, with three types of predictions, for which theoretical results have been established in previous works. In Section 4 I was missing a clear description of what is novel, and what is incremental to previous papers.

**Essential References Not Discussed:**

I think that the most important references are included.

**Experimental Designs Or Analyses:**

See above.

**Methods And Evaluation Criteria:**

For me it is a weird choice to use flat classifiers during training. These classifiers ignore the hierarchy, but more specialized probabilistic models have been proposed in the past, such as nested dichotomies, probabilistic classifier trees and neural networks with hierarchical softmax output layer. It is unclear whether the proposed algorithms can be used together with hierarchical probabilistic classifiers.

The authors only analyze two datasets. I find this quite limited, because there are other hierarchical classification datasets available from several domains (vision, text, biology, etc. ). Having more datasets could be useful to study the issues reported in Section 5.2 and 5.3 more in detail. More specifically, the authors report that the improvement of Bayes-optimal decodings over heuristic is problem dependent. Similar issues have been reported in multi-label classification. For example, the improvement of Bayes-optimal versus heuristic algorithms for the F-measure depends on the underlying probability distribution. For the case where labels are independent, heuristic algorithms work better. Perhaps similar phenomena exist in hierarchical classification, which could also be studied in a more theoretical manner.

**Other Comments Or Suggestions:**

No.

**Other Strengths And Weaknesses:**

Strengths:
- I think that the paper contains novel theoretical results, but I am not sure.
- The paper is relatively well written, but some improvements can be made.

Weaknesses:
- The story is a bit unclear.
- I am not convinced of some of the theoretical results.
- More datasets in the experiments could be useful.

**Questions For Authors:**

See above.

For the moment I am voting for rejecting the paper, but I might change my mind after the rebuttal phase.

**Relation To Broader Scientific Literature:**

I believe that the topic of the paper is relevant, but it is still a bit unclear to me what's novel and what's incremental to previous work.

**Theoretical Claims:**

See above.

---

> ### Author Rebuttal · Authors · 2025-04-01
>
> We thank the reviewer for their valuable feedback and insightful points. Below, we address each question concisely.
>
> *On Theorem 4.7 and the Jaccard Index*
>
> We would like to clarify that Theorem 4.7 and Section 4.3 **only** applies to hFβ-score and not to the Jaccard Index, or any other metric. As stated, Th. 4.7 is correct, though we will refine its formulation for clarity.
>
> During our research, we did attempt to extend Th. 4.7 to the Jaccard Index but were unsuccessful. This highlights the complexity of directly optimizing the Jaccard Index, which is indeed known to be an NP-hard problem.
> The reviewer's observation on the link between F1 and Jaccard aligns with Waegeman et al. (2014), who provide theoretical bounds on the regret of using F1 decoding as a surrogate for the Jaccard Index. This suggests that Th. 4.7 could offer an alternative to bypass the intractability of Jaccard Index decoding in hierarchical classification.
>
> Overall, and as noted by the reviewer, we also decided not to use the Jaccard Index as a metric due to its limited adoption in the context of hierarchical classification.
>
> *On Theorem 4.4 and Complexity*
>
> A brute-force algorithm that evaluates all nodes has a time complexity of O(N * L), not O(N). In fact, for each node n, computing the expectation E[L(n,Y)]= Σ_l p(l) L(n, l) requires O(L) operations. Our O(d_max * L + N) algorithm therefore improves upon the O(N * L) brute-force search.
>
> *On Paper Readability and Algorithm Placement*
>
> We acknowledge the clarity concern and will integrate key algorithms into the main text for better readability.
>
> *On Basic Concept Inclusion*
>
> In fact, we had a hard time deciding what to include and what to exclude. One of our goals was to make the paper accessible to the hierarchical classification community, which may not be deeply familiar with Bayes-optimal decoding concepts as it is often an overlooked aspect in empirical research on hierarchical classification. This led us to include some fundamental explanations which may appear basic for an initiated reader. Still, we will try to figure out a way to make it more straightforward, as the reviewer suggests.
>
> *On Novelty in Section 4*
>
> We acknowledge this remark and will clarify contributions by adding an introduction to Section 4. Key novelties are:
>
> **Node candidate set (Th. 4.4)**: A general optimal decoding algorithm for *hierarchically reasonable* metrics, improving complexity to O(d_max * L + N) from O(N * L).\
> **Subset of nodes candidate set (Th. 4.7)**: An optimal decoding algorithm for hF_β scores in O(d_max² * N), improving on intractable brute-force search.
>
> *On Training with Flat/Hierarchical Classifiers and applicability of our decoding methods to hierarchically-aware probability estimates*
>
> Our decoding algorithms **do not make any assumptions** about the probability estimation procedure. They only require a probability distribution over leaves, which can be obtained from either a flat or a hierarchically-aware classifier. In practice, **we did use hierarchical-aware classifiers**. As detailed in Subsubsection Models of Section 5, our experiments included several hierarchy-aware classifiers, including the hierarchical softmax (referred to as conditional softmax, as in its original introduction). The decision to use flat classifiers from the PyTorch library was primarily driven by their applicability to our task as well their widespread availability, which facilitated the scalability of our experiments.
>
> However, we acknowledge that our selected models did not include nested dichotomies or probabilistic classifier trees.
>
> *On Dataset Scope*
>
> While we acknowledge that additional datasets could have been included, we believe the current scope is already comprehensive. As reported, we provide results for 6 metrics, compare our methods against 8 baselines, and evaluate 19 different models across 2 datasets. We draw inspiration from recent literature (Bertinetto et al., 2020; Kharthik et al., 2021; Valmadre, 2022; Garg et al., 2022; and Jain et al., 2023), which focuses on these 2 datasets.
> That said, we have initiated experiments on a text dataset (Web of Science, Koswari et al., 2018), and we will consider adding them in the paper.
>
> *On Problem-Dependence of Bayes-Optimal Decoding*
>
> The phenomenon highlighted by the reviewer regarding the dependency of improvement on the underlying distribution closely aligns with our experiment using progressive blurring of images. By applying blur, we shift the underlying probability distributions towards the center of the simplex. As shown in Figure 4, increased blurriness further emphasizes the importance of optimal decoding.
>
> We agree that the theoretical study of the regret associated with using a heuristic algorithm versus the optimal algorithm is an interesting topic, as explored in works such as Waegeman et al. (2014). While such theoretical considerations are indeed challenging, we view them as a future research direction.

---

> > ### Comment · Reviewer_vH3L · 2025-04-02
> >
> > I thank the authors for the detailed response. It made a few points more clear to me.

---

> > > ### Author Response · Authors · 2025-04-07
> > >
> > > As we come closer to the end of the rebuttal, we kindly ask the reviewer to reconsider their score in light of the addressed concerns.

---

### Official Review · Reviewer_NL1p · 2025-03-13

**Overall Recommendation:** 2

**Summary:**

This paper tackles optimal decoding in hierarchical classification by developing universal algorithms for hierarchically reasonable metrics and a specialized algorithm for hF$_β$-scores. These methods, designed to find the best prediction given a posterior probability distribution, are particularly effective in underdetermined classification tasks, as demonstrated empirically.

**Claims And Evidence:**

Yes.

**Essential References Not Discussed:**

N/A.

**Experimental Designs Or Analyses:**

Figure 2 and Tables 3 through 9 lack error bars and statistical significance testing. Therefore, I cannot determine the significance of the reported results.

**Methods And Evaluation Criteria:**

Yes.

**Other Comments Or Suggestions:**

The visibility of Figure 2 is poor.

**Other Strengths And Weaknesses:**

The proposed method assumes that the exact posterior probability distribution is given, which is a significant limitation. In fact, I believe that estimating the posterior probability distribution is more challenging than the 'Bayes-optimal decoding' itself. Therefore, I do not consider the method presented in this paper to be significant.

**Questions For Authors:**

1. Could the authors add the missing error bars and statistical significance testing for the experiments?
2. How does the proposed method compare to prior work, for example, Cao et al. (2024), when used with the tree distance loss and its generalized forms?

## Update after rebuttal

The reviewer has addressed several of my earlier concerns, which has helped me better appreciate the main contributions of the paper. However, one important issue remains unresolved: the comparison between the proposed algorithm and that of Cao et al. (2024)—particularly in terms of the tree distance loss and its generalized variants. Despite my explicit request in the review, the authors did not provide any empirical results on this point. I consider this a significant omission, and therefore, I maintain my current score.

**Relation To Broader Scientific Literature:**

The discussion of related work in Section 2 appears reasonable.

**Theoretical Claims:**

I did not check the proofs.

---

> ### Author Rebuttal · Authors · 2025-04-01
>
> We thank the reviewer for their feedback even though we find the review quite unfair and not very detailed.
>
> We try to answer the few elements you point out in the review.
>
> *The proposed method assumes that the exact posterior probability distribution is given, which is a significant limitation.*
>
> We respectfully disagree with the reviewer on this point. We develop theoretical algorithms that are, in fact, based on the knowledge of the exact posterior probability distribution. However, in all our experiments, that restriction is relaxed, and we apply these algorithms to the estimated posterior probability distribution. We then show the empirical superiority of these theoretical algorithms.
>
> *In fact, I believe that estimating the posterior probability distribution is more challenging than the 'Bayes-optimal decoding' itself. Therefore, I do not consider the method presented in this paper to be significant.*
>
> These are, in fact, two distinct research directions, and we believe both are interesting but they also can be seen as orthogonal problems. We strongly believe that Bayes-Optimal Decoding is worth investigating and is significant for several key reasons, which we outline below.
>
> First, the use of pre-trained models is becoming increasingly common and many users lack the computational resources or expertise to train these models themselves: knowing how to correctly decode these pre-trained models for a given application is therefore crucial. Second, the costs associated with misclassification errors are not necessarily known during training. Lastly, these misclassification costs are of course context-dependent across applications, and can vary through time.
>
> Our paper thus aims to provide a general framework for decoding hierarchical classifiers with respect to any specified metric.
>
> *The visibility of Figure 2 is poor.*
>
> We will improve the visibility of Figure 2 for better clarity.
>
> *Could the authors add the missing error bars and statistical significance testing for the experiments?*
>
> Figure 2 is a boxplot, which already provides information about the distribution of results across models.
>
> However, it is true that we did not run multiple experiments for each of the 19 models tested, and there are several reasons for this.
>
> First, unlike learning algorithms, our decoding algorithms are deterministic, meaning the same probability will always produce the same prediction, regardless of the context.
>
> Second, when using pre-trained models, they come with a single checkpoint and a predefined test set, making it uncommon to report confidence intervals in this setup.
>
> Lastly, for fine-tuned models with hierarchical losses, we could have conducted multiple trainings (with different seeds), but we chose not to, in order to avoid creating a dissymmetry with the pre-trained models.
>
> *How does the proposed method compare to prior work, for example, Cao et al. (2024), when used with the tree distance loss and its generalized forms?*
>
> The comparison of our newly introduced Bayes-optimal decodings to prior work is presented throughout the experimental section (Section 5). We have selected 8 baselines for comparison, as shown in Figure 2.
>
> Cao et al. (2024) provide a theoretical result on Bayes-optimal decoding for the generalized version of the tree distance loss. We propose a more general framework and recover the Cao et al. result with Lemma 4.3, as discussed in Section 4.2.1.
>
> In our empirical results, we chose to use only the Tree Distance Loss rather than its generalized version, due to its limited adoption in the hierarchical classification literature. However, if we had used the generalized version, our algorithm would have performed equivalently to Cao et al.'s closed-form decoding, as both are optimal.

---

### Official Review · Reviewer_k32x · 2025-03-13

**Overall Recommendation:** 4

**Summary:**

In this paper, the authors study the problem of hierarchical classification, i.e., a variant of multiclass classification problem with a predefined label hierarchy. The main focus of this paper is the decoding of optimal prediction w.r.t. a family of performance metrics called hierarchically reasonable metric from the estimated class probability. The authors first show that the family of performance metrics include a number of existing metrics, and then provide general decoding algorithms with optimality guarantees. Extensive experiments on different real-world hierarchical classification benchmarks and different metrics demonstrates the effectiveness of the proposed decoding method.

**Claims And Evidence:**

The claims made in the submission are supported by clear and convincing evidence.

**Essential References Not Discussed:**

The related works are thoroughly discussed and compared in Section 2.

**Experimental Designs Or Analyses:**

The experiment designs and analyses are valid and of real-world significance.

**Methods And Evaluation Criteria:**

The selected benchmark datasets are well-suited for hierarchical classification scenarios, while sample blurring effectively simulates the increasing “hardness” of samples.

**Other Comments Or Suggestions:**

it is encouraged to include the proposed algorithms or provide a more detailed description of them in the main body.

**Other Strengths And Weaknesses:**

1. This work provides a unified framework of eliciting Bayes optimal prediction of from class probability, which is flexible and can be combined with arbitrary proper scoring rule for class probability estimation.

2. The perspective of this work is also novel in that it focuses on the inference of optimal prediction over potentially complex prediction space, which is often neglected in the research of traditional multiclass/multilabel classification.

**Questions For Authors:**

1. According to the problem formulation, the class probability of a label can be non-zero only if it is a leaf node of the node hierarchy, while [1,2] is free of this assumption. Indeed, this condition implicitly assumes that the hierarchy is a 'complete' one: an instance belong to a superclass/internal node means that it must be classified into its leaf descendant, which may be a strict condition. It can be helpful discuss the proposed methods without this assumption.

2. While this framework provides theoretically grounded decoding methods from class probability, the estimated 'flat' class probability may not be perfect and may affect the performance of the decoding methods.

[1]. Ramaswamy, H.,Tewari, A., and Agarwal, S. Convex calibrated surrogates for hierarchical classification. ICML'15

[2]. Cao,Y., Feng,L., and An,B. Consistent hierarchical classification with a generalized metric. AISTATS'24.

**Relation To Broader Scientific Literature:**

The prior related findings, including optimality analyses of certain hierarchical classification metrics and their heuristic decoding, are naturally related to this work since they can be seen as the special case of the proposed method.

**Theoretical Claims:**

The claims are valid and rigorously proved.

---

> ### Author Rebuttal · Authors · 2025-04-01
>
> We warmly thank reviewer k32x for its positive feedback and very insightful comments.
>
> We try below to answer your questions.
>
> *it is encouraged to include the proposed algorithms or provide a more detailed description of them in the main body.*
>
> As the reviewer suggests, we will include the proposed algorithms within the core of the paper.
>
> *According to the problem formulation, the class probability of a label can be non-zero only if it is a leaf node of the node hierarchy, while [1,2] is free of this assumption. Indeed, this condition implicitly assumes that the hierarchy is a 'complete' one: an instance belong to a superclass/internal node means that it must be classified into its leaf descendant, which may be a strict condition. It can be helpful discuss the proposed methods without this assumption.*
>
> We thank the reviewer for this insightful question. We acknowledge that our framework may appear more restrictive than that of [1,2], and this was a key consideration in our research.
>
> One practical way to relax this assumption while remaining within our framework is to modify the hierarchy **before** training. Specifically, we can introduce a "stopping" node (a leaf node) at **each** internal node n of the hierarchy. The likelihood of this stopping node would represent the probability of belonging to any subcategory of node n, **excluding** the children of n already present in the hierarchy. This effectively "completes" the hierarchy, making it more general, though it prevents us from using pre-trained models and also introduces additional technical considerations and modifications to the hierarchy structure. In such a case, our theorems remain valid.
>
> Lastly, we also adopted our current setup to align with recent work in the field, including Karthik et al. (2021) and Jain et al. (2023), and because the considered datasets fulfill the leaf node label assumption (meaning each instance is categorized as a leaf).
>
> We appreciate this discussion and will consider incorporating these clarifications into the paper.
>
> *While this framework provides theoretically grounded decoding methods from class probability, the estimated 'flat' class probability may not be perfect and may affect the performance of the decoding methods.*
>
> Again, this represents a very interesting subject for which we plan to explore further in future work. Specifically, one promising direction is to investigate how to adjust either the optimal strategy or the probability distributions when accounting for the fact that these estimations are imperfect.

---

### Official Review · Reviewer_maE5 · 2025-03-21

**Overall Recommendation:** 4

**Summary:**

The paper introduces a framework for optimal decoding in hierarchical classification, where predictions are structured in a tree-like taxonomy. Unlike standard classification, the severity of errors varies based on the distance in the hierarchy between the predicted and true labels.

Most existing methods use heuristics (like argmax or thresholding) for decoding model outputs, which may not align with the evaluation metric. The authors propose post-hoc optimal decision rules tailored to specific hierarchical evaluation metrics (e.g., hFβ score, Tree Distance Loss, Wu-Palmer).

Experimental results show that the optimal decoding method outperforms heuristics by 1–5%, and up to 10% for mistake severity across various datasets and models. A higher value of the method is demonstrated in uncertain scenarios, like when input images are blurred, where simple decoding baselines struggle.

**Claims And Evidence:**

All main claims are supported by clear and convincing evidence, such as:

1. The proof of algorithm’s optimality, which is novel, and a solid contribution in this area;

2. The experiments showing improvement over the baselines, which follows from the optimality principle;

3. Demonstration of the method at various levels of classifier quality, defined by the image blurring setup, clearly explaining that the algorithm works the best in cases when the original classifier is not accurate.

One comment is regarding the runtime estimations, it would make sense to add the runtime based on the number of inference examples, and with that include the pre-processing complexity of Algorithm 1 to overall computations.

**Essential References Not Discussed:**

All references essential for the paper, such as [Karthik, 2021] and [Ramaswamy, 2015] are discussed. Potentially, the authors could relate the work to structured prediction, as in (Kulesza, 2007): Structured learning with approximate inference.

**Experimental Designs Or Analyses:**

The experimental setup is methodologically sound, diverse, and directly aligned with the paper’s claims. The paper could benefit from tests on larger scale datasets, but it is not necessary.

**Methods And Evaluation Criteria:**

The proposed evaluation methods are sufficient and fully explore the leaf/node decoding strategies, showing advantages and pitfalls (related to reduced method’s value vs simple baselines in case of accurate classifiers).

The only missing item, as acknowledged by the authors, is the exploration of the set of nodes decoding strategy, and the paper could benefit from some experimental study on this. However, this is a more complicated and less practical scenario, so in my opinion it doesn’t diminish the value of the paper.

**Other Comments Or Suggestions:**

No other comments

**Other Strengths And Weaknesses:**

Overall, this appears to be a clear and well–structured paper with solid contribution, walking through all necessary details, with a comprehensive experimental study. Some mentioned shortcomings, such as absence of empirical study for this set of nodes decoding strategy, or not applying the strategy in training, to me do not diminish the paper value and can be considered as future work.

**Questions For Authors:**

No other questions

**Relation To Broader Scientific Literature:**

From the broader context, the paper doesn’t consider learning scenario, only taking a given classifier output (it is acknowledged by the authors). From that perspective there is a pool of works which integrate label hierarchy into the training process, from hierarchical softmax and its variations, to the multi-label scenarios. As mentioned by the authors, it would be really interesting to see how the framework can be incorporated in model training.

**Theoretical Claims:**

All theoretical claims appear to be valid, providing solid results in reasonable assumptions on the loss hierarchy.

---

> ### Author Rebuttal · Authors · 2025-04-01
>
> We warmly thank the reviewer maE5 for their positive feedback and their acknowledgement of the significance of our work.
>
> We answer below to your different points.
>
> *One comment is regarding the runtime estimations, it would make sense to add the runtime based on the number of inference examples, and with that include the pre-processing complexity of Algorithm 1 to overall computations.*
>
> Indeed, varying the total number of test instances and analyzing the overall runtime evolution, including the preprocessing time of Algorithm 1, would provide a more comprehensive evaluation. We acknowledge that Table 9 in Appendix C.3 does not currently account for this. We will conduct this experiment and include the results in Appendix C.3.
>
> *The only missing item, as acknowledged by the authors, is the exploration of the set of nodes decoding strategy, and the paper could benefit from some experimental study on this. However, this is a more complicated and less practical scenario, so in my opinion it doesn’t diminish the value of the paper.*
>
> The reviewer is correct when pointing out that we did not develop **heuristic** decoding strategies for *subset of nodes* decoding, which would have ensured a fair comparison with our **optimal** *subset of nodes* decoding strategies for the $hF_{\beta}$-score of Section 4.3. We did attempt to design a simple heuristic based on thresholding derived from Lemma 4.6, but it resulted in poor performance.
>
> *Kulesza, 2007: Structured learning with approximate inference.*
>
> This is indeed a reference we missed and which is relevant to our problem. We will incorporate it in the paper.

---

### Decision · Program_Chairs · 2025-05-01

**Decision:**

Accept (poster)

**Comment:**

The paper addresses the inference problem in hierarchical multi-class classification. The Authors propose a new inference algorithm that operates on estimated label probabilities and achieves reduced time complexity for loss functions that are hierarchically reasonable.

The paper is in general interesting and its contribution is solid. However, some Reviewers noted that the writing could be significantly improved, as the current presentation makes it difficult to grasp the main message. Additionally, the empirical evaluation would benefit from the inclusion of more datasets. If accepted, the paper should be thoroughly revised to address the main issues raised by the Reviewers.